# List-Level Distribution Coupling with Applications to Speculative Decoding and Lossy Compression

**Joseph Rowan**[1]     **Buu Phan**[1]     **Ashish Khisti**[1]
[1]University of Toronto
{joseph.rowan,truong.phan}@mail.utoronto.ca, akhisti@ece.utoronto.ca

## Abstract

We study a relaxation of the problem of coupling probability distributions — a list of samples is generated from one distribution and an *accept* is declared if any one of these samples is identical to the sample generated from the other distribution. We propose a novel method for generating samples, which extends the Gumbel-max sampling suggested in Daliri et al. [9] for coupling probability distributions. We also establish a corresponding lower bound on the acceptance probability, which we call the *list matching lemma*. We next discuss two applications of our setup. First, we develop a new mechanism for multi-draft speculative sampling that is simple to implement and achieves performance competitive with baselines such as SpecTr [38] and SpecInfer [34] across a range of language tasks. Our method also guarantees a certain degree of *drafter invariance* with respect to the output tokens which is not supported by existing schemes. We also provide a theoretical lower bound on the token level acceptance probability. As our second application, we consider distributed lossy compression with side information in a setting where a source sample is compressed and available to multiple decoders, each with independent side information. We propose a compression technique that is based on our generalization of Gumbel-max sampling and show that it provides significant gains in experiments involving synthetic Gaussian sources and the MNIST image dataset.

## 1   Introduction

Coordinated sampling, where samples are drawn from two distributions in such a way that the probability of the samples being equal is maximized, is a fundamental problem in probability [17, 18, 37, 40] with applications to machine learning [9, 25, 38], data compression [12, 39] and information theory [8, 26]. The general problem is as follows. Consider two parties, Alice and Bob, who wish to generate samples $X$ and $Y$ from distributions $p_X$ and $q_Y$ respectively. For now, we limit our attention to the case where $p_X$ and $q_Y$ are discrete; importance sampling will later allow us to extend the discussion to approximate sampling from continuous distributions as well. A first goal is to construct a scheme that Alice and Bob can follow to maximize the matching or acceptance probability $\Pr[X = Y]$ while ensuring each sample follows the appropriate marginal distribution. If they both have access to $p_X$ and $q_Y$, the problem reduces to that of finding the best joint distribution for $X$ and $Y$ subject to the marginal constraints, which is called a *maximal coupling* between $p_X$ and $q_Y$ [40]. For discrete distributions, such an coupling can be found and the resulting optimal matching probability is $\Pr[X = Y] = 1 - d_{\mathrm{TV}}(p_X, q_Y)$, where $d_{\mathrm{TV}}$ is the total variation distance [17, 37, 40].

However, the requirement that both parties can access $p_X$ and $q_Y$ precludes the use of maximal couplings in many settings where it is desirable to limit communication between Alice and Bob, meaning each can no longer sample with full knowledge of the other's distribution. Sampling methods based on common randomness offer a convenient solution, and have been shown to achieve matching probabilities close to that of the maximal coupling despite their relative simplicity [9]. In particular,

39th Conference on Neural Information Processing Systems (NeurIPS 2025).

if Alice and Bob both sample from $p_X$ and $q_Y$ by applying the Gumbel-max trick to shared random numbers, it is possible to achieve $\Pr[X = Y] \geq (1 - d_{\text{TV}}(p_X, q_Y))/(1 + d_{\text{TV}}(p_X, q_Y))$, which is a lower bound in the communication-free setting [2, 9].

In this paper, we are interested in an extension of the communication-free coupling problem where one of the parties, say Alice, generates $K$ independent samples from $p_X$. Letting the set of Alice's samples be $\{X^{(1)}, \ldots, X^{(K)}\}$, the new matching probability is taken to be $\Pr[Y \in \{X^{(1)}, \ldots, X^{(K)}\}]$; intuitively, Alice's output is said to match Bob's if at least one sample agrees. However, it is not immediately obvious how $Y$ and $X^{(1)}, \ldots, X^{(K)}$ should be sampled in this new setting if we wish to maximize the matching probability. As a solution, we introduce a simple algorithm that we call *Gumbel-max list sampling* (GLS) to generate coupled samples, together with a corresponding lower bound on the acceptance probability. On the practical side, we apply GLS to derive a new algorithm for multi-draft speculative decoding [34, 38], a popular technique for accelerating inference from large language models (LLMs). Later, we also demonstrate an application to distributed lossy compression when independent side information is available at each of $K$ separate decoders, extending the classical formulation of Wyner and Ziv [45]. In summary, our contributions are:

1. We present GLS as a conceptually simple framework for coordinated sampling from discrete distributions when one party produces multiple proposals, extending the single-proposal Gumbel-max coupling technique in Daliri et al. [9], and establish a theoretical lower bound on the matching probability.

2. Based on GLS, we describe a novel multi-draft speculative decoding scheme. Our scheme differs from prior approaches [34, 38], as it does not involve rejection sampling and satisfies a certain notion of *drafter invariance* with respect to the output tokens, for which we give a formal definition.

3. We use GLS to devise a compression technique for distributed lossy source coding where a sample is compressed and sent simultaneously to several decoders, each having access to independent side information. We conduct experiments and show improved rate-distortion performance on Gaussian sources and the MNIST image dataset.

## 2   Related work

**Couplings and coordinated sampling.** From a theoretical perspective, couplings between probability distributions have been used to prove results in probability theory, including limit theorems, convergence results and various inequalities [17, 18, 40]. A reference on these mathematical techniques can be found in Thorisson [40], including algorithms for constructing maximal couplings and their relationship to the total variation distance. More relevant to this paper are techniques that create couplings via common random numbers, which can be applied in very general contexts [13, 14]. In particular, applications to categorical distributions have been studied under the name of weighted coordinate sampling [2, 33]. Recently, Daliri et al. [9] demonstrated how Gumbel-max sampling can be used to generate such couplings and simultaneously introduced the idea of drafter-invariant speculative decoding, though their results and theory were limited to the single-draft case. Part of our contribution is an extension of their work to settings with multiple proposals.

**Lossy source coding via channel simulation.** Lossy compression schemes based on channel simulation rely on similar probabilistic tools to those examined in this paper. While perfect reconstruction can be achieved with an unbounded number of samples, as established by Li and El Gamal [27] through the Poisson functional representation lemma (PFRL), practical methods often resort to importance sampling to communicate approximate samples from continuous distributions [15, 39]. Extending this to the classical problem of source coding with side information at the decoder as posed by Wyner and Ziv [45], Li and Anantharam [26] introduced the Poisson matching lemma (PML), which they used to prove a one-shot version of the Wyner-Ziv theorem along with non-asymptotic variants of other standard results from information theory. However, the PML construction still requires access to an infinite number of samples. Phan et al. [36] adapted the PML to practical settings through importance sampling at the cost of allowing a bounded error probability, introducing a framework called the importance matching lemma (IML). We use GLS to derive an extension where there are several independent decoders with independent side information.

**Speculative decoding.** Speculative decoding, concurrently introduced by Leviathan et al. [25] and Chen et al. [5], is a popular technique for accelerating inference from LLMs. The main idea is to

use a small draft model to parallelize autoregressive decoding from a larger target model, adopting a draft-then-verify approach. Follow-up works have focused on aligning the drafter more closely with the target [32, 49], decreasing the cost of running the draft model [4, 28] or optimizing the length of speculative generation [31]. Most pertinent to our work are multi-draft extensions of speculative decoding where several draft tokens are selected as candidates for verification to increase the expected number of accepted tokens [20]. While maximal couplings are generally used for single-draft methods, they prove intractable in the multi-draft case, where methods based on heuristics or approximations of optimal transport are often used instead [34, 38]. Directly relevant to our contribution is the single-draft drafter-invariant speculative decoding technique recently proposed by Daliri et al. [9], which demonstrates a scheme based on common random numbers instead of rejection sampling. Our work extends this idea to the multi-draft setting.

## 3 Gumbel-max List Sampling

Recalling the setup from section 1, where Bob samples $Y$ from $q_Y$ and Alice samples $X^{(1)}, \ldots, X^{(K)}$ from $p_X$ without communication, how should these samples be generated to maximize the acceptance probability? As a motivating example, take $K = 2$ and let the support of $p_X$ and $q_Y$ be $\{1, 2\}$. Using the Gumbel-max approach from Daliri et al. [9], we could start by choosing shared i.i.d. random numbers $S_1, S_2 \sim \text{Exp}(1)$ and sampling

$$Y = \arg\min\left\{\frac{S_1}{q_Y(1)}, \frac{S_2}{q_Y(2)}\right\} \text{ and } X^{(1)} = \arg\min\left\{\frac{S_1}{p_X(1)}, \frac{S_2}{p_X(2)}\right\}, \tag{1}$$

which would ensure that $\Pr[X^{(1)} = Y] \geq (1 - d_{\text{TV}}(p_X, q_Y))/(1 + d_{\text{TV}}(p_X, q_Y))$ [9]. But, how should we choose $X^{(2)}$? Unfortunately, if we use $S_1$ and $S_2$ again to sample $X^{(2)}$, we would always get $X^{(1)} = X^{(2)}$. Hence, the matching probability would not increase with the number of samples from Alice. An alternative is to create independent random numbers $S_3, S_4 \sim \text{Exp}(1)$ and take

$$X^{(2)} = \arg\min\left\{\frac{S_3}{p_X(1)}, \frac{S_4}{p_X(2)}\right\}. \tag{2}$$

This *does* increase the acceptance probability, but there is now no coupling between $X^{(2)}$ and $Y$. As a result $P[X^{(2)} = Y]$ in general can be very small and we cannot expect a large gain compared to when Alice only generates $X^{(1)}$. The key idea of our GLS algorithm is to instead couple $X^{(1)}$ and $X^{(2)}$ with $Y$ simultaneously using a minimum operation over the shared exponential random variables. More precisely, we choose

$$Y = \arg\min\left\{\frac{\min\{S_1, S_3\}}{q_Y(1)}, \frac{\min\{S_2, S_4\}}{q_Y(2)}\right\},$$

while $X^{(1)}$ and $X^{(2)}$ are sampled as in (1) and (2). Intuitively, our approach coordinates $X^{(1)}$ and $X^{(2)}$ symmetrically with Bob's choice of $Y$, increasing the probability that at least one matches. We now describe GLS for arbitrary $N$ and $K$ and provide a lower bound on the acceptance probability.

We assume without loss of generality that $p_X$ and $q_Y$ are both on the alphabet $\Omega = \{1, \ldots, N\}$. In what follows we will call $p_X$ the proposal or draft distribution and $q_Y$ the target distribution; to simplify the notation we define $p_i := p_X(i)$ and $q_i := q_Y(i)$. Further let $\{\{S_i^{(k)}\}_{i=1}^N\}_{k=1}^K$ be $K$ sets of $N$ i.i.d. random variables, with $S_i^{(k)} \sim \text{Exp}(1)$ for all $i$ and $k$. In practice, we can easily generate the $S_i^{(k)}$'s given a source of uniform random numbers by taking $S_i^{(k)} = -\ln U_i^{(k)}$ where each $U_i^{(k)} \sim \text{Unif}[0, 1]$. The GLS procedure is as follows, and is also summarized in algorithm 1.

1. Select $Y = \arg\min_{1 \leq i \leq N} \min_{1 \leq k \leq K} S_i^{(k)}/q_i$ to generate a sample from $q_Y$.

2. Select $X^{(k)} = \arg\min_{1 \leq i \leq N} S_i^{(k)}/p_i$, $k = 1, \ldots, K$, to generate i.i.d. samples from $p_X$.

The acceptance probability is $\Pr[Y \in \{X^{(1)}, \ldots, X^{(K)}\}]$. Before stating our main theorem, which will give a lower bound on this quantity, the following proposition is needed to establish that GLS returns valid samples from $p_X$ and $q_Y$. The proof can be found in appendix A.1.

---

**Algorithm 1** Gumbel-max List Sampling

---
1: **function** SAMPLEGLS($p_X, q_Y$)
2:     Choose i.i.d. uniform random variables $\{\{U_i^{(k)}\}_{i=1}^N\}_{k=1}^K$ on $[0,1]$.
3:     $Y \leftarrow \arg\min_{1 \leq i \leq N} \min_{1 \leq k \leq K}[-\ln U_i^{(k)}/q_i]$
4:     $X^{(k)} \leftarrow \arg\min_{1 \leq i \leq N}[-\ln U_i^{(k)}/p_i], 1 \leq k \leq K$
5:     **if** $Y \in \{X^{(1)}, \ldots, X^{(K)}\}$ **then return** accept **else return** reject

---

**Proposition 1.** *The procedure described above (GLS) generates samples such that:*

1. $\Pr[X^{(k)} = j] = p_j$ *for all* $k \in \{1, \ldots, K\}$ *and* $j \in \{1, \ldots, N\}$.

2. $\Pr[Y = j] = q_j$ *for all* $j \in \{1, \ldots, N\}$.

With these preliminaries out of the way, we can state our lower bound, which we call the *list matching lemma* (LML). Again, the proof is deferred to appendix A.2.

**Theorem 1** (List matching lemma)**.** *The matching probability is bounded below as*

$$\Pr[Y \in \{X^{(1)}, \ldots, X^{(K)}\}] \geq \sum_{j=1}^N \frac{K}{\sum_{i=1}^N [\max\{q_i/q_j, p_i/p_j\} + (K-1)q_i/q_j]}. \tag{3}$$

*Furthermore, conditioned on* $Y = j$*, we have*

$$\Pr[Y \in \{X^{(1)}, \ldots, X^{(K)}\} \mid Y = j] \geq \left(1 + \frac{q_j}{Kp_j}\right)^{-1}. \tag{4}$$

*Remark* 1. The bound in Equation (3) recovers the exact matching probability in three important cases, namely for degenerate distributions where $p_X$ has all its mass on one element, when $K = 1$ for any pair of distributions, and when $p_X$ and $q_Y$ are identical. These special cases are considered further and proven in appendix A.7.

Theorem 1 can be seen as a direct extension to the discrete case of the PML in Li and Anantharam [26, Lemma 1 on p. 3] and is in fact identical when there is a single proposal. Li and Anantharam did not however consider any precise counterpart to theorem 1, though they did discuss a different list decoding setting with multiple samples from one decoder [26, Remark 10 on p. 10]. From (4), we see that, for any $j$ such that $q_j > 0$ and $p_j > 0$, the matching probability achieved by GLS approaches 1 for large $K$. Moreover, an analog to proposition 1 holds if we instead sample independently from $K$ potentially different draft distributions $p_X^{(1)}, \ldots, p_X^{(K)}$. Since the notation becomes somewhat cumbersome, the formal statement and proof of this extension are relegated to appendix A.3.

## 4 Application to drafter-invariant multi-draft speculative decoding

Using GLS to sample from the output distribution of an LLM suggests a simple but novel algorithm for drafter-invariant multi-draft speculative decoding, analogous to the single-draft procedure from Daliri et al. [9], which uses standard Gumbel-max sampling. In this section, we review the mathematical formulation of multi-draft speculative decoding, define what exactly we mean by drafter invariance and present our new algorithm along with a lower bound on the token acceptance probability.

### 4.1 Problem setup and drafter invariance

Let $\mathcal{M}_b$ and $\mathcal{M}_s^{(k)}$ be the target and draft LLMs respectively, where $1 \leq k \leq K$. $\mathcal{M}_b$ and $\mathcal{M}_s^{(k)}$ take the form of conditional distributions $\mathcal{M}_b(\cdot \mid x_{1:t})$ and $\mathcal{M}_s^{(k)}(\cdot \mid x_{1:t})$, which give the probability of a token appearing at position $t + 1$ given the context $x_{1:t} \coloneqq (x_1, \ldots, x_t)$. For convenience, we will refer to the context as $c$ and assume the alphabet is $\Omega = \{1, \ldots, N\}$. In multi-draft speculative decoding [34, 38], $K$ independent drafts of length $L$, which we denote $X_{1:L}^{(1)}, \ldots, X_{1:L}^{(K)}$, are generated using either batching or tree attention [34] from $\mathcal{M}_s^{(1)}, \ldots, \mathcal{M}_s^{(K)}$ and then verified in parallel by the target model. In practice, the draft tokens are often i.i.d. and only a single draft model $\mathcal{M}_s$ is used [38]. If at least one of the $K$ candidate tokens is accepted at each step, the first such token is appended to the output sequence. If all are rejected, the verification procedure stops and an extra

token is sampled from an appropriately chosen residual distribution. The final output sequence is $Y_{1:\tau}$, where $\tau$ is the number of accepted tokens plus one.

We propose the following notion of drafter invariance, which we later show empirically can be accommodated without decreasing the inference speed. Intuitively, our notion requires that a given set of draft sequences must always induce the same output distribution regardless of the draft models that generated them. To state this condition formally, we write $X_{1:L}^{(k)} = X_{1:L}(\mathcal{M}_s^{(k)})$ in what follows to show the dependence between each draft sequence and the language model that produced it. In appendix B, we further connect our notion to one introduced in Daliri et al. [9] for single-draft speculative decoding only.

**Definition 1** (Conditional drafter invariance). *A multi-draft speculative decoding algorithm is conditionally drafter invariant if, for all $1 \leq j \leq \tau$,*

$$\Pr[Y_{1:j} = y_{1:j} \mid \mathcal{R},\ \boldsymbol{c},\ X_{1:L}(\mathcal{M}_s^{(1)}) = x_{1:L}^{(1)}, \ldots, X_{1:L}(\mathcal{M}_s^{(K)}) = x_{1:L}^{(K)}]$$
$$= \Pr[Y_{1:j} = y_{1:j} \mid \mathcal{R},\ \boldsymbol{c},\ X_{1:L}(\widetilde{\mathcal{M}}_s^{(1)}) = x_{1:L}^{(1)}, \ldots, X_{1:L}(\widetilde{\mathcal{M}}_s^{(K)}) = x_{1:L}^{(K)}]$$

*for any choice of language models $\mathcal{M}_s^{(1)}, \ldots, \mathcal{M}_s^{(K)}$ and $\widetilde{\mathcal{M}}_s^{(1)}, \ldots, \widetilde{\mathcal{M}}_s^{(K)}$.*

In the following section, we will present a conditionally drafter-invariant speculative decoding algorithm based on GLS. Existing schemes, including SpecTr [38] and SpecInfer [34], do not satisfy conditional drafter invariance since their token verification procedures depend explicitly on the draft model's logits. Consequently, modifications affecting the draft model will propagate to the output even if the draft tokens themselves remain unchanged.

### 4.2 An algorithm for drafter-invariant multi-draft speculative decoding

Our approach involves coupled sampling from the draft and target models via GLS. Supposing we wish to use $K$ i.i.d. drafts from a single model $\mathcal{M}_s$, at each decoding step we use the proposal distribution $p_X = \mathcal{M}_s(\cdot \mid \boldsymbol{c})$ and the target distribution $q_Y = \mathcal{M}_b(\cdot \mid \boldsymbol{c})$, then draw coupled samples according to algorithm 1. This procedure is detailed in algorithm 2, which also keeps track of the set of currently viable drafts and accounts for the fact that, in practical speculative decoding implementations, all the draft tokens must be generated autoregressively ahead of time.

The results developed in section 3 allow us to state some important properties of our method. Using the LML directly with the proposal and target distributions as above immediately gives the following lower bound on the token-level acceptance probability.

**Proposition 2.** *Let $\mathcal{M}_s(\cdot \mid \boldsymbol{c})$ be $p_X$ and $\mathcal{M}_b(\cdot \mid \boldsymbol{c})$ be $q_Y$. Then, the probability of accepting at least one token at the current step with $K$ active drafts and context $\boldsymbol{c}$ satisfies*

$$\Pr[\text{accept}] \geq \sum_{j=1}^{N} \frac{K}{\sum_{i=1}^{N}[\max\{q_i/q_j, p_i/p_j\} + (K-1)q_i/q_j]}.$$

Moreover, proposition 1 leads to a guarantee of sequence-level correctness, such that the distribution over all sequences of output tokens matches that of autoregressive inference from the target model. We can also show conditional drafter invariance. Proofs can be found in appendix A.4.

**Proposition 3.** *For any given $\tau$ and for all $1 \leq j \leq \tau$, the output of algorithm 2 satisfies $\Pr[Y_{1:j} = y_{1:j}] = \mathcal{M}_b(y_{1:j} \mid \boldsymbol{c})$. Also, algorithm 2 is conditionally drafter invariant in the sense of definition 1.*

### 4.3 Experiments

**LLM inference with i.i.d. drafts.** Our first experiment evaluates our drafter-invariant multi-draft scheme in a typical LLM inference setting with i.i.d. draws from one draft model. The target model is Qwen 2.5-7 B [47] and, following Leviathan et al. [25], we use a smaller model from the same series, Qwen 2.5-0.5 B, as the drafter. To measure performance across a range of language tasks, prompts are executed from the GSM8K [7], HumanEval [6] and NaturalReasoning [48] datasets. All experiments are run on a single Nvidia RTX6000 Ada GPU with 48 GB of memory. Results are summarized in table 1, where we show the number of accepted tokens for each call to the large model, which is referred to as the block efficiency (BE), along with the percentage speedup in token

| Strategy | GSM8K | | HumanEval | | NaturalReasoning | |
|---|---|---|---|---|---|---|
| | BE | TR (%) | BE | TR (%) | BE | TR (%) |
| SpecInfer [34] | $4.75 \pm 0.00$ | $4.56 \pm 0.20$ | $4.54 \pm 0.01$ | $7.89 \pm 0.85$ | $4.19 \pm 0.01$ | $12.49 \pm 0.56$ |
| SpecTr [38] | $4.78 \pm 0.00$ | $3.75 \pm 0.17$ | $4.56 \pm 0.01$ | $7.09 \pm 0.77$ | $4.22 \pm 0.01$ | $12.89 \pm 0.96$ |
| Our scheme | $4.78 \pm 0.00$ | $4.83 \pm 0.24$ | $4.55 \pm 0.01$ | $8.03 \pm 0.97$ | $4.18 \pm 0.00$ | $12.75 \pm 1.14$ |
| Daliri et al. [9] | $4.16 \pm 0.01$ | $0.63 \pm 0.24$ | $3.69 \pm 0.01$ | $0.04 \pm 0.30$ | $3.32 \pm 0.00$ | $-2.23 \pm 0.29$ |

Table 1: LLM inference with i.i.d. drafts; we use $L = 4$ and $K = 8$ for the multi-draft methods. Token rates (TR) are shown as percentage speedups relative to single-draft speculative decoding, which has mean block efficiency (BE) $4.18$, $3.75$ and $3.43$ on GSM8K, HumanEval and NaturalReasoning respectively.

rate (TR) relative to single-draft speculative decoding [25]. Further experimental results and details are provided in appendix D.1. Our method's token-rate performance matches that of SpecInfer [34] and SpecTr [38] within one standard error of the mean across all cases, even though these schemes do not offer drafter invariance, and exceeds that of the single-draft invariant scheme in Daliri et al. [9]. Our observations agree with recent results in Khisti et al. [20] showing that existing multi-draft algorithms perform almost equally well when i.i.d. drafts are used.

**Guidelines on the choice of $K$.** The choice of $K$ in practical speculative decoding applications presents a crucial tradeoff. While increasing $K$ boosts the acceptance probability in line with proposition 2, it requires more computation to generate and verify the additional drafts, especially when large models are used. Adding more drafts offers performance gains so long as these operations can be done in parallel, yet there comes a point when available resources become oversubscribed and slowdowns ensue if $K$ is made too large. Furthermore, as memory requirements scale linearly with $K$, VRAM capacity can be become a limiting factor, all the more so when KV caching is enabled. We find that increasing $K$ up to 6 and in many cases 8 gives consistent wall-clock speedups, consistent with previous work on multi-draft speculative decoding [19, 34, 38]; concrete measurements for different values of $K$ may be found in appendix D.1.

**LLM inference with diverse drafts.** Our second experiment examines a more challenging scenario where several different drafters are used, each misaligned with the target. Using a collection of draft models, each exhibiting some degree of diversity during the training process, can naturally lead to improved efficiency. For example, training or fine-tuning the draft models on different tasks may alleviate the challenge of finding a single general-purpose, fast drafter. In our present paper we introduce diversity across different models as well as potential misalignment with the target model by varying the sampling temperature.

Specifically, we focus on experiments with $K = 2$ drafts where the temperature of each drafter is changed independently and is mismatched to the target, which itself has temperature $2.0$. SpecTr verification can no longer be used because it is specialized to identically distributed proposals; comparisons to SpecInfer, which is the dominant verification strategy in concurrent empirical work [28, 29], remain possible. As shown in table 2, our method outperforms SpecInfer with respect to token rates on GSM8K [7], HumanEval [6] and MBPP [1] in the mismatched draft setting. Moreover,

---

**Algorithm 2** Drafter-invariant multi-draft speculative decoding

1: Choose $L + 1$ sets of i.i.d. uniform random variables $\{\{U_i^{(j,k)}\}_{i=1}^N\}_{k=1}^K$, where $1 \leq j \leq L + 1$.
2: **for** $j = 1, \ldots, L$ **do**
3:     **for** $k = 1, \ldots, K$ **do in parallel**
4:         $p_X^{(j,k)} \leftarrow \mathcal{M}_s(\cdot \mid X_{1:(j-1)}^{(k)}, \boldsymbol{c})$, $X_j^{(k)} \leftarrow \arg\min_{1 \leq i \leq N}[-\ln U_i^{(j,k)}/p_i^{(j,k)}]$
5: **for** $j = 1, \ldots, L + 1$ and $k = 1, \ldots, K$ **do in parallel**
6:     $q_Y^{(j,k)} \leftarrow \mathcal{M}_b(\cdot \mid X_{1:(j-1)}^{(k)}, \boldsymbol{c})$
7: Let the set of active drafts be $\mathcal{S} = \{1, \ldots, K\}$.
8: **for** $j = 1, \ldots, L$ **do**
9:     $Y_j \leftarrow \arg\min_{1 \leq i \leq N} \min_{k \in \mathcal{S}}[-\ln U_i^{(j,k)}/q_i^{(j,k)}]$
10:     **for** $k \in \mathcal{S}$ **do**
11:         **if** $X_j^{(k)} \neq Y_j$ **then** $\mathcal{S} \leftarrow \mathcal{S} \setminus \{k\}$
12:     **if** $\mathcal{S} = \emptyset$ **then return** $Y_1, \ldots, Y_j$
13: $Y_{L+1} \leftarrow \arg\min_{1 \leq i \leq N} \min_{k \in \mathcal{S}}[-\ln U_i^{(L+1,k)}/q_i^{(L+1,k)}]$
14: **return** $Y_1, \ldots, Y_{L+1}$

| Strategy | Tmp. 1/2 | GSM8K | | HumanEval | | MBPP | |
|---|---|---|---|---|---|---|---|
| | | BE | TR (%) | BE | TR (%) | BE | TR (%) |
| SpecInfer [34] | 0.5/1.0 | $4.26 \pm 0.02$ | $0.06 \pm 1.02$ | $3.57 \pm 0.02$ | $-1.96 \pm 0.67$ | $3.66 \pm 0.01$ | $-1.87 \pm 0.79$ |
| | 1.0/0.5 | $4.44 \pm 0.03$ | $4.57 \pm 1.80$ | $3.80 \pm 0.03$ | $4.13 \pm 1.60$ | $3.90 \pm 0.02$ | $4.77 \pm 0.55$ |
| | 1.0/1.0 | $4.51 \pm 0.02$ | $6.02 \pm 1.35$ | $3.87 \pm 0.02$ | $6.32 \pm 0.36$ | $3.95 \pm 0.02$ | $5.17 \pm 1.13$ |
| Our scheme | 0.5/1.0 | $4.75 \pm 0.02$ | $11.50 \pm 1.78$ | $4.00 \pm 0.01$ | $9.80 \pm 0.82$ | $3.94 \pm 0.02$ | $5.64 \pm 0.66$ |
| | 1.0/0.5 | $4.75 \pm 0.02$ | $11.40 \pm 1.58$ | $3.96 \pm 0.02$ | $8.77 \pm 0.99$ | $3.96 \pm 0.02$ | $5.99 \pm 1.01$ |
| | 1.0/1.0 | $4.83 \pm 0.02$ | $13.68 \pm 1.67$ | $4.08 \pm 0.02$ | $12.15 \pm 0.83$ | $4.08 \pm 0.01$ | $8.57 \pm 0.60$ |

Table 2: LLM inference with diverse drafts; we use $L = 5$, $K = 2$ and the target temperature is 2.0. The temperatures of drafters 1 and 2 vary and are reported in the second column. Token rates (TR) are shown as percentage speedups relative to single-draft speculative decoding with drafter temperature 1.0, which has mean block efficiency (BE) 4.28, 3.65 and 3.71 on GSM8K, HumanEval and MBPP respectively.

our scheme is less sensitive to the order in which the drafts appear, whereas SpecInfer's recursive rejection scheme [34] favors coupling with the first proposal — comparing the first and second rows of table 2, SpecInfer often exhibits slower token rates than single-draft speculative decoding when the first drafter's temperature deviates the most from the target in the $0.5/1.0$ configuration, yet this weakness disappears when the drafts' order is swapped. Again, additional experimental results can be found in appendix D.1.

## 4.4 Limitations

While our GLS-based speculative decoding algorithm as presented in algorithm 2 mostly functions as a drop-in replacement for standard speculative decoding [25], care must be taken when the vocabulary size is large. As in other multi-draft algorithms [34, 38], memory requirements to store the draft and target logits for each step scale as $\mathcal{O}(NK)$, while we also need to generate a similarly large quantity of shared random numbers and compute the $\arg\min$ across these sets. In practice, enabling top-K sampling significantly reduces this burden by allowing us to throw away all but $M \ll N$ high-probability tokens. Our experiments use the full Qwen 2.5 set of $151\,936$ possible tokens, demonstrating that large vocabularies are well-handled by GLS in a real-world inference setting. On the other hand, speculative decoding may fail to enhance performance with a poorly aligned draft model or extreme temperature mismatch; our approach can benefit from the proliferation of complementary techniques that encourage better draft-target alignment to obtain more robust performance [46, 49].

## 5 Application to lossy compression with side information

We now present an application of GLS to a distributed lossy compression problem with one encoder and $K$ decoders, where each has access to independent side information, thereby extending the compression technique presented in Phan et al. [36] to a list decoding setting. Our setup is distinct from a more common multi-decoder extension of Wyner-Ziv coding where the decoders access side information with different levels of statistical correlation with the source, and are required to produce reconstructions of varying fidelity [41]. In contrast we allow multiple chances to decode the compressed source representation using independent side information samples, related to list decoding approaches in information theory [11]. We are motivated by practical scenarios where shared data is compressed and sent to each decoder for later use in a downstream task, and overall success is predicated on at least one decoder completing the task correctly. Conceptually, this mirrors the set-membership definition of matching probability used in theorem 1.

As a concrete example, consider a decentralized vision-based aircraft detection system where a base station captures a large-scale image of the sky, while several remote sub-stations obtain images with a narrower field of view from particular locations. After a compressed representation of the base station's image is broadcast, each sub-station uses its private image as side information to reconstruct the global view, then uses this reconstruction along with the local image as input to a detection algorithm. The system as a whole is declared successful if at least one sub-station detects the aircraft overhead, reflecting the standard notion of detection probability in distributed detection theory [42]. In this paper, we limit our discussion to the intermediate reconstruction error, leaving the mechanics of any downstream detection to future work as these will depend on the specifics of the application.

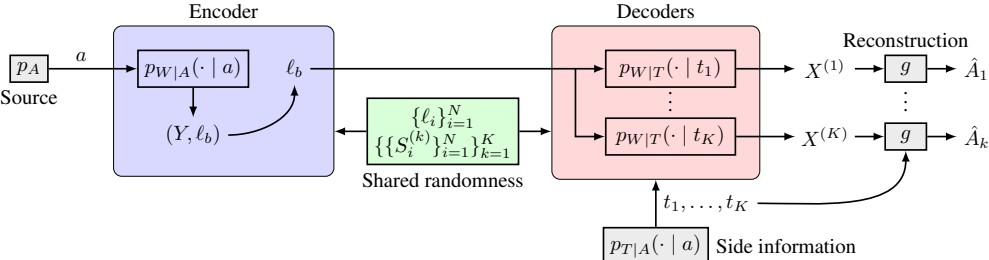

Figure 1: Problem setting for lossy compression with side information at the decoder.

## 5.1 Problem setup and coding scheme

Figure 1 gives an overview of our coding scheme. The encoder observes a realization of $A \sim p_A$ from the source, which should be broadcast identically to $K$ decoders at a rate of $R$ bits per sample via a message $M$. Decoder $k$, where $1 \leq k \leq K$, further observes the side information $T_k \sim p_{T|A}$ and combines this with the message sent by the encoder to produce an output $W_k$ that aims to follow the conditional distribution $p_{W|A}$ and marginal $p_W$. The $T_k$'s are taken to be i.i.d. and $p_{W|A}$ is usually chosen to satisfy an expected distortion constraint $\mathrm{E}[d(A, \hat{A})]$, where $d$ is a distortion metric and $\hat{A}$ is the final reconstruction, calculated as $\hat{A} = g(W, T)$ for some decoding function $g$. We limit our attention here to discrete probability distributions, although we further show in appendix C how our scheme can be extended to continuous distributions through importance sampling.

Following Phan et al. [36], we let $p_{W|T}$ be the conditional distribution of $W$ given the side information $T$, which can readily be calculated as $p_{W|T}(w \mid t) = \sum_{a'} p_{W|A}(w \mid a') p_{A|T}(a' \mid t)$. Assuming $W \in \{1, \dots, N\}$, we generate $N$ uniform integers $\ell_1, \dots, \ell_N$ i.i.d. on $\{1, \dots, L_{\max}\}$, where $L_{\max}$ is a positive integer. We define the random tuple $B = (W, \ell)$ distributed according to $p_B(w, \ell) = p_W(w)/L_{\max}$ and the associated samples $B_i = (i, \ell_i)$ for $1 \leq i \leq N$, thereby covering the full sample space of $W$. Using the procedure described in section 3, we also let $\{\{S_i^{(k)}\}_{i=1}^N\}_{k=1}^K$ be $K$ sets of $N$ i.i.d. $\mathrm{Exp}(1)$ random variables. To sample from $p_{B|A}$, given that $A = a$ is observed, the encoder selects an index $Y$ given by

$$Y = \arg\min_{1 \leq i \leq N} \min_{1 \leq k \leq K} \frac{S_i^{(k)}}{p_{B|A}(B_i \mid a)} = \arg\min_{1 \leq i \leq N} \min_{1 \leq k \leq K} \frac{S_i^{(k)}}{p_{W|A}(i \mid a)}.$$

Supposing $Y = j$, the encoder transmits the message $M = \ell_j$ so that decoder $k$ has access to the tuple $Z_k = (T_k, \ell_j)$. The rate is therefore $R = \log L_{\max}$ bits, as this is the bit-width of $M$. The target distribution used by each decoder is taken to be $p_{B|Z}(w, \ell \mid t, \ell_j) = p_{W|T}(w \mid t)\mathbb{1}\{\ell = \ell_j\}$. Finally, given $T_k = t_k$, decoder $k$ selects the index $X^{(k)}$ according to

$$X^{(k)} = \arg\min_{1 \leq i \leq N} \frac{S_i^{(k)}}{p_{B|Z}(B_i \mid t_k, \ell_j)} = \arg\min_{1 \leq i \leq N} \frac{S_i^{(k)}}{p_{W|T}(i \mid t_k)\mathbb{1}\{\ell_i = \ell_j\}}.$$

## 5.2 Bound on the error probability

To apply GLS in the compression setting, our analysis must include the source, side information and message sent by the encoder. We therefore start by outlining a modified GLS procedure that allows the encoder to condition its output on the source $A$, while the decoder's output is conditioned on a set of related random variables $Z_1^K := \{Z_k\}_{k=1}^K$. As in section 3, we consider discrete marginal distributions $p_X$ and $q_Y$ on $\Omega = \{1, \dots, N\}$, but now assume they are related by arbitrary conditional distributions $q_{Y|A}$, $p_{Z|Y,A}$ and $p_{X|Z}$. The common randomness $\{\{S_i^{(k)}\}_{i=1}^N\}_{k=1}^K$ is as before. We also define $q_j(a) := q_{Y|A}(j \mid a)$ and $p_j(z) := p_{X|Z}(j \mid z)$. The new sampling strategy is as follows:

1. Upon observing $A = a$, the encoder selects $Y = \arg\min_{1 \leq i \leq N} \min_{1 \leq k \leq K} S_i^{(k)}/q_i(a)$ to generate a sample from $q_{Y|A}$.

2. Given $Y = j$, we sample $Z_1, \dots, Z_K$ i.i.d. according to $p_{Z|Y,A}(\cdot \mid j, a)$.

3. Given $Z_k = z_k$, decoder $k$ selects $X^{(k)} = \arg\min_{1 \leq i \leq N} S_i^{(k)}/p_i(z_k)$.

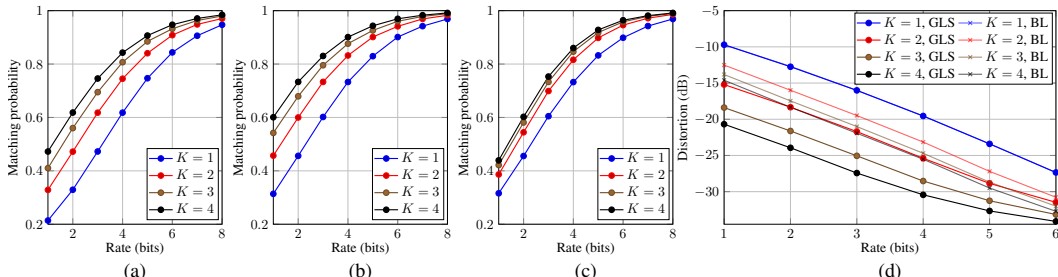

Figure 2: Experiments on a Gaussian source. (a)–(c): Matching probability, from left to right: GLS without side information, GLS with side information, baseline with side information. The baseline does not benefit from multiple decoders without side information. (d): Rate-distortion curves for GLS and baseline (BL) schemes.

We also derive a corresponding extension of the LML to bound the matching probability of the conditional GLS scheme. Intuitively, while the LML in theorem 1 deals with the case where no communication is permitted, here we control the amount of communication by choosing $p_{Z|Y,A}$.

**Theorem 2** (Conditional LML). *Using the strategy above, the error probability satisfies*

$$\Pr[Y \in \{X^{(1)}, \dots, X^{(K)}\} \mid Y = j,\ A = a,\ Z_1^K = z_1^K] \geq \sum_{k=1}^{K} \left( K + \frac{q_j(a)}{p_j(z_k)} \right)^{-1}.$$

Theorem 2 follows from the LML by realizing that $\{\{S_i^{(k)}\}_{i=1}^N\}_{k=1}^K \to (Y, A) \to Z_1^K$ forms a Markov chain, and therefore the $Z_k$'s are conditionally independent of the shared randomness given $Y$ and $A$. The full proof is given in appendix A.5. Using theorem 2, we now have the following bound on the error probability of our coding scheme. The proof can be found in appendix A.6.

**Proposition 4.** *For the coding scheme in section 5.1, the error probability is bounded above as*

$$\Pr[Y \notin \{X^{(1)}, \dots, X^{(K)}\}] \leq 1 - \mathrm{E}_{A,W,T} \left[ \left( 1 + \frac{2^{i(W;A|T)}}{K L_{\max}} \right)^{-1} \right] \tag{5}$$

*where $i_{W,A|T}(w; a \mid t) = \log(p_{W|A}(w \mid a)/p_{W|T}(w \mid t))$ is the conditional information density.*

**Synthetic Gaussian source.** As a canonical example, we consider a Gaussian source $A \sim \mathcal{N}(0, 1)$. The side information is $T_k = A + \zeta_k$, where $\zeta_k \sim \mathcal{N}(0, \sigma_{T|A}^2)$ and $\sigma_{T|A}^2 = 0.5$. The encoder's target distribution follows $p_{W|A}(\cdot \mid a) = \mathcal{N}(a, \sigma_{W|A}^2)$, with $\sigma_{W|A}^2$ interpreted as the distortion permitted by the compression scheme. Note that since the target distribution is now continuous, it is not possible to achieve perfect reconstruction with a finite number of samples. However, we can get arbitrarily close by using importance sampling and then applying GLS to an empirical distribution defined by the importance weights, as detailed in appendix C. Moreover, a closed form for the decoder's target distribution $p_{W|T}$ exists in the Gaussian case, and turns out to be $p_{W|T}(\cdot \mid t) = \mathcal{N}(t/\sigma_T^2, \sigma_W^2 - 1/\sigma_T^2)$. Concerning the reconstruction function $g$, decoder $k$ forms the minimum mean squared error (MMSE) estimate of $A$ given $T_k$ and $W_k$, and we then choose the estimate with the least distortion among all decoders. The target distribution and MMSE estimator are both derived in appendix D.3, along with other experimental details.

Figure 2 illustrates how the matching probability increases in both the rate and the number of decoders, while the observed distortion decreases. We also show comparisons to a baseline scheme where all decoders share the same set of random numbers; we are not aware of any other competitive baseline for the list decoding setting considered here. GLS-based decoding improves substantially over the baseline when $K > 1$, particularly at low rates, as the probability of matching with the encoder at least once is increased. For $K = 1$, both methods are equivalent to the single-decoder technique described in Phan et al. [36, p. 6].

**Lossy compression on MNIST.** We now apply our technique to distributed image compression [35, 43] using the MNIST dataset [24]. In our experiments, the right half of each image is the source while the side information is a randomly selected $7 \times 7$ crop from the left half. We adopt a neural compression technique similar to that in Phan et al. [36] using a $\beta$-variational autoencoder

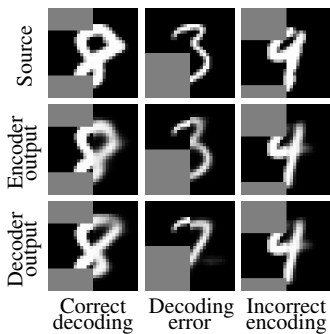

Figure 3: Examples showing success and failure modes of our compression scheme on MNIST.

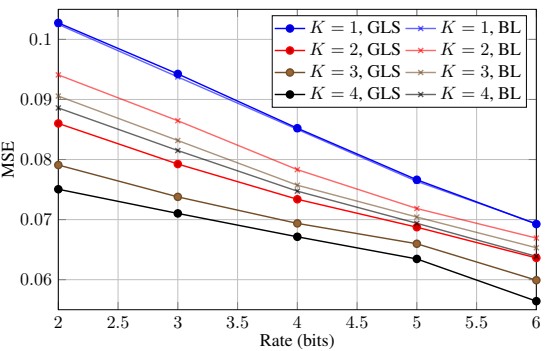

Figure 4: Rate-distortion curves on MNIST for GLS and baseline (BL) schemes.

($\beta$-VAE) [16]. Details of our setup are given in appendix D.4, where we also describe how the $\beta$-VAE architecture fits into our coding scheme as laid out in section 5.1.

Figure 3 shows some representative examples from our image compression pipeline. Errors may occur at the encoder, decoder or both; cases where the encoder's output is correct but the decoder's is not are caused by error events of the type dealt with in (5). Our scheme's rate-distortion performance improves with the number of decoders as shown in figure 4; comparisons to the same baseline used in the previous experiment, with only one set of random numbers, demonstrate consistent improvements particularly at lower rates.

## 5.3 Limitations

Unlike speculative decoding applications, which most often operate in single-GPU or homogeneous multi-GPU environments [4, 28], lossy compression algorithms may conceivably see deployment in distributed systems and we therefore must consider how to communicate the shared randomness between the encoder and decoder. This is usually done by sharing or agreeing upon a random seed before starting the compression procedure, the cost of which may be amortized across many transmissions [39]. Furthermore, our error probability analysis assumes that floating-point operations have the same deterministic behavior at both the encoder and decoder, which may be violated in practice. We emphasize that these limitations are not unique to GLS but rather applicable to methods based on Gumbel-max sampling or common randomness more generally, which benefit from numerous implementation techniques within the machine learning [22], compression [36, 39] and coordinated sampling [2, 8] literature.

## 6 Conclusion

We studied the problem of coupling probability distributions without communication when several samples are available from one of the distributions, and introduced the GLS algorithm to draw coupled samples in this setting along with a lower bound on the resulting acceptance probability. These results were then used to derive novel algorithms for drafter-invariant multi-draft speculative decoding and lossy compression with side information. Avenues for future work include applying GLS with importance sampling to generalize speculative decoding to models with continuous sample spaces such as diffusion models [3]. Furthermore, an alternative relaxation of distribution coupling might allow both parties to generate a list and declare an *accept* if the intersection between the lists is nonempty. Finding relevant practical applications as well as efficient sampling techniques for such a generalization is another interesting direction for further research. We have made the code for all our experiments available at `https://github.com/jsrowan/MultiDraftSpeculativeDecoding`.

## 7 Acknowledgements

We thank Hitachi Solutions for funding the present research. Resources used in preparing this research were moreover provided, in part, by the Province of Ontario, the Government of Canada through CIFAR, and companies sponsoring the Vector Institute (`www.vectorinstitute.ai/partnerships/`).

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

# Appendices

## A  Proofs

In this section we prove the results in the main paper. For clarity, each theorem is restated and then a proof is given.

### A.1  Proof of proposition 1

**Proposition 1.** *The GLS procedure, as described in section 3, generates samples such that:*

1. $\Pr[X^{(k)} = j] = p_j$ *for all* $k \in \{1, \ldots, K\}$ *and* $j \in \{1, \ldots, N\}$.
2. $\Pr[Y = j] = q_j$ *for all* $j \in \{1, \ldots, N\}$.

*Proof.* Since the $S_i^{(k)}$'s are i.i.d., the $X^{(k)}$'s will be also and it therefore suffices to check the distribution of $X^{(1)}$. Note that

$$X^{(1)} = j \implies \frac{S_j^{(1)}}{p_j} \leq \frac{S_i^{(1)}}{p_i} \ \forall \, i \neq j$$
$$\implies S_j^{(1)} \leq \min_{i \neq j} \frac{S_i^{(1)}}{p_i/p_j}.$$

The left-hand side is an exponential random variable with parameter $\lambda = 1$, and the right-hand side is an independent exponential random variable with parameter $\lambda = \sum_{i \neq j} p_i/p_j$. So,

$$\Pr[X^{(1)} = j] = \frac{1}{1 + \sum_{i \neq j} p_i/p_j} = p_j$$

as required. Next, we look at the distribution of $Y$. Define $S_j^* = \min_{1 \leq k \leq K} S_j^{(k)}$. Then,

$$Y = j \implies \frac{S_j^*}{q_j} \leq \frac{S_i^*}{q_i} \ \forall \, i \neq j$$
$$\implies S_j^* \leq \min_{i \neq j} \frac{S_i^*}{q_i/q_j}.$$

On the left-hand side, $S_j^*$ is an exponential random variable with parameter $\lambda = K$, while the right-hand side is an independent exponential random variable with parameter $\lambda = K \sum_{i \neq j} q_i/q_j$. Finally,

$$\Pr[Y = j] = \frac{K}{K + K \sum_{i \neq j} q_i/q_j} = q_j.$$

$\square$

### A.2  Proof of theorem 1

**Theorem 1** (List matching lemma)**.** *The matching probability is bounded below as*

$$\Pr[Y \in \{X^{(1)}, \ldots, X^{(K)}\}] \geq \sum_{j=1}^{N} \frac{K}{\sum_{i=1}^{N} [\max\{q_i/q_j, p_i/p_j\} + (K-1)q_i/q_j]}. \tag{3}$$

*Furthermore, conditioned on* $Y = j$, *we have*

$$\Pr[Y \in \{X^{(1)}, \ldots, X^{(K)}\} \mid Y = j] \geq \left(1 + \frac{q_j}{Kp_j}\right)^{-1}. \tag{4}$$

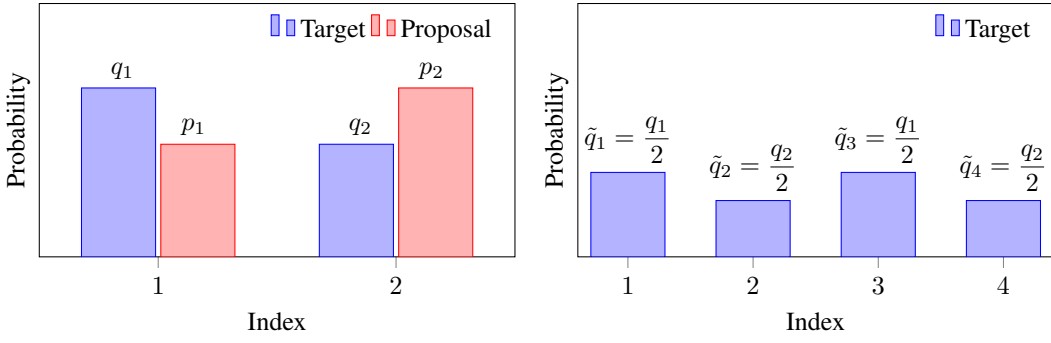

(a) Target and proposal distributions.    (b) Augmented target distribution.

Figure 5: Distributions used in the proof of theorem 1 in the case of $K = 2$ and $N = 2$.

*Proof.* To analyze the matching probability, we conceptualize the scheme slightly differently as follows. Instead of taking the minimum over $1 \leq k \leq K$ to obtain $\{S_i^*\}_{i=1}^N$ like in the proof of proposition 1, we form a flattened sequence of the $S_i^{(k)}$'s by defining

$$\{T_i\}_{i=1}^{NK} = \{S_1^{(1)}, \ldots, S_N^{(1)}, S_1^{(2)}, \ldots, S_N^{(2)}, \ldots, S_1^{(K)}, \ldots, S_N^{(K)}\}.$$

We also introduce an augmented target distribution $\tilde{q}_{\tilde{Y}}$ on the extended alphabet $\tilde{\Omega} = \{1, \ldots, KN\}$ defined by the probabilities

$$(\tilde{q}_1, \ldots, \tilde{q}_{KN}) = \left(\frac{q_1}{K}, \ldots, \frac{q_N}{K}, \ldots, \frac{q_1}{K}, \ldots, \frac{q_N}{K}\right) \tag{6}$$

and corresponding output $\tilde{Y}$. The setup is visualized in figure 5 for the simplest case of $N = 2$ and $K = 2$. There, $Y = 1$ for example corresponds to either $\tilde{Y} = 1$ or $\tilde{Y} = 3$. In general,

$$Y = j \iff \tilde{Y} = j + (k-1)N \text{ for some } k \in \{1, \ldots, K\}.$$

By symmetry of the construction,

$$\Pr[Y \in \{X^{(1)}, \ldots, X^{(K)}\}] = \sum_{j=1}^N \Pr[Y = j, \; j \in \{X^{(1)}, \ldots, X^{(K)}\}]$$

$$= \sum_{j=1}^N K \Pr[\tilde{Y} = j, \; j \in \{X^{(1)}, \ldots, X^{(K)}\}]$$

$$\geq \sum_{j=1}^N K \Pr[\tilde{Y} = j, \; X^{(1)} = j]. \tag{7}$$

Since the analysis will be the same for any $j$, we now focus on finding the probability of the event $\tilde{Y} = 1$ and $X^{(1)} = 1$

$$\implies \frac{T_1}{\tilde{q}_1} \leq \min_{2 \leq i \leq KN} \frac{T_i}{\tilde{q}_i} \text{ and } \frac{T_1}{p_1} \leq \min_{2 \leq i \leq N} \frac{T_i}{p_i}$$

$$\implies T_1 \leq \min\left\{\frac{T_2}{\max\{\tilde{q}_2/\tilde{q}_1, p_2/p_1\}}, \ldots, \frac{T_N}{\max\{\tilde{q}_N/\tilde{q}_1, p_N/p_1\}}, \frac{T_{N+1}}{\tilde{q}_{N+1}/\tilde{q}_1}, \ldots, \frac{T_{KN}}{\tilde{q}_{KN}/\tilde{q}_1}\right\}.$$

The right-hand side is an exponential random variable independent of the left-hand side. If its parameter is $\lambda$ then, taking into account the definition of the $\tilde{q}_i$'s, we have

$$1 + \lambda = \sum_{i=1}^N \max\left\{\frac{q_i}{q_1}, \frac{p_i}{p_1}\right\} + \sum_{k=2}^K \sum_{i=1}^N \frac{q_i}{q_1}$$

$$= \sum_{i=1}^N \left[\max\left\{\frac{q_i}{q_1}, \frac{p_i}{p_1}\right\} + (K-1)\frac{q_i}{q_1}\right].$$

Therefore, by the properties of independent exponential random variables,

$$\Pr[\tilde{Y} = 1, \ X^{(1)} = 1] = \frac{1}{\sum_{i=1}^{N}[\max\{q_i/q_1, p_i/p_1\} + (K-1)q_i/q_1]}. \tag{8}$$

Combining with (7) establishes the bound in (3). We now turn our attention to (4) and find

$$\Pr[Y = j, \ Y \in \{X^{(1)}, \ldots, X^{(K)}\}] = \Pr[Y = j, \ j \in \{X^{(1)}, \ldots, X^{(K)}\}]$$
$$\geq K \Pr[\tilde{Y} = j, \ X^{(1)} = j] \tag{9}$$

by the same symmetry argument used to show (7). Since our choice of $j = 1$ in establishing (8) was arbitrary, we can apply that result for each $j$ to get

$$\Pr[Y = j, \ Y \in \{X^{(1)}, \ldots, X^{(K)}\}] \geq \frac{K}{\sum_{i=1}^{N}[\max\{q_i/q_j, p_i/p_j\} + (K-1)q_i/q_j]}.$$

Finally,

$$\Pr[Y \in \{X^{(1)}, \ldots, X^{(K)}\} \mid Y = j] = \Pr[Y = j, \ Y \in \{a^{(1)}, \ldots, a^{(K)}\}]/\Pr[Y = j]$$
$$\geq \frac{K}{q_j \sum_{i=1}^{N}[\max\{q_i/q_j, p_i/p_j\} + (K-1)q_i/q_j]}$$
$$= \frac{K}{K + q_j \sum_{i=1}^{N} \max\{0, p_i/p_j - q_i/q_j\}}$$
$$\geq \frac{K}{K + q_j/p_j}$$
$$= \left(1 + \frac{q_j}{Kp_j}\right)^{-1}$$

$\square$

As an aside, we can now easily find a related relaxed version of (3) by means of the relationship

$$\Pr[Y \in \{X^{(1)}, \ldots, X^{(2)}\}] = \sum_{j=1}^{N} \Pr[Y = j] \Pr[Y \in \{X^{(1)}, \ldots, X^{(2)}\} \mid Y = j]$$
$$\geq \sum_{j=1}^{N} q_j \left(1 + \frac{q_j}{Kp_j}\right)^{-1}$$

since $\Pr[Y = j] = q_j$ by proposition 1.

### A.3 Extension of proposition 1 to non-identically distributed proposals

We briefly consider the case where the proposals are drawn independently from $K$ different distributions. Let these distributions be $p_X^{(1)}, \ldots, p_X^{(K)}$ and define $p_i^{(K)} := p_X^{(K)}(i)$ for convenience. In this setting, $X^{(k)}$ is sampled from the corresponding $p_X^{(k)}$, but the sampling procedure and the common randomness are otherwise identical to the setup in proposition 1. We then have the following extension of that result.

**Proposition 5.** *The procedure described above generates samples such that:*

1. $\Pr[X^{(k)} = j] = p_j^{(k)}$ *for all* $k \in \{1, \ldots, K\}$ *and* $j \in \{1, \ldots, N\}$.

2. $\Pr[Y = j] = q_j$ *for all* $j \in \{1, \ldots, N\}$.

*Proof.* The proof is very similar to that of proposition 1. We start by checking the distribution of any $X^{(k)}$ for $1 \leq k \leq K$. Note that

$$X^{(k)} = j \implies \frac{S_j^{(k)}}{p_j^{(k)}} \leq \frac{S_i^{(k)}}{p_i^{(k)}} \ \forall \ i \neq j$$
$$\implies S_j^{(k)} \leq \min_{i \neq j} \frac{S_i^{(k)}}{p_i^{(k)}/p_j^{(k)}}.$$

The left-hand side is an exponential random variable with parameter $\lambda = 1$, and the right-hand side is an independent exponential random variable with parameter $\lambda = \sum_{i \neq j} p_i^{(k)} / p_j^{(k)}$. So,

$$\Pr[X^{(k)} = j] = \frac{1}{1 + \sum_{i \neq j} p_i^{(k)} / p_j^{(k)}} = p_j^{(k)}$$

as required. Next, note that the selection procedure for $Y$ does not involve the $p_i^{(k)}$'s at all. Because the introduction of these probabilities is the only deviation from the setting of proposition 1, the correctness of $Y$'s distribution follows immediately from that result. □

### A.4 Proof of proposition 3

**Proposition 3.** *For any given $\tau$ and for all $1 \leq j \leq \tau$, the output of algorithm 2 satisfies $\Pr[Y_{1:j} = y_{1:j}] = \mathcal{M}_b(y_{1:j} \mid \mathbf{c})$. Also, algorithm 2 is conditionally drafter invariant in the sense of definition 1.*

*Proof.* We start by proving the first part of the proposition, which establishes sequence-level correctness. For convenience and to connect the setup more easily to GLS, we first abstract the details of Gumbel-max sampling by defining sets of independent random variables $\{\{S_i^{(j,k)}\}_{i=1}^N\}_{k=1}^K$ for all $1 \leq j \leq L$, where $S_i^{(j,k)} = -\ln U_i^{(j,k)}$. We then have that each $S_i^{(j,k)} \sim \text{Exp}(1)$. Further let $\mathcal{S}_{j-1}$ be the set of viable candidate drafts immediately before sampling $Y_j$, which allows us to keep track of changes to $\mathcal{S}$ in algorithm 2 during the proof. When the maximum draft length is $L$, we have $\tau \in \{1, \ldots, L+1\}$ due to the possibility of selecting an extra token if the full draft sequence is accepted. For any $\tau$, we need to verify the distribution of $Y_1, \ldots, Y_\tau$, which we do by induction. First, consider $Y_1$. Before the first step, $\mathcal{S}_0 = \{1, \ldots, K\}$, so algorithm 2 makes the selection

$$Y_1 = \arg\min_{1 \leq i \leq N} \min_{1 \leq k \leq K} \frac{S_i^{(1,k)}}{q_i^{(1,k)}} = \arg\min_{1 \leq i \leq N} \min_{1 \leq k \leq K} \frac{S_i^{(1,k)}}{\mathcal{M}_b(i \mid \mathbf{c})}.$$

The second equality follows because the algorithm assigns $q_i^{(1,k)} = \mathcal{M}_b(i \mid \mathbf{c})$ for all $k$. Then, $\Pr[Y_1 = y_1] = \mathcal{M}_b(y_1 \mid \mathbf{c})$ for all $y_1 \in \Omega$, by proposition 1. Next, the rejection loop keeps only the drafts $k$ satisfying $X_1^{(k)} = Y_1$. Hence, $X_1^{(k)} = Y_1$ for any $k \in \mathcal{S}_1$.

Next we look at the distribution of $Y_2$. Here, we get

$$Y_2 = \arg\min_{1 \leq i \leq N} \min_{k \in \mathcal{S}_1} \frac{S_i^{(2,k)}}{q_i^{(2,k)}} = \arg\min_{1 \leq i \leq N} \min_{k \in \mathcal{S}_1} \frac{S_i^{(2,k)}}{\mathcal{M}_b(i \mid X_1^{(k)}, \mathbf{c})}.$$

However, from the previous step we know that $X_1^{(k)} = Y_1$ for any $k \in \mathcal{S}_1$. Therefore, $\Pr[Y_2 = y_2 \mid Y_1 = y_1] = \mathcal{M}_b(y_2 \mid y_1, \mathbf{c})$ by proposition 1 and consequently

$$\Pr[Y_{1:2} = y_{1:2}] = \mathcal{M}_b(y_2 \mid y_1, \mathbf{c}) \mathcal{M}_b(y_1 \mid \mathbf{c}) = \mathcal{M}_b(y_{1:2} \mid \mathbf{c}).$$

Moreover, if $L > 1$, the subsequent rejection stage removes from the set of viable candidates any draft not satisfying $X_2^{(k)} = Y_2$, and so $\mathcal{S}_2 = \{k \mid X_1^{(k)} = Y_1, X_2^{(k)} = Y_2\} = \{k \mid X_{1:2}^{(k)} = Y_{1:2}\}$.

In general, assume that $\Pr[Y_{1:j} = y_{1:j}] = \mathcal{M}_b(y_{1:j} \mid \mathbf{c})$ and $X_{1:j}^{(k)} = Y_{1:j}$ for all $k \in \mathcal{S}_j$, for some $1 \leq j < \tau$. Implicitly, $j < L+1$, otherwise the generation would have already been completed. The next token is selected according to

$$Y_{j+1} = \arg\min_{1 \leq i \leq N} \min_{k \in \mathcal{S}_j} \frac{S_i^{(j+1,k)}}{q_i^{(j+1,k)}} = \arg\min_{1 \leq i \leq N} \min_{k \in \mathcal{S}_j} \frac{S_i^{(j+1,k)}}{\mathcal{M}_b(i \mid X_{1:j}^{(k)}, \mathbf{c})} = \arg\min_{1 \leq i \leq N} \min_{k \in \mathcal{S}_j} \frac{S_i^{(j+1,k)}}{\mathcal{M}_b(i \mid Y_{1:j}, \mathbf{c})}$$

and hence, $\Pr[Y_{j+1} = y_{j+1} \mid Y_{1:j} = y_{1:j}] = \mathcal{M}_b(y_{j+1} \mid y_{1:j}, \mathbf{c})$. Also,

$$\Pr[Y_{1:(j+1)} = y_{1:(j+1)}] = \mathcal{M}_b(y_{j+1} \mid y_{1:j}, \mathbf{c}) \mathcal{M}_b(y_{1:j} \mid \mathbf{c}) = \mathcal{M}_b(y_{1:(j+1)} \mid \mathbf{c}).$$

If $j < L$, the rejection step enures that

$$\mathcal{S}_{j+1} = \mathcal{S}_j \cap \{k \mid X_{j+1}^{(k)} = Y_{j+1}\} = \{k \mid X_{1:j}^{(k)} = Y_{1:j}\} \cap \{k \mid X_{j+1}^{(k)} = Y_{j+1}\} = \{k \mid X_{1:j}^{(k)} = Y_{1:j}\}.$$

On the other had, if $j = L$, the entire sequence up to $\tau$ is now complete because $\tau \leq L + 1$, and there are no more sampling or rejection steps. This concludes the inductive step which, along with the case for $j = 1$, makes up the proof for any $1 \leq j \leq \tau$.

We next prove our claim of conditional drafter invariance. The randomness is encapsulated by $\mathcal{R} = \{\{S_i^{(j,k)}\}_{i=1}^N\}_{k=1}^K$. Above, as an intermediate step to proving sequence-level correctness, we showed by induction that for any $0 \leq j < \tau$, any $k$ in $\mathcal{S}_j$ satisfies $X_{1:j}^{(k)} = Y_{1:j}$. Then, looking at the selection of $Y_{j+1}$, we have

$$Y_{j+1} = \arg\min_{1 \leq i \leq N} \min_{k \in \mathcal{S}_j} \frac{S_i^{(j+1,k)}}{q_i^{(j+1,k)}} = \arg\min_{1 \leq i \leq N} \min_{k \in \mathcal{S}_j} \frac{S_i^{(j+1,k)}}{\mathcal{M}_b(i \mid X_{1:j}^{(k)}, \boldsymbol{c})} = \arg\min_{1 \leq i \leq N} \min_{k \in \mathcal{S}_j} \frac{S_i^{(j+1,k)}}{\mathcal{M}_b(i \mid Y_{1:j}, \boldsymbol{c})}$$

as seen previously. From this, given $\mathcal{R}$, $\boldsymbol{c}$ and $Y_{1:j}$, $Y_{j+1}$ only depends on the draft sequences through $\mathcal{S}_j$. Starting from $Y_1$, we see that since $\mathcal{S}_0 = \{1, \ldots, K\}$,

$$Y_1 = \arg\min_{1 \leq i \leq N} \min_{1 \leq k \leq K} \frac{S_i^{(1,k)}}{\mathcal{M}_b(i \mid \boldsymbol{c})}.$$

As the draft tokens do not play any role in this expression, we get

$$\Pr[Y_1 = y_1 \mid \mathcal{R}, \boldsymbol{c}, \{X_{1:L}^{(k)}\}_{k=1}^K = \{x_{1:L}^{(k)}\}_{k=1}^K] = \Pr[Y_1 = y_1 \mid \mathcal{R}, \boldsymbol{c}]$$

which proves drafter invariance for $Y_1$. Note that we have written $\{X_{1:L}^{(k)}\}_{k=1}^K$ instead of $X_{1:L}^{(1)}, \ldots, X_{1:L}^{(K)}$ to simplify the notation somewhat. Now $Y_2$ is chosen according to

$$Y_2 = \arg\min_{1 \leq i \leq N} \min_{k \in \mathcal{S}_1} \frac{S_i^{(2,k)}}{q_i^{(2,k)}} = \arg\min_{1 \leq i \leq N} \min_{k \in \mathcal{S}_1} \frac{S_i^{(2,k)}}{\mathcal{M}_b(i \mid X_1^{(k)}, \boldsymbol{c})}.$$

Explicitly, $\mathcal{S}_1 = \{k \mid X_1^{(k)} = Y_1\}$. Hence, the choice of $Y_2$ depends only on the values of the draft tokens $\{X_{1:L}^{(k)}\}_{k=1}^K$ and not on the language models used to generate them. We can then write

$$\Pr[Y_2 = y_2 \mid Y_1 = y_1, \mathcal{R}, \boldsymbol{c}, \{X_{1:L}(\mathcal{M}_s^{(k)})\}_{k=1}^K = \{x_{1:L}^{(k)}\}_{k=1}^K]$$
$$= \Pr[Y_2 = y_2 \mid Y_1 = y_1, \mathcal{R}, \boldsymbol{c}, \{X_{1:L}(\widetilde{\mathcal{M}}_s^{(k)})\}_{k=1}^K = \{x_{1:L}^{(k)}\}_{k=1}^K]$$

for any choice of $\mathcal{M}_s^{(1)}, \ldots, \mathcal{M}_s^{(K)}$ and $\widetilde{\mathcal{M}}_s^{(1)}, \ldots, \widetilde{\mathcal{M}}_s^{(K)}$. Also,

$$\Pr[Y_{1:2} = y_{1:2} \mid \mathcal{R}, \boldsymbol{c}, \{X_{1:L}(\mathcal{M}_s^{(k)})\}_{k=1}^K = \{x_{1:L}^{(k)}\}_{k=1}^K]$$
$$= \Pr[Y_2 = y_2 \mid Y_1 = y_1, \mathcal{R}, \boldsymbol{c}, \{X_{1:L}(\mathcal{M}_s^{(k)})\}_{k=1}^K = \{x_{1:L}^{(k)}\}_{k=1}^K]$$
$$\times \Pr[Y_1 = y_1 \mid \mathcal{R}, \boldsymbol{c}, \{X_{1:L}(\mathcal{M}_s^{(k)})\}_{k=1}^K = \{x_{1:L}^{(k)}\}_{k=1}^K]$$
$$= \Pr[Y_2 = y_2 \mid Y_1 = y_1, \mathcal{R}, \boldsymbol{c}, \{X_{1:L}(\widetilde{\mathcal{M}}_s^{(k)})\}_{k=1}^K = \{x_{1:L}^{(k)}\}_{k=1}^K]$$
$$\times \Pr[Y_1 = y_1 \mid \mathcal{R}, \boldsymbol{c}, \{X_{1:L}(\widetilde{\mathcal{M}}_s^{(k)})\}_{k=1}^K = \{x_{1:L}^{(k)}\}_{k=1}^K]$$
$$= \Pr[Y_{1:2} = y_{1:2} \mid \mathcal{R}, \boldsymbol{c}, \{X_{1:L}(\widetilde{\mathcal{M}}_s^{(k)})\}_{k=1}^K = \{x_{1:L}^{(k)}\}_{k=1}^K]$$

Therefore, conditional drafter invariance is satisfied for $Y_{1:2}$. To extend the general case and thus complete the proof, we assume that $Y_{1:j}$ satisfies conditional drafter invariance, where $1 \leq j < \tau$. That is,

$$\Pr[Y_{1:j} = y_{1:j} \mid \mathcal{R}, \boldsymbol{c}, \{X_{1:L}(\mathcal{M}_s^{(k)})\}_{k=1}^K = \{x_{1:L}^{(k)}\}_{k=1}^K]$$
$$= \Pr[Y_{1:j} = y_{1:j} \mid \mathcal{R}, \boldsymbol{c}, \{X_{1:L}(\widetilde{\mathcal{M}}_s^{(k)})\}_{k=1}^K = \{x_{1:L}^{(k)}\}_{k=1}^K]$$

for any choice of $\mathcal{M}(s)^{(1)}, \ldots, \mathcal{M}_s^{(K)}$ and $\widetilde{\mathcal{M}}(s)^{(1)}, \ldots, \widetilde{\mathcal{M}}_s^{(K)}$. Since

$$Y_{j+1} = \arg\min_{1 \leq i \leq N} \min_{k \in \mathcal{S}_j} \frac{S_i^{(j+1,k)}}{\mathcal{M}_b(i \mid Y_{1:j}, \boldsymbol{c})}$$

and $\mathcal{S}_j = \{k \mid X_{1:j}^{(k)} = Y_{1:j}\}$, we again see that $Y_{j+1}$ only depends on the values of the draft tokens $\{X_{1:L}^{(k)}\}_{k=1}^K$ and not on the underlying probability model. Therefore,

$$\Pr[Y_{j+1} = y_{j+1} \mid Y_{1:j} = y_{1:j}, \ \mathcal{R}, \ \boldsymbol{c}, \ \{X_{1:L}(\mathcal{M}_s^{(k)})\}_{k=1}^K = \{x_{1:L}^{(k)}\}_{k=1}^K]$$
$$= \Pr[Y_{j+1} = y_{j+1} \mid Y_{1:j} = y_{1:j}, \ \mathcal{R}, \ \boldsymbol{c}, \ \{X_{1:L}(\widetilde{\mathcal{M}}_s^{(k)})\}_{k=1}^K = \{x_{1:L}^{(k)}\}_{k=1}^K]$$

By the induction hypothesis,

$$\Pr[Y_{1:(j+1)} = y_{1:(j+1)} \mid \mathcal{R}, \ \boldsymbol{c}, \ \{X_{1:L}(\mathcal{M}_s^{(k)})\}_{k=1}^K = \{x_{1:L}^{(k)}\}_{k=1}^K]$$
$$= \Pr[Y_{j+1} = y_{j+1} \mid Y_{1:j} = y_{1:j}, \ \mathcal{R}, \ \boldsymbol{c}, \ \{X_{1:L}(\mathcal{M}_s^{(k)})\}_{k=1}^K = \{x_{1:L}^{(k)}\}_{k=1}^K]$$
$$\times \Pr[Y_{1:j} = y_{1:j} \mid \mathcal{R}, \ \boldsymbol{c}, \ \{X_{1:L}(\mathcal{M}_s^{(k)})\}_{k=1}^K = \{x_{1:L}^{(k)}\}_{k=1}^K]$$
$$= \Pr[Y_{j+1} = y_{j+1} \mid Y_{1:j} = y_{1:j}, \ \mathcal{R}, \ \boldsymbol{c}, \ \{X_{1:L}(\widetilde{\mathcal{M}}_s^{(k)})\}_{k=1}^K = \{x_{1:L}^{(k)}\}_{k=1}^K]$$
$$\times \Pr[Y_{1:j} = y_{1:j} \mid \mathcal{R}, \ \boldsymbol{c}, \ \{X_{1:L}(\widetilde{\mathcal{M}}_s^{(k)})\}_{k=1}^K = \{x_{1:L}^{(k)}\}_{k=1}^K]$$
$$= Pr[Y_{1:(j+1)} = y_{1:(j+1)} \mid \mathcal{R}, \ \boldsymbol{c}, \ \{X_{1:L}(\widetilde{\mathcal{M}}_s^{(k)})\}_{k=1}^K = \{x_{1:L}^{(k)}\}_{k=1}^K]$$

Since we have already shown that conditional drafter invariance holds for $Y_1$, it then holds for all sequences $Y_{1:j}$, where $1 \le j \le \tau$. □

### A.5 Proof of theorem 2

**Theorem 2** (Conditional LML). *Using the strategy above, the error probability satisfies*

$$\Pr[Y \in \{X^{(1)}, \dots, X^{(K)}\} \mid Y = j, \ A = a, \ Z_1^K = z_1^K] \ge \sum_{k=1}^K \left( K + \frac{q_j(a)}{p_j(z_k)} \right)^{-1}.$$

*Proof.* We begin by following a similar approach to the proof of theorem 1, but we condition on $A$ and the $Z_k$'s where necessary. Following the setup introduced in appendix A.2, we define

$$\{T_i\}_{i=1}^{KN} = \{S_1^{(1)}, \dots, S_N^{(1)}, S_1^{(2)}, \dots, S_N^{(2)}, \dots, S_1^{(K)}, \dots, S_N^{(K)}\}$$

and this time use a conditional augmented target distribution $\tilde{q}_{\tilde{Y}|A}$ on $\tilde{\Omega} = \{1, \dots, KN\}$ with output $\tilde{Y}$. Given $A = a$, this distribution is defined by the probabilities

$$(\tilde{q}_1(a), \dots, \tilde{q}_{KN}(a)) = \left( \frac{q_1(a)}{K}, \dots, \frac{q_N(a)}{K}, \dots, \frac{q_1(a)}{K}, \dots, \frac{q_N(a)}{K} \right).$$

With this setup, we find

$$\Pr[Y = j, \ j \in \{X^{(1)}, \dots, X^{(K)}\} \mid A = a, \ Z_1^K = z_1^K]$$
$$= \sum_{k=1}^K \Pr[\tilde{Y} = j + (k-1)N, \ j \in \{X^{(1)}, \dots, X^{(K)}\} \mid A = a, \ Z_1^K = z_1^K]$$
$$\ge \sum_{k=1}^K \Pr[\tilde{Y} = j + (k-1)N, \ X^{(k)} = j \mid A = a, \ Z_1^K = z_1^K]. \tag{10}$$

As before, the analysis proceeds in the same manner regardless of the values of $j$ and $k$. For simplicity, we therefore consider the probability

$$\Pr[\tilde{Y} = 1, \ X^{(1)} = 1 \mid A = a, \ Z_1^K = z_1^K]$$
$$= \Pr[X^{(1)} = 1 \mid \tilde{Y} = 1, \ A = a, \ Z_1^K = z_1^K] \Pr[\tilde{Y} = 1 \mid A = a, \ Z_1^K = z_1^K]. \tag{11}$$

We start by computing

$$\Pr[X^{(1)} = 1 \mid \tilde{Y} = 1, \ A = a, \ Z_1^K = z_1^K]$$
$$= \Pr \left[ \frac{T_1}{p_1(z_1)} \le \min_{2 \le i \le N} \frac{T_i}{p_i(z_1)} \ \middle| \ \tilde{Y} = 1, \ A = a, \ Z_1^K = z_1^K \right]$$
$$= \Pr \left[ \frac{T_1}{p_1(z_1)} \le \min_{2 \le i \le N} \frac{T_i}{p_i(z_1)} \ \middle| \ \tilde{Y} = 1, \ A = a \right]. \tag{12}$$

To get the second equality above, we note that by construction $\{T_i\}_{i=1}^{KN} \to (Y, A) \to Z_1^K$ forms a Markov chain, as noted in section 5.2. Since $Y$ is a deterministic function of $\tilde{Y}$, $\{T_i\}_{i=1}^{KN} \to (\tilde{Y}, A) \to Z_1^K$ is also a Markov chain and hence the $T_i$'s are conditionally independent of $Z_1^K$ given $\tilde{Y}$ and $A$. We can now leverage a similar analysis to that in appendix A.2 to compute the required probability by first noting that

$$\Pr\left[\frac{T_1}{p_1(z_1)} \le \min_{2 \le i \le N} \frac{T_i}{p_i(z_1)} \,\middle|\, \tilde{Y} = 1,\ A = a\right]$$

$$= \Pr\left[\frac{T_1}{p_1(z_1)} \le \min_{2 \le i \le N} \frac{T_i}{p_i(z_1)},\ \tilde{Y} = 1 \,\middle|\, A = a\right] \Big/ \Pr[\tilde{Y} = 1 \mid A = a]. \quad (13)$$

Now, conditioned on $A = a$,

$$\frac{T_1}{p_1(z_1)} \le \min_{2 \le i \le N} \frac{T_i}{p_i(z_1)} \text{ and } \tilde{Y} = 1$$

$$\implies \frac{T_1}{p_1(z_1)} \le \min_{2 \le i \le N} \frac{T_i}{p_i(z_1)} \text{ and } \frac{T_1}{\tilde{q}_1(a)} \le \min_{2 \le i \le KN} \frac{T_i}{\tilde{q}_i(a)}$$

$$\implies T_1 \le \min\left\{\frac{T_2}{\max\{\tilde{q}_2(a)/\tilde{q}_1(a), p_2(z_1)/p_1(z_1)\}}, \ldots, \right.$$

$$\left.\frac{T_N}{\max\{\tilde{q}_N(a)/\tilde{q}_1(a), p_N(z_1)/p_1(z_1)\}}, \frac{T_{N+1}}{\tilde{q}_{N+1}(a)/\tilde{q}_1(a)}, \ldots, \frac{T_{KN}}{\tilde{q}_{KN}(a)/\tilde{q}_1(a)}\right\}.$$

We now note that the $T_i$'s are generated independently of the source $A$, and therefore remain i.i.d. $\mathrm{Exp}(1)$ random variables after conditioning on $A = a$. We can then follow our earlier approach leveraging the properties of independent exponential random variables to see that the right-hand side is an exponential random variable and is independent of the left-hand side. Let its parameter be $\lambda$. Then, using the definition of the $\tilde{q}_i(a)$'s to simplify the result, we have

$$1 + \lambda = \sum_{i=1}^{N} \max\left\{\frac{q_i(a)}{q_1(a)}, \frac{p_i(z_1)}{p_1(z_1)}\right\} + \sum_{k=2}^{K}\sum_{i=1}^{N} \frac{q_i(a)}{q_1(a)}$$

$$= \sum_{i=1}^{N}\left[\max\left\{\frac{q_i(a)}{q_1(a)}, \frac{p_i(z_1)}{p_1(z_1)}\right\} + (K-1)\frac{q_i(a)}{q_1(a)}\right].$$

As before, we then get

$$\Pr\left[\frac{T_1}{p_1(z_1)} \le \min_{2 \le i \le N} \frac{T_i}{p_i(z_1)},\ \tilde{Y} = 1 \,\middle|\, A = a\right]$$

$$= \frac{1}{\sum_{i=1}^{N}[\max\{q_i(a)/q_1(a), p_i(z_1)/p_1(z_1)\} + (K-1)q_i(a)/q_1(a)]}. \quad (14)$$

Putting together (11)–(14) gives

$$\Pr[\tilde{Y} = 1,\ X^{(1)} = 1 \mid A = a,\ Z_1^K = z_1^K]$$

$$= \frac{\Pr[\tilde{Y} = 1 \mid A = a,\ Z_1^K = z_1^K]/\Pr[\tilde{Y} = 1 \mid A = a]}{\sum_{i=1}^{N}[\max\{q_i(a)/q_1(a), p_i(z_1)/p_1(z_1)\} + (K-1)q_i(a)/q_1(a)]}$$

$$= \frac{q_1(a)\Pr[\tilde{Y} = 1 \mid A = a,\ Z_1^K = z_1^K]/\Pr[\tilde{Y} = 1 \mid A = a]}{K + q_1(a)\sum_{i=1}^{N}\max\{0, p_i(z_1)/p_1(z_1) - q_i(a)/q_1(a)\}}$$

$$\ge \frac{q_1(a)}{K + q_1(a)/p_1(z_1)}\frac{\Pr[\tilde{Y} = 1 \mid A = a,\ Z_1^K = z_1^K]}{\Pr[\tilde{Y} = 1 \mid A = a]}$$

$$= q_1(a)\left(K + \frac{q_1(a)}{p_1(z_1)}\right)^{-1}\frac{\Pr[\tilde{Y} = 1 \mid A = a,\ Z_1^K = z_1^K]}{\Pr[\tilde{Y} = 1 \mid A = a]}.$$

We next note that, given $A = a$, the selection procedure for $Y$ is

$$Y = \arg\min_{1 \le i \le N} \min_{1 \le k \le K} \frac{S_i^{(k)}}{q_i(a)}$$

and the $S_i^{(k)}$'s are i.i.d. Exp(1) random variables independent of $A$. As a result, the distribution guarantee of proposition 1 still holds for $Y$ such that $Y$ is sampled exactly from $q_{Y|A}$ and $\Pr[Y = 1 \mid A = a] = q_1(a)$. Then,

$$\Pr[\tilde{Y} = 1 \mid A = a] = q_1(a)/K \tag{15}$$

by symmetry and so

$$\Pr[\tilde{Y} = 1, \ X^{(1)} = 1 \mid A = a, \ Z_1^K = z_1^K] = \left(1 + \frac{q_1(a)}{Kp_1(z_1)}\right)^{-1} \Pr[\tilde{Y} = 1 \mid A = a, \ Z_1^K = z_1^K]$$

or, generalizing to any $j$ and $k$,

$$\Pr[\tilde{Y} = j + (k-1)N, \ X^{(k)} = j \mid A = a, \ Z_1^K = z_1^K]$$
$$= \left(1 + \frac{q_j(a)}{Kp_j(z_k)}\right)^{-1} \Pr[\tilde{Y} = j + (k-1)N \mid A = a, \ Z_1^K = z_1^K].$$

Going back to (10), we see that

$$\Pr[Y = j, \ j \in \{X^{(1)}, \ldots, X^{(K)}\} \mid A = a, \ Z_1^K = z_1^K]$$
$$\geq \sum_{k=1}^{K} \left(1 + \frac{q_j(a)}{Kp_j(z_k)}\right)^{-1} \Pr[\tilde{Y} = j + (k-1)N \mid A = a, \ Z_1^K = z_1^K].$$

Then, going from the joint to the conditional probability,

$$\Pr[Y \in \{X^{(1)}, \ldots, X^{(K)}\} \mid Y = j, \ A = a, \ Z_1^K = z_1^K]$$
$$= \frac{\Pr[Y = j, \ j \in \{X^{(1)}, \ldots, X^{(K)}\} \mid A = a, \ Z_1^K = z_1^K]}{\Pr[Y = j \mid A = a, \ Z_1^K = z_1^K]}$$
$$\geq \sum_{k=1}^{K} \left(1 + \frac{q_j(a)}{Kp_j(z_k)}\right)^{-1} \frac{\Pr[\tilde{Y} = j + (k-1)N \mid A = a, \ Z_1^K = z_1^K]}{\Pr[Y = j \mid A = a, \ Z_1^K = z_1^K]}. \tag{16}$$

We now claim that

$$\Pr[\tilde{Y} = j + (k-1)N \mid A = a, \ Z_1^K = z_1^K] = \Pr[Y = j \mid A = a, \ Z_1^K = z_1^K]/K. \tag{17}$$

Without loss of generality, assume $j = 1$ and $k = 1$. Applying Bayes' rule, we see that

$$\Pr[\tilde{Y} = 1 \mid A = a, \ Z_1^K = z_1^K] = \frac{\Pr[Z_1^K = z_1^K \mid \tilde{Y} = 1, \ A = a] \Pr[\tilde{Y} = 1 \mid A = a]}{\Pr[Z_1^K = z_1^K \mid A = a]}.$$

As seen earlier in (15), $\Pr[\tilde{Y} = 1 \mid A = a] = \Pr[Y = 1 \mid A = a]/K$ by symmetry. Furthermore, since $Y$ is a deterministic function of $\tilde{Y}$ and $\tilde{Y} \to (Y, A) \to Z_1^K$ forms a Markov chain,

$$\Pr[Z_1^K = z_1^K \mid \tilde{Y} = 1, \ A = a] = \Pr[Z_1^K = z_1^K \mid Y = 1, \ \tilde{Y} = 1, \ A = a]$$
$$= \Pr[Z_1^K = z_1^K \mid Y = 1, \ A = a]$$

and so

$$\Pr[\tilde{Y} = 1 \mid A = a, \ Z_1^K = z_1^K] = \frac{\Pr[Z_1^K = z_1^K \mid Y = 1, \ A = a] \Pr[Y = 1 \mid A = a]/K}{\Pr[Z_1^K = z_1^K \mid A = a]}$$
$$= \Pr[Y = 1 \mid A = a, \ Z_1^K = z_1^K]/K.$$

Since the same argument works for arbitrary $j$ and $k$, we have shown (17). Finally, substituting back into (16) gives

$$\Pr[Y \in \{X^{(1)}, \ldots, X^{(K)}\} \mid Y = j, \ A = a, \ Z_1^K = z_1^K] \geq \sum_{k=1}^{K} \left(K + \frac{q_j(a)}{p_j(z_k)}\right)^{-1}$$

$\square$

## A.6 Proof of proposition 4

**Proposition 4.** *For the coding scheme in section 5.1, the error probability is bounded above as*

$$\Pr[Y \notin \{X^{(1)}, \dots, X^{(K)}\}] \leq 1 - \mathrm{E}_{A,W,T}\left[\left(1 + \frac{2^{i(W;A|T)}}{KL_{\max}}\right)^{-1}\right] \tag{5}$$

*where* $i_{W,A|T}(w; a \mid t) = \log(p_{W|A}(w \mid a)/p_{W|T}(w \mid t))$ *is the conditional information density.*

*Proof.* To obtain the desired result, it is first necessary to identify the target distributions used at the encoder and decoder in the coding scheme from section 5.1, then match these to $q_{Y|A}$ and $p_{X|Z}$ in theorem 2. Doing this, we have $q_{Y|A}(i \mid a) = p_{B|A}(B_i \mid a)$ and $p_{X|Z}(i \mid t, \ell) = p_{B|Z}(B_i \mid t, \ell)$. Recall that we defined $Z_k = (T_k, \ell_j)$ to encapsulate the side information and message available to decoder $k$ when the selected index is $Y = j$. We also defined $B_i = (i, \ell_i)$. Given $T_k = t_k$, each $X^{(k)}$ is then sampled using the target distribution $p_{X|Z}(\cdot \mid t_k, \ell_j)$, and our sampling process generates $X^{(1)}, \dots, X^{(K)}$. Applying theorem 2, we get

$$\Pr[Y \in \{X^{(1)}, \dots, X^{(K)}\} \mid Y = j, \ A = a, \ Z_1^K = \{(t_k, \ell_j)\}_{k=1}^K]$$

$$\geq \sum_{k=1}^K \left(K + \frac{q_{Y|A}(j \mid a)}{p_{X|Z}(j \mid t_k, \ell_j)}\right)^{-1}$$

$$= \sum_{k=1}^K \left(K + \frac{p_{B|A}(j, \ell_j \mid a)}{p_{B|Z}(j, \ell_j \mid t_k, \ell_j)}\right)^{-1}$$

$$= \sum_{k=1}^K \left(K + \frac{p_{W|A}(j \mid a)/L_{\max}}{p_{W|T}(j \mid t_k)}\right)^{-1}$$

$$= \sum_{k=1}^K \left(K + 2^{i_{W,A|T}(j;a|t_k)}/L_{\max}\right)^{-1}$$

where $i_{W,A|T}(j; a \mid t_k) = \log(p_{W|A}(j \mid a)/p_{W|T}(j \mid t_k))$ is the conditional information density. After removing the conditioning, we get

$$\Pr[Y \in \{X^{(1)}, \dots, X^{(K)}\}] \geq \mathrm{E}_{A,W,T}\left[\sum_{k=1}^K (K + 2^{i(W;A|T)}/L_{\max})^{-1}\right]$$

$$= K\,\mathrm{E}_{A,W,T}\left[(K + 2^{i(W;A|T)}/L_{\max})^{-1}\right]$$

$$= \mathrm{E}_{A,W,T}\left[\left(1 + \frac{2^{i(W;A|T)}}{KL_{\max}}\right)^{-1}\right].$$

By taking the complement we finally get a bound on the error probability,

$$\Pr[Y \notin \{X^{(1)}, \dots, X^{(K)}\}] \leq 1 - \mathrm{E}_{A,W,T}\left[\left(1 + \frac{2^{i(W;A|T)}}{KL_{\max}}\right)^{-1}\right].$$

$\square$

## A.7 Proof of remark 1

**Degenerate proposal distribution.** First consider the case where the distribution $p_X$ has all its mass on one element. Without loss of generality, let us assume $p_1 = 1$ and $p_j = 0$ for all $j \neq 1$. On the left-hand side of (3), since $X^{(1)} = X^{(2)} = \cdots = X^{(K)} = 1$ for every outcome in the sample space, we get

$$\Pr[Y \in \{X^{(1)}, \dots, X^{(K)}\}] = \Pr[Y = 1].$$

Now evaluate the right-hand side of (3). Since $q_i/q_1 \geq p_i/p_1$ for all $1 \leq i \leq N$, the $j = 1$ term in the outer sum evaluates to

$$\frac{K}{\sum_{i=1}^N [\max\{q_i/q_1, p_i/p_1\} + (K-1)q_i/q_1]} = \frac{K}{\sum_{i=1}^N [q_i/q_1 + (K-1)q_i/q_1]} = \frac{K}{K/q_1} = q_1.$$

We will evaluate the remaining terms more carefully by taking limits. For $j \neq 1$,

$$\lim_{p_2,\ldots,p_N \to 0} \frac{K}{\sum_{i=1}^N [\max\{q_i/q_j, p_i/p_j\} + (K-1)q_i/q_j]}$$

$$= \lim_{p_2,\ldots,p_N \to 0} \frac{Kp_j}{\sum_{i=1}^N [\max\{q_i p_j/q_j, p_i\} + (K-1)q_i p_j/q_j]}$$

$$= \frac{\lim_{p_2,\ldots,p_N \to 0}[Kp_j]}{\lim_{p_2,\ldots,p_N \to 0} \sum_{i=1}^N [\max\{q_i p_j/q_j, p_i\} + (K-1)q_i p_j/q_j]}.$$

We can now directly evaluate the numerator and the denominator separately. The numerator becomes zero, and as for the denominator we get

$$\lim_{p_2,\ldots,p_N \to 0} \sum_{i=1}^N \left[ \max\left\{ \frac{q_i p_j}{q_j}, p_i \right\} + (K-1)\frac{q_i p_j}{q_j} \right]$$

$$= (K-1)\frac{p_j}{q_j} + \max\left\{ \frac{q_1 p_j}{q_j}, p_1 \right\} + \sum_{i=2}^N \max\left\{ \frac{q_i p_j}{q_j}, p_i \right\}\bigg|_{p_2,\ldots,p_N=0}$$

$$= p_1 = 1.$$

Therefore, $j \neq 1$ terms are all zero. The right-hand side of (3) becomes $q_1 = \Pr[Y = 1]$, verifying that the bound is exact for this class of distributions.

**The case of $K = 1$.** Next, we will work out the matching probability for $K = 1$ explicitly as a special case to confirm that our bound recovers the exact expression. When $K = 1$, the matching probability is

$$\Pr[Y = X^{(1)}] = \sum_{j=1}^N \Pr[Y = j, \, X^{(1)} = j].$$

For convenience, we will drop the superscript on $X^{(1)}$ and also on the $S_i^{(1)}$'s during the analysis. With $K = 1$, the GLS procedure can therefore be written more simply as

1. Select $Y = \arg\min_{1 \leq i \leq N} S_i/q_i$ to generate a sample from $q_Y$.
2. Select $X = \arg\min_{1 \leq i \leq N} S_i/p_i$ to generate a sample from $p_X$.

Without loss of generality assume $j = 1$. We can then focus on finding the probability of the event

$$Y = 1 \text{ and } X = 1$$

$$\implies \frac{S_1}{q_1} \leq \min_{2 \leq i \leq N} \frac{S_i}{q_i} \text{ and } \frac{S_1}{p_1} \leq \min_{2 \leq i \leq N} \frac{S_i}{p_i}$$

$$\implies S_1 \leq \min_{2 \leq i \leq N} \frac{S_i}{\max\{q_i/q_1, p_i/p_1\}}.$$

The RHS is an exponential random variable independent of the LHS. If its parameter is $\lambda$, then we get

$$1 + \lambda = \sum_{i=1}^N \max\left\{ \frac{q_i}{q_1}, \frac{p_i}{p_1} \right\}.$$

Therefore, by the properties of independent exponential random variables,

$$\Pr[Y = 1, \, X = 1] = \frac{1}{\sum_{i=1}^N \max\{q_i/q_1, p_i/p_1\}}.$$

Therefore after generalizing to arbitrary $j$ we get

$$\Pr[Y = X] = \sum_{j=1}^N \frac{1}{\sum_{i=1}^N \max\{q_i/q_j, p_i/p_j\}}.$$

This is exactly the RHS of (3) evaluated for $K = 1$.

**Identical draft and proposal distributions.** Next, we examine the case when $p_X$ and $q_Y$ are identical, i.e. $p_i = q_i$ for all $1 \leq i \leq N$. From our proof of proposition 1 in appendix A.1 recall that

$$Y = j \implies \frac{S_j^*}{q_j} \leq \frac{S_i^*}{q_i} \; \forall \, i \neq j$$

where $S_j^* = \min_{1 \leq k \leq K} S_j^{(k)}$. Therefore, choosing $Y = j$ implies that there exists some $k$ such that

$$\frac{S_j^{(k)}}{q_j} \leq \frac{S_i^{(l)}}{q_i} \; \forall \, (i,l) \neq (j,k) \implies \frac{S_j^{(k)}}{q_j} \leq \frac{S_i^{(k)}}{q_i} \; \forall \, i \neq j.$$

But, since $p_i = q_i$ for all $1 \leq i \leq N$,

$$\frac{S_j^{(k)}}{p_j} \leq \frac{S_i^{(k)}}{p_i} \; \forall \, i \neq j \implies X^{(k)} = j.$$

In summary, if $Y = j$ then there exists some $k$ such that $X^{(k)} = j$ also, and hence $\Pr[Y \in \{X^{(1)}, \ldots, X^{(K)}\}] = 1$. And under our assumption that $p_i = q_i$ for all $1 \leq i \leq N$, the right-hand side of (3) simplifies to

$$\sum_{j=1}^{N} \frac{K}{\sum_{i=1}^{N}[\max\{q_i/q_j, p_i/p_j\} + (K-1)q_i/q_j]} = \sum_{j=1}^{N} \frac{K}{\sum_{i=1}^{N}[q_i/q_j + (K-1)q_i/q_j]}$$

$$= \sum_{i=1}^{N} q_j = 1.$$

# B    Connection to the notion of drafter invariance from Daliri et al. [9]

In Daliri et al. [9], an intuitive notion of drafter invariance is proposed for single-draft speculative decoding with the following interpretation: given fixed random numbers, the output of the speculative decoding algorithm is a function of the context and target model weights only. Our notion as stated in definition 1 is somewhat weaker, since we allow the output to depend also on the draft sequences, yet we still require that a given set of draft sequences always produce the same conditional output distribution. To formalize the notion of Daliri et al. [9] and extend it to the multi-draft case, we propose the following more restrictive definition, which we call *strong drafter invariance*.

**Definition 2** (Strong drafter invariance)**.** *Let $\mathcal{R}$ be the source of randomness used to draw samples, e.g. the output of a random number generator. A multi-draft speculative decoding algorithm is strongly drafter invariant if, for all $1 \leq j \leq \tau$,*

$$\Pr[Y_{1:j} = y_{1:j} \mid \mathcal{R}, \, \boldsymbol{c}, \, X_{1:L}^{(1)} = x_{1:L}^{(1)}, \ldots, X_{1:L}^{(K)} = x_{1:L}^{(K)}] = \Pr[Y_{1:j} = y_{1:j} \mid \mathcal{R}, \, \boldsymbol{c}].$$

Unfortunately, satisfying definition 2 incurs a performance penalty in multi-draft implementations, which can be seen empirically from the extended results in appendix D.1. To see why, note that the choice of $Y_j$ at each step $j$ in algorithm 2 depends on the current set of active drafts $\mathcal{S}$, which itself is a function of the preceding draft tokens. To completely remove any dependence on the draft sequences as required by definition 2, we would need to keep $\mathcal{S}$ fixed throughout the procedure, wastefully coupling the target model's output to draft tokens that have already been rejected. Regardless, we now show how our method as described in algorithm 2 can be modified to support strong drafter invariance by way of the following proposition.

**Proposition 6.** *If the minimum is taken over all $k \in \{1, \ldots, K\}$ in lines 9 and 13 of algorithm 2 instead of over $k \in \mathcal{S}$, strong drafter invariance holds in the sense of definition 2.*

*Proof.* We extend the proof of conditional drafter invariance set out in appendix A.4. There, we showed that

$$\Pr[Y_1 = y_1 \mid \mathcal{R}, \, \boldsymbol{c}, \, \{X_{1:L}^{(k)}\}_{k=1}^{K} = \{x_{1:L}^{(k)}\}_{k=1}^{K}] = \Pr[Y_1 = y_1 \mid \mathcal{R}, \, \boldsymbol{c}]$$

which is enough to prove strong drafter invariance for $Y_1$. As a result, we only need to modify the inductive step of the proof. Take some $1 \leq j < \tau$. With the modification that the minimum is taken over all $k \in \{1, \ldots, K\}$ instead of over $k \in \mathcal{S}$ in lines 9 and 13 of algorithm 2, $Y_{j+1}$ is selected as

$$Y_{j+1} = \arg\min_{1 \leq i \leq N} \min_{1 \leq k \leq K} \frac{S_i^{(j+1,k)}}{\mathcal{M}_b(i \mid Y_{1:j}, \boldsymbol{c})}.$$

From this, given $\mathcal{R}$, $\boldsymbol{c}$ and $Y_{1:j}$, the draft tokens are not involved, either directly or indirectly, in the choice of $Y_{j+1}$. More precisely,

$$\Pr[Y_{j+1} = y_{j+1} \mid Y_{1:j} = y_{1:j}, \ \mathcal{R}, \ \boldsymbol{c}, \ \{X_{1:L}(\mathcal{M}_s^{(k)})\}_{k=1}^K = \{x_{1:L}^{(k)}\}_{k=1}^K]$$
$$= \Pr[Y_{j+1} = y_{j+1} \mid Y_{1:j} = y_{1:j}, \ \mathcal{R}, \ \boldsymbol{c}].$$

Now assume that $Y_{1:j}$ satisfies strong drafter invariance. As a result,

$$\Pr[Y_{1:(j+1)} = y_{1:(j+1)} \mid \mathcal{R}, \ \boldsymbol{c}, \ \{X_{1:L}(\mathcal{M}_s^{(k)})\}_{k=1}^K = \{x_{1:L}^{(k)}\}_{k=1}^K]$$
$$= \Pr[Y_{j+1} = y_{j+1} \mid Y_{1:j} = y_{1:j}, \ \mathcal{R}, \ \boldsymbol{c}, \ \{X_{1:L}(\mathcal{M}_s^{(k)})\}_{k=1}^K = \{x_{1:L}^{(k)}\}_{k=1}^K]$$
$$\times \Pr[Y_{1:j} = y_{1:j} \mid \mathcal{R}, \ \boldsymbol{c}, \ \{X_{1:L}(\mathcal{M}_s^{(k)})\}_{k=1}^K = \{x_{1:L}^{(k)}\}_{k=1}^K]$$
$$= \Pr[Y_{j+1} = y_{j+1} \mid Y_{1:j} = y_{1:j}, \ \mathcal{R}, \ \boldsymbol{c}] \Pr[Y_{1:j} = y_{1:j} \mid \mathcal{R}, \ \boldsymbol{c}]$$
$$= \Pr[Y_{1:(j+1)} = y_{1:(j+1)} \mid \mathcal{R}, \ \boldsymbol{c}]$$

Therefore, strong drafter invariance holds for all sequences $Y_{1:j}$, where $1 \leq j \leq \tau$. $\qquad\square$

We can also obtain a lower bound on the token-level acceptance probability of the strongly drafter-invariant scheme. Assume the drafts are i.i.d. from the same model $\mathcal{M}_s$, thus using the same setting as proposition 2. Reexamining the steps in the proof of theorem 1 found in appendix A.2, we see that if the number of active drafts at the current step is $J \leq K$, one of these $J$ candidates must match the target for a token to be accepted. On the other hand, rejection now occurs if the target matches any of the remaining $K - J$ drafts, or none at all. Specifically, assuming without loss of generality that the active drafts are $X^{(1)}, \ldots, X^{(J)}$, (7) becomes

$$\Pr[Y \in \{X^{(1)}, \ldots, X^{(J)}\}] \geq \sum_{j=1}^N J \Pr[\tilde{Y} = j, \ X^{(1)} = j].$$

However, the augmented target distribution described in (6) remains unchanged, because all $K$ drafts remain coupled through the common randomness. This includes the $K - J$ drafts that have already been rejected during previous steps. Following the same analysis as in appendix A.2 then shows that the probability of accepting at least one token at the current step with context $\boldsymbol{c}$ is bounded below as

$$\Pr[\text{accept}] \geq \sum_{j=1}^N \frac{J}{\sum_{i=1}^N [\max\{q_i/q_j, p_i/p_j\} + (K-1)q_i/q_j]}.$$

where $p_X := \mathcal{M}_s(\cdot \mid \boldsymbol{c})$ and $q_Y := \mathcal{M}_b(\cdot \mid \boldsymbol{c})$. Conversely, with $J$ active drafts proposition 2 gives

$$\Pr[\text{accept}] \geq \sum_{j=1}^N \frac{J}{\sum_{i=1}^N [\max\{q_i/q_j, p_i/p_j\} + (J-1)q_i/q_j]}.$$

for the original conditionally drafter-invariant algorithm. Since $J \leq K$, we can see that requiring strong drafter invariance reduces the lower bound, providing some theoretical insight into the poor performance observed in appendix D.1 when using this scheme for LLM inference.

## C Extension to continuous distributions via importance sampling

It is not possible to enumerate the entire sample space when dealing with continuous distributions. At first glance, this appears to preclude the use of GLS in such settings. However, we can obtain approximate samples through importance sampling as described in Phan et al. [36]. For clarity, we skip the general case and instead proceed directly to the source coding application set out in

section 5.1 of the main paper, removing the assumption that $W$ is discrete. To start, we choose some sufficiently large $N$ and generate $N$ i.i.d. samples of $W$ following $p_W$. Let these samples be $U_1^N \coloneqq \{U_i\}_{i=1}^N$. Recall that $p_W$ is the marginal target distribution for the decoder's output. Again, let $\ell_1, \ldots, \ell_N$ be uniform random integers selected from $\{1, \ldots, L_{\max}\}$. We then have the tuples $B_i = (U_i, \ell_i)$ forming part of the common randomness, the difference from section 5.1 being that the $U_i$'s are now also random, whereas in the discrete case they were fixed to enumerate the whole sample space.

At the encoder, we want to sample $Y$ approximately from $p_{B|A}$ given the observation $A = a$. Therefore, we introduce the *unnormalized* importance weights

$$\tilde{\lambda}_{q,i}(U_i) = \frac{p_{B|A}(B_i \mid a)}{p_B(B_i)} = \frac{p_{W|A}(U_i \mid a)}{p_W(U_i)}.$$

Similarly, the decoders should use the target distribution $p_{B|Z}$, where decoder $k$ has access to $Z_k = (T_k, \ell_j)$. Here, $j$ is the selected index at the encoder and $\ell_j$ is the associated random integer. Given $T_k = t_k$, we define

$$\tilde{\lambda}_{p,i}^{(k)}(U_i) = \frac{p_{B|Z}(B_i \mid t_k, \ell_j)}{p_B(B_i)} = \frac{p_{W|T}(U_i \mid t_k)\mathbb{1}\{\ell_i = \ell_j\}}{p_W(U_i)/L_{\max}}.$$

The final normalized importance weights are

$$\lambda_{q,i}(U_1^N) = \frac{\tilde{\lambda}_{q,i}(U_i)}{\sum_{j=1}^N \tilde{\lambda}_{q,j}(U_j)} \text{ and } \lambda_{p,i}^{(k)}(U_1^N) = \frac{\tilde{\lambda}_{p,i}^{(k)}(U_i)}{\sum_{j=1}^N \tilde{\lambda}_{p,j}^{(k)}(U_j)}.$$

The index selection at the encoder and decoder simply proceeds as

$$Y = \arg\min_{1 \le i \le N} \min_{1 \le k \le K} \frac{S_i^{(k)}}{\lambda_{q,i}(U_1^N)} \text{ and } X^{(k)} = \arg\min_{1 \le i \le N} \frac{S_i^{(k)}}{\lambda_{p,i}^{(k)}(U_1^N)}.$$

The output of decoder $k$ is then taken to be $W_k = U_{X^{(k)}}$. Conditional on $U_1^N$, the bound on the index matching probability stated in proposition 4 holds. However, we need a bound that is conditional only on $U_j$. To start, we write down the bound when conditioning on $U_1^N$, drawing from the development in appendix A.6.

$$\Pr[Y \in \{X^{(1)}, \ldots, X^{(K)}\} \mid Y = j, \ A = a, \ Z_1^K = z_1^K, \ U_1^N = u_1^N]$$
$$\ge \sum_{k=1}^K \left(K + \frac{\lambda_{q,j}(u_1^N)}{\lambda_{p,j}^{(k)}(u_1^N)}\right)^{-1}$$
$$= \sum_{k=1}^K \left(K + \frac{\tilde{\lambda}_{q,j}(u_j)}{\tilde{\lambda}_{p,j}^{(k)}(u_j)} \frac{\sum_{i=1}^N \tilde{\lambda}_{p,i}^{(k)}(u_1^N)}{\sum_{i=1}^N \tilde{\lambda}_{q,i}(u_1^N)}\right)^{-1}.$$

For simplicity, let us assume for now without loss of generality that $j = 1$. We want to remove the conditioning on $U_2, \ldots, U_N$, which can be done by taking the expectation to get

$$\Pr[Y \in \{X^{(1)}, \ldots, X^{(K)}\} \mid Y = 1, \ A = a, \ Z_1^K = z_1^K, \ U_1 = u_1]$$
$$\ge \mathrm{E}_{U_2^N}\left[\sum_{k=1}^K \left(K + \frac{\tilde{\lambda}_{q,1}(u_1)}{\tilde{\lambda}_{p,1}^{(k)}(u_1)} \frac{\sum_{i=1}^N \tilde{\lambda}_{p,i}^{(k)}(U_1^N)}{\sum_{i=1}^N \tilde{\lambda}_{q,i}(U_1^N)}\right)^{-1} \middle| Y = 1, \ A = a, \ Z_1^K = z_1^K, \ U_1 = u_1\right]$$
$$= \sum_{k=1}^K \mathrm{E}_{U_2^N}\left[\left(K + \frac{\tilde{\lambda}_{q,1}(u_1)}{\tilde{\lambda}_{p,1}^{(k)}(u_1)} \frac{\sum_{i=1}^N \tilde{\lambda}_{p,i}^{(k)}(U_1^N)}{\sum_{i=1}^N \tilde{\lambda}_{q,i}(U_1^N)}\right)^{-1} \middle| Y = 1, \ A = a, \ Z_1^K = z_1^K, \ U_1 = u_1\right]$$
$$\ge \sum_{k=1}^K \left(K + \frac{\tilde{\lambda}_{q,1}(u_1)}{\tilde{\lambda}_{p,1}^{(k)}(u_1)} \mathrm{E}_{U_2^N}\left[\frac{\sum_{i=1}^N \tilde{\lambda}_{p,i}^{(k)}(U_1^N)}{\sum_{i=1}^N \tilde{\lambda}_{q,i}(U_1^N)} \middle| Y = 1, \ A = a, \ Z_1^K = z_1^K, \ U_1 = u_1\right]\right)^{-1}$$

where the last step uses Jensen's inequality. To simplify the inner expectation, we use the following lemma, stated here without proof, which extracted from the proof of theorem 3 in Phan et al. [36].

**Lemma 1** ([36, p. 23]). *We have that*

$$\mathrm{E}_{U_2^N}\left[\left.\frac{\sum_{i=1}^N \tilde{\lambda}_{p,i}^{(k)}(U_1^N)}{\sum_{i=1}^N \tilde{\lambda}_{q,i}(U_1^N)}\ \right|\ Y=1,\ A=a,\ Z_1^K=z_1^K,\ U_1=u_1\right] \le \mu_k(N,u_1) \qquad (18)$$

*where*

$$\mu_k(N,u_1) = \frac{\tilde{\lambda}_{p,1}^{(k)}(u_1)+\bar{N}}{\tilde{\lambda}_{q,1}(u_1)+\bar{N}} + \frac{K(\bar{N})}{\bar{N}}\left(1+\frac{\tilde{\lambda}_{q,1}(u_1)}{\bar{N}}\right) + \frac{2\omega L(\bar{N})}{\bar{N}}\left(1+\frac{\tilde{\lambda}_{q,1}(u_1)}{\bar{N}}\right).$$

*In the equation above, we define $\bar{N}=N-1$ and*

$$K(\bar{N}) = \frac{4(\omega-1)}{(1+\tilde{\lambda}_{q,1}(u_1)/\bar{N})^2}\left(1+\frac{(N+1)\omega}{N}\right)$$

$$\times\sqrt{2+4\left(\frac{\tilde{\lambda}_{p,1}^{(k)}(u_1)+\bar{N}}{2\tilde{\lambda}_{q,1}(u_1)+\bar{N}}\right)^2\left[\left(1+\frac{(N+1)\omega}{\bar{N}}\right)^2+\frac{\omega-1}{\bar{N}}\right]}$$

$$L(\bar{N}) = \sqrt{\omega-1}\sqrt{d_5(p_W\,\|\,p_{W|A}(\cdot\mid a)) - d_3(p_W\,\|\,p_{W|A}(\cdot\mid a))^2}$$
$$+ (\omega-1)d_3(p_W\,\|\,p_{W|A}(\cdot\mid a))$$

*where, for all $m\ge 1$,*

$$d_{m+1}(p_W,p_{W|A}(\cdot\mid a)) = \mathrm{E}_{W\sim p_W}\left[\frac{p_W(W)^m}{p_{W|A}(W\mid a)^m}\right]$$

*and $\omega$ is chosen so that, for all $a$ and $w$, $p_{W|A}(w\mid a)/p_W(w)\le\omega$.*

Using lemma 1, or more specifically (18), and generalizing to arbitrary $j$, we see that

$$\Pr[Y\in\{X^{(1)},\dots,X^{(K)}\}\mid Y=j,\ A=a,\ Z_1^K=z_1^K,\ U_j=u_j]$$

$$\ge \sum_{k=1}^K\left(K+\mu_k(N,u_j)\frac{\tilde{\lambda}_{q,j}(u_j)}{\tilde{\lambda}_{p,j}^{(k)}(u_j)}\right)^{-1}$$

$$\ge \sum_{k=1}^K\left(K+\hat{\mu}(N,u_j)\frac{\tilde{\lambda}_{q,j}(u_j)}{\tilde{\lambda}_{p,j}^{(k)}(u_j)}\right)^{-1}$$

where $\hat{\mu}(N,u_j):=\max_{1\le k\le K}\mu_k(N,u_j)$. Using the definitions of $\tilde{\lambda}_{q,j}(u_j)$ and $\tilde{\lambda}_{p,j}^{(k)}(u_j)$,

$$\Pr[Y\in\{X^{(1)},\dots,X^{(K)}\}\mid Y=j,\ A=a,\ Z_1^K=z_1^K,\ U_j=u_j]$$

$$\ge \sum_{k=1}^K\left(K+\hat{\mu}(N,u_j)\frac{p_{W|A}(u_j\mid a)/L_{\max}}{p_{W|T}(u_j\mid t_k)}\right)^{-1}$$

$$= \sum_{k=1}^K\left(K+\hat{\mu}(N,u_j)2^{i_{W,A|T}(u_j;a|t_k)}/L_{\max}\right)^{-1}.$$

To proceed and find a counterpart to the coding theorem in proposition 4, we use the fact that for any $\varepsilon>0$, there exists some $M_k$ such that $\mu_k(N,u_j)\le 1+\varepsilon$ for all $N\ge M_k$ and $1\le j\le N$ [36, p. 24]. Hence, with $M=\max_{1\le k\le K}M_k$, we see that $\hat{\mu}(N,u_j)\le 1+\varepsilon$ for all $N\ge M$ and $1\le j\le N$. As a result, after removing the conditioning and following the steps used in the proof of proposition 4 in appendix A.6, we get

$$\Pr[Y\in\{X^{(1)},\dots,X^{(K)}\}] \ge \mathrm{E}_{A,W,T}\left[\left(1+(1+\varepsilon)\frac{2^{i(W;A|T)}}{KL_{\max}}\right)^{-1}\right]$$

and the associated error probability bound is

$$\Pr[Y\notin\{X^{(1)},\dots,X^{(K)}\}] \le 1-\mathrm{E}_{A,W,T}\left[\left(1+(1+\varepsilon)\frac{2^{i(W;A|T)}}{KL_{\max}}\right)^{-1}\right].$$

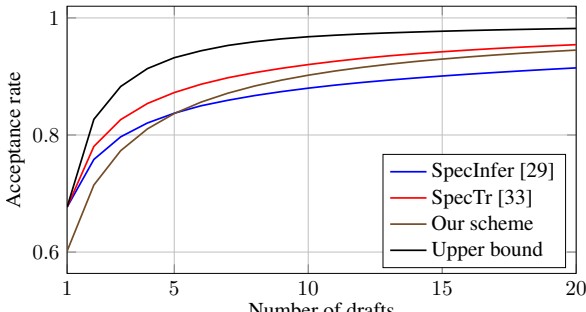

Figure 6: Proof of concept on random toy distributions.

# D    Additional experimental details

## D.1    Multi-draft speculative decoding

**Proof of concept on toy distributions.** As a simple demonstration of our method with arbitrary discrete distributions, we generate 100 random instances of $p_X$ and $q_Y$ each containing $N = 10$ elements, while the number of proposals is varied between 1 and 20. Results are shown in figure 6. As well as showing the token-level matching rate achieved by SpecTr [38], SpecInfer [34] and our algorithm, we also plot the optimal multi-draft acceptance rate *with* communication, which can be computed via a linear programming approach [38], at least for distributions on small alphabets. Note that while this calculation provides a useful upper bound, there is currently no multi-draft token selection scheme that can achieve the optimum in practice. Despite involving no communication between the drafter and the target, our algorithm is competitive with state-of-the-art methods, especially when the number of drafts is large.

**LLM inference.** Implementations of SpecInfer [34], SpecTr [38] and our drafter-invariant schemes can be found in the provided code. To obtain performance measurements, each speculative decoding configuration is tested on 200 prompts from the GSM8K [7], NaturalReasoning [48], MBPP [1] and DROP [10] datasets, and 164 prompts from HumanEval [6]. Our full set of results is given in tables 3 and 4; please note that not all datasets and configurations are reported in the main paper. We also include results for the strongly drafter-invariant scheme described in appendix B. The target model is Qwen 2.5-7 B [47] while the drafter is Qwen 2.5-0.5 B, and we use top-K sampling with $K = 50$. For our experiments with i.i.d. drafts in table 3, the temperature is 1.0 throughout and the maximum draft length is $L = 4$. When we use diverse drafts in table 4, the target temperature is 2.0, the two draft temperatures are varied and $L = 5$.

As described in the main text, the block efficiency is equal to the average number of tokens accepted during each iteration of the speculative decoding algorithm, while token rates are calculated from wall-clock measurements and reported as percentage speedups relative to single-draft speculative decoding with the same draft length. We compute the mean across all prompts for each dataset, then repeat the experiments 5 times with different random seeds. For a configuration-dataset pair, this gives us five measurements each for the average block efficiency and token rate. Let these be $BE_1, \ldots, BE_5$ and $TR_1, \ldots, TR_5$. As our final result, we report

$$BE = \text{mean}(BE_1, \ldots, BE_5), \; TR = \text{mean}(TR_1, \ldots, TR_5)$$

while the error bars show one standard error of the mean,

$$\sigma_{BE} = \text{std}(BE_1, \ldots, BE_5)/\sqrt{5}, \; \sigma_{TR} = \text{std}(TR_1, \ldots, TR_5)/\sqrt{5}$$

where the $\text{std}$ operator is the usual sample standard deviation formula. If $x$ is a vector of $M$ independent measurements in $\mathbb{R}$, this is

$$\text{std}(x) = \sqrt{\frac{\sum_{i=1}^{M}(x_i - \bar{x})^2}{M - 1}}, \; \bar{x} = \text{mean}(x).$$

On a NVIDIA RTX6000 Ada GPU with $48\,\text{GB}$ of memory, each configuration can be tested on a single dataset in around 1 hour.

| Strategy | $K$ | GSM8K | | HumanEval | | NaturalReasoning | | MBPP | | DROP | |
|---|---|---|---|---|---|---|---|---|---|---|---|
| | | BE | TR (%) | BE | TR (%) | BE | TR (%) | BE | TR (%) | BE | TR (%) |
| SpecInfer [34] | 2 | $4.47 \pm 0.01$ | $3.75 \pm 0.21$ | $4.11 \pm 0.02$ | $6.07 \pm 0.53$ | $3.75 \pm 0.01$ | $6.26 \pm 0.26$ | $4.04 \pm 0.01$ | $6.51 \pm 0.51$ | $3.33 \pm 0.01$ | $8.08 \pm 0.61$ |
| | 4 | $4.64 \pm 0.01$ | $5.20 \pm 0.23$ | $4.35 \pm 0.01$ | $8.69 \pm 0.27$ | $3.99 \pm 0.01$ | $10.63 \pm 0.32$ | $4.30 \pm 0.01$ | $10.92 \pm 0.53$ | $3.58 \pm 0.00$ | $10.46 \pm 0.19$ |
| | 6 | $4.72 \pm 0.00$ | $5.39 \pm 0.23$ | $4.46 \pm 0.01$ | $8.60 \pm 0.92$ | $4.12 \pm 0.01$ | $12.35 \pm 0.31$ | $4.41 \pm 0.01$ | $12.65 \pm 0.45$ | $3.72 \pm 0.01$ | $10.98 \pm 0.41$ |
| | 8 | $4.75 \pm 0.00$ | $4.56 \pm 0.20$ | $4.54 \pm 0.01$ | $7.89 \pm 0.85$ | $4.19 \pm 0.01$ | $12.49 \pm 0.56$ | $4.49 \pm 0.01$ | $13.81 \pm 0.47$ | $3.82 \pm 0.01$ | $10.60 \pm 0.24$ |
| SpecTr [38] | 2 | $4.46 \pm 0.01$ | $2.27 \pm 0.29$ | $4.13 \pm 0.01$ | $4.80 \pm 1.04$ | $3.76 \pm 0.01$ | $5.44 \pm 0.54$ | $4.04 \pm 0.01$ | $5.21 \pm 0.43$ | $3.31 \pm 0.01$ | $6.35 \pm 0.34$ |
| | 4 | $4.66 \pm 0.01$ | $4.08 \pm 0.13$ | $4.36 \pm 0.01$ | $7.39 \pm 0.96$ | $4.02 \pm 0.01$ | $10.00 \pm 0.33$ | $4.32 \pm 0.01$ | $10.07 \pm 0.43$ | $3.60 \pm 0.02$ | $10.03 \pm 0.55$ |
| | 6 | $4.74 \pm 0.01$ | $4.30 \pm 0.35$ | $4.48 \pm 0.01$ | $7.70 \pm 0.78$ | $4.13 \pm 0.01$ | $11.51 \pm 0.43$ | $4.43 \pm 0.00$ | $11.77 \pm 0.29$ | $3.72 \pm 0.01$ | $9.97 \pm 0.34$ |
| | 8 | $4.78 \pm 0.00$ | $3.75 \pm 0.17$ | $4.56 \pm 0.01$ | $7.09 \pm 0.77$ | $4.22 \pm 0.01$ | $12.89 \pm 0.96$ | $4.50 \pm 0.01$ | $12.81 \pm 0.50$ | $3.79 \pm 0.01$ | $8.82 \pm 0.32$ |
| Our scheme | 2 | $4.47 \pm 0.00$ | $3.02 \pm 0.28$ | $4.08 \pm 0.01$ | $4.70 \pm 0.78$ | $3.68 \pm 0.02$ | $4.52 \pm 1.42$ | $4.02 \pm 0.00$ | $5.46 \pm 0.27$ | $3.27 \pm 0.01$ | $5.74 \pm 0.57$ |
| | 4 | $4.66 \pm 0.01$ | $5.18 \pm 0.26$ | $4.37 \pm 0.01$ | $8.52 \pm 0.77$ | $3.96 \pm 0.01$ | $9.86 \pm 0.96$ | $4.30 \pm 0.01$ | $10.72 \pm 0.28$ | $3.56 \pm 0.00$ | $9.98 \pm 0.30$ |
| | 6 | $4.74 \pm 0.01$ | $5.33 \pm 0.25$ | $4.47 \pm 0.01$ | $8.54 \pm 1.02$ | $4.10 \pm 0.01$ | $12.14 \pm 0.85$ | $4.43 \pm 0.01$ | $12.79 \pm 0.46$ | $3.73 \pm 0.01$ | $10.90 \pm 0.26$ |
| | 8 | $4.78 \pm 0.00$ | $4.83 \pm 0.24$ | $4.55 \pm 0.01$ | $8.03 \pm 0.97$ | $4.18 \pm 0.00$ | $12.75 \pm 1.14$ | $4.51 \pm 0.01$ | $13.54 \pm 0.41$ | $3.82 \pm 0.01$ | $10.60 \pm 0.34$ |
| Strongly invariant | 2 | $4.45 \pm 0.01$ | $3.02 \pm 0.41$ | $4.03 \pm 0.01$ | $3.18 \pm 0.87$ | $3.65 \pm 0.01$ | $3.56 \pm 1.10$ | $4.00 \pm 0.02$ | $4.29 \pm 0.60$ | $3.25 \pm 0.02$ | $3.74 \pm 0.24$ |
| | 4 | $4.63 \pm 0.00$ | $4.45 \pm 0.57$ | $4.28 \pm 0.01$ | $6.23 \pm 0.76$ | $3.91 \pm 0.01$ | $8.25 \pm 1.01$ | $4.26 \pm 0.01$ | $8.71 \pm 0.17$ | $3.53 \pm 0.01$ | $7.87 \pm 0.44$ |
| | 6 | $4.72 \pm 0.00$ | $5.23 \pm 0.47$ | $4.40 \pm 0.01$ | $6.69 \pm 0.55$ | $4.02 \pm 0.01$ | $9.84 \pm 1.08$ | $4.38 \pm 0.01$ | $10.85 \pm 0.44$ | $3.67 \pm 0.00$ | $7.92 \pm 0.66$ |
| | 8 | $4.76 \pm 0.01$ | $4.49 \pm 0.44$ | $4.46 \pm 0.01$ | $5.99 \pm 0.69$ | $4.09 \pm 0.01$ | $10.32 \pm 1.13$ | $4.44 \pm 0.01$ | $11.46 \pm 0.48$ | $3.75 \pm 0.02$ | $7.40 \pm 0.99$ |
| Daliri et al. [9] | 1 | $4.16 \pm 0.01$ | $0.63 \pm 0.24$ | $3.69 \pm 0.01$ | $0.04 \pm 0.30$ | $3.32 \pm 0.00$ | $-2.23 \pm 0.29$ | $3.61 \pm 0.01$ | $-0.69 \pm 0.33$ | $2.94 \pm 0.01$ | $0.10 \pm 0.46$ |

Table 3: LLM inference with i.i.d. drafts; we use $L = 4$. Token rates (TR) are shown as percentage speedups relative to single-draft speculative decoding, which has mean block efficiency (BE) 4.18, 3.75, 3.43, 3.68, 3.00 and token rate 58.83, 53.08, 48.30, 52.54, 42.44 tokens/s on GSM8K, HumanEval, NaturalReasoning, MBPP and DROP respectively.

| Strategy | Temp. | GSM8K | | HumanEval | | NaturalReasoning | | MBPP | | DROP | |
|---|---|---|---|---|---|---|---|---|---|---|---|
| | | BE | TR (%) | BE | TR (%) | BE | TR (%) | BE | TR (%) | BE | TR (%) |
| SpecInfer [34] | 0.5/1.0 | 4.26 ± 0.02 | 0.06 ± 1.02 | 3.57 ± 0.02 | −1.96 ± 0.67 | 3.04 ± 0.02 | −6.55 ± 0.61 | 3.66 ± 0.01 | −1.87 ± 0.79 | 3.21 ± 0.21 | 4.22 ± 0.99 |
| | 1.0/0.5 | 4.44 ± 0.03 | 4.57 ± 1.80 | 3.80 ± 0.03 | 4.13 ± 1.60 | 3.29 ± 0.02 | 1.12 ± 0.99 | 3.90 ± 0.02 | 4.77 ± 0.55 | 3.31 ± 0.02 | 7.83 ± 1.08 |
| | 1.5/1.0 | 4.63 ± 0.02 | 8.75 ± 1.70 | 3.99 ± 0.03 | 9.67 ± 1.26 | 3.60 ± 0.02 | 11.28 ± 0.88 | 4.05 ± 0.01 | 10.70 ± 0.59 | 3.31 ± 0.02 | 9.11 ± 0.74 |
| | 1.0/1.5 | 4.57 ± 0.03 | 7.43 ± 1.60 | 3.95 ± 0.02 | 8.19 ± 0.82 | 3.44 ± 0.01 | 6.06 ± 0.37 | 3.98 ± 0.02 | 8.48 ± 0.91 | 3.30 ± 0.02 | 9.15 ± 0.83 |
| | 2.0/1.0 | 4.53 ± 0.01 | 6.38 ± 1.43 | 3.88 ± 0.01 | 6.75 ± 0.72 | 3.51 ± 0.02 | 9.12 ± 0.98 | 3.91 ± 0.01 | 6.83 ± 0.63 | 3.19 ± 0.01 | 5.50 ± 0.44 |
| | 1.0/2.0 | 4.55 ± 0.02 | 6.53 ± 1.79 | 3.82 ± 0.02 | 4.73 ± 1.22 | 3.42 ± 0.02 | 5.71 ± 0.73 | 3.91 ± 0.01 | 6.89 ± 0.33 | 3.21 ± 0.02 | 6.28 ± 0.79 |
| | 1.0/1.0 | 4.51 ± 0.02 | 6.02 ± 1.35 | 3.87 ± 0.02 | 6.32 ± 0.36 | 3.36 ± 0.01 | 3.37 ± 0.44 | 3.95 ± 0.02 | 5.17 ± 1.13 | 3.30 ± 0.02 | 9.36 ± 0.93 |
| Our scheme | 0.5/1.0 | 4.75 ± 0.02 | 11.50 ± 1.78 | 4.00 ± 0.01 | 9.80 ± 0.82 | 3.27 ± 0.01 | −0.37 ± 0.15 | 3.94 ± 0.02 | 5.64 ± 0.66 | 3.21 ± 0.01 | 4.67 ± 0.66 |
| | 1.0/0.5 | 4.75 ± 0.02 | 11.40 ± 1.58 | 3.96 ± 0.02 | 8.77 ± 0.99 | 3.25 ± 0.01 | −0.94 ± 0.57 | 3.96 ± 0.02 | 5.99 ± 1.01 | 3.23 ± 0.01 | 4.79 ± 0.51 |
| | 1.5/1.0 | 4.77 ± 0.02 | 11.37 ± 1.81 | 4.03 ± 0.01 | 10.42 ± 0.41 | 3.41 ± 0.01 | 4.58 ± 0.36 | 4.02 ± 0.02 | 9.59 ± 0.83 | 3.22 ± 0.01 | 6.26 ± 0.51 |
| | 1.0/1.5 | 4.77 ± 0.02 | 11.56 ± 1.57 | 3.99 ± 0.03 | 9.17 ± 0.98 | 3.42 ± 0.02 | 4.71 ± 0.88 | 4.00 ± 0.01 | 9.00 ± 0.63 | 3.23 ± 0.02 | 6.19 ± 0.43 |
| | 2.0/1.0 | 4.63 ± 0.02 | 8.15 ± 1.16 | 3.87 ± 0.02 | 5.79 ± 0.77 | 3.31 ± 0.01 | 1.74 ± 0.69 | 3.86 ± 0.03 | 4.95 ± 0.64 | 3.12 ± 0.01 | 2.80 ± 0.36 |
| | 1.0/2.0 | 4.62 ± 0.03 | 8.25 ± 1.29 | 3.87 ± 0.01 | 6.08 ± 0.72 | 3.32 ± 0.01 | 1.67 ± 0.56 | 3.88 ± 0.02 | 5.74 ± 0.43 | 3.13 ± 0.01 | 3.17 ± 0.76 |
| | 1.0/1.0 | 4.83 ± 0.02 | 13.68 ± 1.67 | 4.08 ± 0.02 | 12.15 ± 0.83 | 0.00 ± 0.01 | 4.79 ± 0.91 | 4.08 ± 0.01 | 8.57 ± 0.60 | 3.27 ± 0.01 | 7.67 ± 0.29 |
| Strongly invariant | 0.5/1.0 | 4.22 ± 0.03 | −1.17 ± 1.21 | 3.49 ± 0.04 | −4.74 ± 1.34 | 2.93 ± 0.02 | −10.09 ± 0.96 | 3.60 ± 0.03 | −3.92 ± 1.27 | 3.05 ± 0.02 | −1.12 ± 0.38 |
| | 1.0/0.5 | 4.27 ± 0.03 | −0.07 ± 1.00 | 3.48 ± 0.03 | −5.20 ± 1.31 | 2.95 ± 0.01 | −9.19 ± 0.76 | 3.57 ± 0.03 | −4.49 ± 0.75 | 3.06 ± 0.02 | −0.58 ± 0.96 |
| | 1.5/1.0 | 4.44 ± 0.02 | 3.98 ± 0.91 | 3.56 ± 0.03 | −2.39 ± 0.60 | 3.14 ± 0.01 | −3.31 ± 0.81 | 3.69 ± 0.02 | −0.83 ± 1.34 | 3.05 ± 0.00 | −0.90 ± 0.71 |
| | 1.0/1.5 | 4.35 ± 0.02 | 1.71 ± 0.61 | 3.63 ± 0.03 | 0.85 ± 1.15 | 3.10 ± 0.01 | −4.35 ± 1.16 | 3.71 ± 0.01 | −0.63 ± 0.88 | 3.08 ± 0.02 | 0.22 ± 0.86 |
| | 2.0/1.0 | 4.31 ± 0.02 | 0.68 ± 0.70 | 3.47 ± 0.03 | −5.00 ± 0.97 | 3.02 ± 0.02 | −6.89 ± 1.07 | 3.57 ± 0.01 | −4.24 ± 0.95 | 2.97 ± 0.02 | −3.72 ± 0.68 |
| | 1.0/2.0 | 4.29 ± 0.01 | 0.73 ± 0.73 | 3.50 ± 0.02 | −4.17 ± 1.18 | 3.02 ± 0.02 | −6.78 ± 0.77 | 3.55 ± 0.02 | −4.55 ± 1.21 | 2.98 ± 0.01 | −3.25 ± 0.68 |
| | 1.0/1.0 | 4.34 ± 0.02 | 1.56 ± 0.68 | 3.61 ± 0.01 | −1.30 ± 0.94 | 3.08 ± 0.01 | −4.82 ± 1.14 | 3.73 ± 0.01 | 0.06 ± 0.88 | 3.08 ± 0.02 | 0.39 ± 0.56 |

Table 4: LLM inference with diverse drafts; we use $L = 5$, $K = 2$ and the target temperature is 2.0. The temperatures of drafters 1 and 2 vary and are reported in the second column. Token rates (TR) are shown as percentage speedups relative to single-draft speculative decoding with drafter temperature 1.0, which has mean block efficiency (BE) 4.28, 3.65, 3.19, 3.71, 3.06 and token rate 52.00, 44.15, 38.61, 44.70, 36.42 tokens/s on GSM8K, HumanEval, NaturalReasoning, MBPP and DROP respectively.

| Strategy | GSM8K | | HumanEval | | NaturalReasoning | |
|---|---|---|---|---|---|---|
| | BE | TR (%) | BE | TR (%) | BE | TR (%) |
| SpecInfer [34] | $2.35 \pm 0.00$ | $14.16 \pm 1.55$ | $1.98 \pm 0.01$ | $-0.29 \pm 0.66$ | $2.14 \pm 0.01$ | $11.44 \pm 0.73$ |
| SpecTr [38] | $2.35 \pm 0.01$ | $13.41 \pm 2.10$ | $1.99 \pm 0.01$ | $-0.39 \pm 1.01$ | $2.14 \pm 0.00$ | $11.32 \pm 0.48$ |
| Our scheme | $2.34 \pm 0.01$ | $14.65 \pm 1.90$ | $2.00 \pm 0.01$ | $1.58 \pm 0.74$ | $2.13 \pm 0.00$ | $12.08 \pm 0.38$ |
| Daliri et al. [9] | $1.64 \pm 0.00$ | $1.39 \pm 0.30$ | $1.48 \pm 0.00$ | $2.69 \pm 0.49$ | $1.54 \pm 0.00$ | $2.83 \pm 0.66$ |

Table 5: LLM inference with i.i.d. drafts using $L = 4$; the drafter is Llama 2-68 M and the target is Llama 2-7 B. Token rates (TR) are shown as percentage speedups relative to single-draft speculative decoding, which has mean block efficiency (BE) 1.65, 1.48 and 1.55 on GSM8K, HumanEval and NaturalReasoning respectively.

| Strategy | Tmp. 1/2 | GSM8K | | HumanEval | | MBPP | |
|---|---|---|---|---|---|---|---|
| | | BE | TR (%) | BE | TR (%) | BE | TR (%) |
| SpecInfer [34] | 0.5/1.0 | $2.21 \pm 0.01$ | $-12.51 \pm 1.09$ | $2.61 \pm 0.04$ | $-4.73 \pm 1.67$ | $2.46 \pm 0.03$ | $-5.99 \pm 1.14$ |
| | 1.0/0.5 | $2.44 \pm 0.02$ | $-2.47 \pm 0.88$ | $2.79 \pm 0.03$ | $3.02 \pm 1.17$ | $2.65 \pm 0.02$ | $1.09 \pm 1.85$ |
| | 1.0/1.0 | $2.57 \pm 0.01$ | $2.11 \pm 0.73$ | $2.94 \pm 0.02$ | $7.50 \pm 1.04$ | $2.79 \pm 0.01$ | $6.86 \pm 1.23$ |
| Our scheme | 0.5/1.0 | $2.67 \pm 0.03$ | $5.88 \pm 1.46$ | $3.63 \pm 0.02$ | $33.85 \pm 1.11$ | $3.33 \pm 0.02$ | $25.71 \pm 1.81$ |
| | 1.0/0.5 | $2.69 \pm 0.02$ | $6.33 \pm 1.49$ | $3.67 \pm 0.04$ | $34.67 \pm 1.63$ | $3.31 \pm 0.03$ | $23.87 \pm 1.47$ |
| | 1.0/1.0 | $2.88 \pm 0.01$ | $13.00 \pm 1.12$ | $3.72 \pm 0.03$ | $35.90 \pm 1.40$ | $3.51 \pm 0.03$ | $32.20 \pm 2.37$ |

Table 6: LLM inference with diverse drafts using $L = 5$ and a target temperature of 2.0; the drafter is Llama 3-1 B and the target is Llama 3-8 B. Token rates (TR) are shown as percentage speedups relative to single-draft speculative decoding with drafter temperature 1.0, which has mean block efficiency (BE) 2.42, 2.73 and 2.61 on GSM8K, HumanEval and MBPP respectively.

**Results on Llama models.** To illustrate our speculative decoding framework's effectiveness on other popular families of LLMs, we run a smaller set of experiments on open-source Llama models, mirroring those shown in tables 1 and 2 in the main paper. As in those results, we set $K = 8$. In table 5, we use a lightweight Llama 2-68 M drafter distilled by Miao et al. [34] for SpecInfer coupled with a Llama 2-7 B target, and set $K = 8$. Since the small 68 M drafter performs badly with a temperature mismatch, we use larger models from the Llama 3 series in table 6 with a 1 B drafter and 8 B target. There, we use $K = 2$.

**Empirical matching probability, comparison to the LML bound and scaling with $K$.** We further provide some numerical results in table 7 comparing the empirical average matching probability, using the target and proposal distributions for the next token seen during our speculative decoding experiments, to our lower bound in theorem 1. These demonstrate that the matching probability and lower bound both increase monotonically with $K$, and that the LML bound is quite close to the true probability for practical distributions of interest.

For completeness, we also show below the same empirical matching probabilities for SpecInfer [34] and SpecTr [38] in table 8. Our approach performs similarly to or in many cases better than these methods, even though they do not offer any drafter-invariance property.

**Absolute token rate measurements.** Since concrete performance measurements are tightly coupled to our specific hardware configuration and unrelated optimizations enabled by the LLM inference library, in our case Huggingface Transformers [44], we focus on relative speedups to provide a fair comparison between different speculative decoding algorithms. Nevertheless, to provide context around our reported gains, it may be necessary to convert to throughput in tokens per second. To facilitate this, we provide the throughput for single-draft speculative decoding in the captions of tables 3 and 4. Since all other numbers are relative to this baseline, they may be converted to absolute quantities as needed.

| Dataset | $K = 2$ | | $K = 4$ | | $K = 6$ | | $K = 8$ | |
|---|---|---|---|---|---|---|---|---|
| | Emp. | LML | Emp. | LML | Emp. | LML | Emp. | LML |
| GSM8K | 0.949 | 0.936 | 0.972 | 0.956 | 0.979 | 0.965 | 0.982 | 0.971 |
| HumanEval | 0.919 | 0.903 | 0.950 | 0.931 | 0.962 | 0.945 | 0.971 | 0.953 |
| NaturalReasoning | 0.862 | 0.844 | 0.905 | 0.884 | 0.926 | 0.904 | 0.937 | 0.916 |

Table 7: Empirical (Emp.) matching probability during speculative decoding compared to the LML bound.

| Dataset | K = 2 | | K = 4 | | K = 6 | | K = 8 | |
|---|---|---|---|---|---|---|---|---|
| | SI | ST | SI | ST | SI | ST | SI | ST |
| GSM8K | 0.948 | 0.927 | 0.969 | 0.938 | 0.976 | 0.944 | 0.981 | 0.948 |
| HumanEval | 0.919 | 0.891 | 0.948 | 0.908 | 0.960 | 0.916 | 0.967 | 0.922 |
| NaturalReasoning | 0.864 | 0.837 | 0.904 | 0.864 | 0.921 | 0.878 | 0.932 | 0.887 |

Table 8: Empirical matching probability for both SpecInfer (SI) [34] and SpecTr (ST) [38].

| Dataset | Algorithm | ROUGE-1 | ROUGE-2 | ROUGE-L |
|---|---|---|---|---|
| GSM8K | SpecInfer | $0.737 \pm 0.004$ | $0.592 \pm 0.007$ | $0.647 \pm 0.008$ |
| | Our scheme | $0.801 \pm 0.005$ | $0.701 \pm 0.011$ | $0.745 \pm 0.009$ |
| HumanEval | SpecInfer | $0.656 \pm 0.003$ | $0.466 \pm 0.006$ | $0.545 \pm 0.005$ |
| | Our scheme | $0.710 \pm 0.001$ | $0.555 \pm 0.003$ | $0.628 \pm 0.002$ |
| MBPP | SpecInfer | $0.676 \pm 0.004$ | $0.497 \pm 0.003$ | $0.580 \pm 0.004$ |
| | Our scheme | $0.728 \pm 0.006$ | $0.588 \pm 0.011$ | $0.659 \pm 0.010$ |

Table 9: ROUGE scores measuring decoding consistency for our scheme and the non-invariant SpecInfer [34].

### D.2 Ablation and empirical analysis of drafter invariance

Here, we further justify our definition of drafter invariance by way of an ablation study that sets out to show that adopting drafter-invariant speculative decoding has a positive impact on decoding consistency across runs in practice when the draft model is changed. We also provide some salient examples to demonstrate the property's robustness.

To quantify the effect of drafter invariance on decoding consistency, we run 200 prompts from the GSM8K [7], HumanEval [6] and NaturalReasoning [48] datasets using $K = 2$ drafts. Each prompt is run twice using the same random seed. The drafts are i.i.d., but the drafter model temperature is changed from $1.0$ in the first run to $0.5$ in the second. We compute ROUGE scores [30] to measure the similarity of the two generated output sequences; perfect decoding consistency would mean a score of $1.0$ for all of the ROUGE-1, ROUGE-2 and ROUGE-L metrics. We let the draft length be $L = 4$ and average over 4 random seeds. Results are given in table 9. Our conditionally drafter-invariant scheme givens higher similarity scores in each case compared to the non-invariant SpecInfer baseline, confirming that drafter invariance provides more consistent outputs when faced with changes to the draft model, even though our notion is not strong enough to guarantee that the outputs always remain exactly the same. Moreover, in appendix B we demonstrate a slightly modified scheme that provides what we refer to as *strong drafter invariance*, which provably gives perfect decoding consistency. However, this comes at a cost to the block efficiency, as seen in tables 3 and 4.

While the ROUGE scores from our ablation study provide a more comprehensive way to examine the real-world impact of drafter invariance, we also give a concrete example from GSM8K where non-invariant speculative decoding causes the output to diverge significantly from the original after changing the draft model, all while retaining the same random seed. We adopt the same setting as before with $K = 2$ i.i.d. drafts and the draft model temperature changing from $1.0$ to $0.5$ between runs, comparing against SpecInfer [34]. It should be emphasized that all forms of speculative decoding preserve the output distribution exactly; therefore, it is not the quality of outputs that is degraded but rather their stability or consistency faced with changes to the drafter. In this particular example, our scheme achieves perfect consistency between the two trials; we do not guarantee this in general, but the ROUGE scores from our ablation study demonstrate that we do perform better in this sense on average.

*Prompt:*

```
Weng earns $12 an hour for babysitting. Yesterday, she just did 50 minutes of
babysitting. How much did she earn?
```

*SpecInfer (not invariant), draft temperature 1.0:*

```
To determine how much Weng earned for babysitting for 50 minutes, we first need to
convert the time from minutes to hours. We know that there are 60 minutes in an
hour, so we can convert 50 minutes to hours by dividing 50 by 60:
\[
```

```
\text{Time in hours} = \frac{50}{60} = \frac{5}{6} \text{ hours}
\]
Next, we calculate the earnings by multiplying the time in hours by her hourly rate
of $12:
\[
\text{Earnings} = 12 \times \
```

*SpecInfer (not invariant), draft temperature 0.5:*

```
To determine how much Weng earned for her 50 minutes of babysitting, we need to
follow these steps:
1. Identify Weng's hourly wage.
2. Convert the babysitting time from minutes to hours.
3. Calculate the earnings based on the time worked.
First, Weng's hourly wage is $12. Since she babysat for 50 minutes, we need to
convert 50 minutes into hours. There are 60 minutes in an hour, so we can convert 50
minutes to hours by dividing 50 by 60:
\[ \text{Time in hours} = \frac{
```

*Our scheme (invariant), draft temperature 1.0:*

```
To determine how much Weng earned for 50 minutes of babysitting at a rate of $12 per
hour, we need to follow these steps:
1. Convert the time from minutes to hours.
2. Multiply the time in hours by the hourly rate.
First, we convert 50 minutes to hours. Since there are 60 minutes in an hour, we
divide 50 by 60:
\[
\frac{50}{60} = \frac{5}{6} \text{ hours}
\]
Next, we multiply the time in hours by the hourly rate of $12:
```

*Our scheme (invariant), draft temperature 0.5:*

```
To determine how much Weng earned for 50 minutes of babysitting at a rate of $12 per
hour, we need to follow these steps:
1. Convert the time from minutes to hours.
2. Multiply the time in hours by the hourly rate.
First, we convert 50 minutes to hours. Since there are 60 minutes in an hour, we
divide 50 by 60:
\[
\frac{50}{60} = \frac{5}{6} \text{ hours}
\]
Next, we multiply the time in hours by the hourly rate of $12:
```

### D.3 Synthetic Gaussian source

In this section, we provide some key derivations and details of our experimental procedure for evaluating GLS on the synthetic Gaussian source, and give more numerical results.

**Decoder target distribution.** To determine the decoder target distribution $p_{W|T}$, we first recapitulate the problem setting and the random variables involved. The Gaussian source is $A \sim \mathcal{N}(0,1)$ while the target distribution at the encoder given $A = a$ is $p_{W|A}(\cdot \mid a) = \mathcal{N}(a, \sigma_{W|A}^2)$. Meanwhile, the side information at decoder $k$ is $T_k = A + \zeta_k$, where $\zeta_k \sim \mathcal{N}(0, \sigma_{T|A}^2)$. Since we are only analyzing one decoder individually, we will drop the $k$ subscript in what follows and more simply write $T$ and $\zeta$. To summarize, we have

$$W = A + \eta \text{ and } T = A + \zeta$$

where $\eta$ and $\zeta$ are independent zero-mean Gaussians with variances $\sigma_\eta^2 = \sigma_{W|A}^2$ and $\sigma_\zeta^2 = \sigma_{T|A}^2$ respectively. From this, we see that $W$ and $T$ are jointly distributed as

$$\begin{bmatrix} W \\ T \end{bmatrix} \sim \mathcal{N}\left(0, \Sigma_{W,T}\right), \text{ where } \Sigma_{W,T} = \begin{bmatrix} \mathrm{E}[W^2] & \mathrm{E}[WT] \\ \mathrm{E}[TW] & \mathrm{E}[T^2] \end{bmatrix} = \begin{bmatrix} 1 + \sigma_\eta^2 & 1 \\ 1 & 1 + \sigma_\zeta^2 \end{bmatrix}. \quad (19)$$

The variance of $W$ is $\sigma_W^2 = 1 + \sigma_\eta^2$ and that of $T$ is $\sigma_T^2 = 1 + \sigma_\zeta^2$. We then know that $p_{W|T}$ is a Gaussian distribution with mean and variance

$$\mu_{W|T} = \frac{T}{1 + \sigma_\zeta^2} = \frac{T}{\sigma_T^2}, \ \sigma_{W|T}^2 = 1 + \sigma_\eta^2 - \frac{1}{1 + \sigma_\zeta^2} = \sigma_W^2 - \frac{1}{\sigma_T^2}.$$

That is, $p_{W|T}(\cdot \mid t) = \mathcal{N}(t/\sigma_T^2, \sigma_W^2 - 1/\sigma_T^2)$ as asserted in the main paper.

**MMSE estimator.** We now derive the MMSE estimator for the synthetic Gaussian source when side information is available at the decoder. To find the estimator, we assume that the encoder and decoder indices match, i.e. $W_k = W$. Recall that proposition 4 in the main paper gives a lower bound for the probability of this event. We proceed by finding the joint distribution of $A$, $W$ and $T$. In fact,

$$\begin{bmatrix} A \\ W \\ T \end{bmatrix} \sim \mathcal{N} \left( 0, \begin{bmatrix} 1 & \Sigma_{A,(W,T)} \\ \Sigma_{(W,T),A} & \Sigma_{W,T} \end{bmatrix} \right), \text{ where } \Sigma_{(W,T),A} = \Sigma_{A,(W,T)}^T = \begin{bmatrix} \mathrm{E}[AW] \\ \mathrm{E}[AT] \end{bmatrix} = \begin{bmatrix} 1 \\ 1 \end{bmatrix}$$

and $\Sigma_{W,T}$ was found earlier in (19). Then, the MMSE estimate is

$$\hat{A} = \mathrm{E}[A \mid W, \ T] = \Sigma_{A,(W,T)} \Sigma_{W,T}^{-1} \begin{bmatrix} W \\ T \end{bmatrix} = \frac{\sigma_\zeta^2 W + \sigma_\eta^2 T}{\sigma_\eta^2 + \sigma_\zeta^2 + \sigma_\eta^2 \sigma_\zeta^2}.$$

To conclude, given $T_k = t_k$ and $W_k = w_k$, the reconstruction output by decoder $k$ is given by

$$g(w_k, t_k) = \frac{\sigma_\zeta^2 w_k + \sigma_\eta^2 t_k}{\sigma_\eta^2 + \sigma_\zeta^2 + \sigma_\eta^2 \sigma_\zeta^2}.$$

**Experiment parameters and further results.** First, we briefly outline the parameters used for the experiment. The number of samples from the prior is $N = 2^{15}$ for all tests and the source, as mentioned in the main paper, is $A \sim \mathcal{N}(0,1)$. The conditional variance of the side information given $A$ is fixed at $\sigma_{T|A}^2 = 0.5$ throughout. We control the rate by varying $L_{\max}$, considering $L_{\max} \in \{2^1, 2^2, 2^3, 2^4, 2^5, 2^6\}$. For each $L_{\max}$, the resulting distortion is minimized over the encoder's target distribution by exploring different values of $\sigma_{W|A}^2$, selecting from $\sigma_{W|A}^2 \in \{0.01, 0.008, 0.006, 0.005, 0.003, 0.002, 0.001\}$ and choosing the best across $10^4$ trials. The distortion incurred by the best configuration is then further evaluated on $10^5$ trials. This procedure is carried out for $K \in \{1, 2, 3, 4\}$, where $K$ is the number of decoders. Finally, the entire experiment is repeated 10 times and the results are averaged to obtain those reported in table 10. The error bars show one standard error of the mean, calculated as in appendix D.1 using all 10 trials.

Running one full repetition of the experiment takes around 4 hours when performing the calculations on a Nvidia Tesla T4 GPU with 16 GB of memory. The exact same procedure is used to generate results for the baseline scheme described in the main paper, and these are shown in table 11. We also show the value of $\sigma_{W|A}^2$ that most often minimizes the distortion in each case.

| $K$ | $L_{\max}$ | $\sigma_{W|A}^2$ | Distortion (dB) | $K$ | $L_{\max}$ | $\sigma_{W|A}^2$ | Distortion (dB) |
|---|---|---|---|---|---|---|---|
| 1 | $2^1$ | 0.008 | $-9.7032 \pm 0.0193$ | 3 | $2^1$ | 0.005 | $-18.3884 \pm 0.0163$ |
| | $2^2$ | 0.010 | $-12.7474 \pm 0.0226$ | | $2^2$ | 0.003 | $-21.6187 \pm 0.0164$ |
| | $2^3$ | 0.010 | $-16.0116 \pm 0.0369$ | | $2^3$ | 0.001 | $-25.0515 \pm 0.0213$ |
| | $2^4$ | 0.003 | $-19.5491 \pm 0.0237$ | | $2^4$ | 0.001 | $-28.5329 \pm 0.0128$ |
| | $2^5$ | 0.002 | $-23.4012 \pm 0.0132$ | | $2^5$ | 0.001 | $-31.2575 \pm 0.0161$ |
| | $2^6$ | 0.001 | $-27.3470 \pm 0.0183$ | | $2^6$ | 0.001 | $-33.1515 \pm 0.0108$ |
| 2 | $2^1$ | 0.010 | $-15.2069 \pm 0.0148$ | 4 | $2^1$ | 0.005 | $-20.6834 \pm 0.0176$ |
| | $2^2$ | 0.005 | $-18.3377 \pm 0.0164$ | | $2^2$ | 0.001 | $-23.9418 \pm 0.0197$ |
| | $2^3$ | 0.002 | $-21.7032 \pm 0.0101$ | | $2^3$ | 0.001 | $-27.4313 \pm 0.0106$ |
| | $2^4$ | 0.001 | $-25.3886 \pm 0.0104$ | | $2^4$ | 0.001 | $-30.4379 \pm 0.0188$ |
| | $2^5$ | 0.001 | $-28.8619 \pm 0.0169$ | | $2^5$ | 0.001 | $-32.6616 \pm 0.0106$ |
| | $2^6$ | 0.001 | $-31.4737 \pm 0.0152$ | | $2^6$ | 0.001 | $-34.1082 \pm 0.0102$ |

Table 10: Results using GLS with a Gaussian source.

| $K$ | $L_{\max}$ | $\sigma^2_{W|A}$ | Distortion (dB) | $K$ | $L_{\max}$ | $\sigma^2_{W|A}$ | Distortion (dB) |
|---|---|---|---|---|---|---|---|
| 1 | $2^1$ | 0.010 | $-9.7163 \pm 0.0195$ | 3 | $2^1$ | 0.010 | $-13.8197 \pm 0.0368$ |
|   | $2^2$ | 0.008 | $-12.6968 \pm 0.0273$ |   | $2^2$ | 0.010 | $-17.4640 \pm 0.0211$ |
|   | $2^3$ | 0.008 | $-16.0124 \pm 0.0153$ |   | $2^3$ | 0.010 | $-21.0096 \pm 0.0185$ |
|   | $2^4$ | 0.003 | $-19.5518 \pm 0.0270$ |   | $2^4$ | 0.003 | $-24.6996 \pm 0.0126$ |
|   | $2^5$ | 0.001 | $-23.3905 \pm 0.0105$ |   | $2^5$ | 0.001 | $-28.6864 \pm 0.0123$ |
|   | $2^6$ | 0.001 | $-27.3705 \pm 0.0135$ |   | $2^6$ | 0.001 | $-32.0109 \pm 0.0156$ |
| 2 | $2^1$ | 0.010 | $-12.5143 \pm 0.0356$ | 4 | $2^1$ | 0.010 | $-14.6125 \pm 0.0272$ |
|   | $2^2$ | 0.010 | $-15.9916 \pm 0.0205$ |   | $2^2$ | 0.010 | $-18.3350 \pm 0.0157$ |
|   | $2^3$ | 0.008 | $-19.4843 \pm 0.0137$ |   | $2^3$ | 0.008 | $-21.9300 \pm 0.0238$ |
|   | $2^4$ | 0.003 | $-23.1495 \pm 0.0240$ |   | $2^4$ | 0.002 | $-25.5269 \pm 0.0210$ |
|   | $2^5$ | 0.001 | $-27.1988 \pm 0.0092$ |   | $2^5$ | 0.001 | $-29.4994 \pm 0.0100$ |
|   | $2^6$ | 0.001 | $-30.7772 \pm 0.0169$ |   | $2^6$ | 0.001 | $-32.7141 \pm 0.0208$ |

Table 11: Results using the baseline decoding scheme with a Gaussian source.

## D.4 Distributed image compression

We now give details on our distributed image compression experiments.

**Neural network architectures.** The following notations are used to denote the different layers in our networks:

1. $\mathrm{conv}(a, b, c, d, e)$: A convolution layer with $a$ input features, $b$ output features, kernel size $c$, stride $d$ and input padding $e$.
2. $\mathrm{upconv}(a, b, c, d, e, f)$: A transposed convolution layer with $a$ input features, $b$ output features, kernel size $c$, stride $d$, input padding $e$ and output padding $f$.
3. $\mathrm{fc}(a, b)$: A fully-connected layer with input size $a$ and output size $b$.
4. $\mathrm{do}(p)$: A dropout layer with dropout probability $p$.
5. $\mathrm{cat}(a, b)$: Concatenates two tensors of shapes $a$ and $b$.

The network layers are enumerated in table 12 and follow the network constructions in Phan et al. [36]. The encoder's target distribution $p_{W|A}$ is taken to be a four-dimensional Gaussian with uncorrelated components, where the mean and variance of each component are generated by the encoder network from an input image. More precisely, if we let the image be $a$, the encoder network produces two embeddings $e_1(a)$ and $e_2(a)$, each in $\mathbb{R}^4$. Then, $p_{W|A}(\cdot \mid a) = \mathcal{N}(e_1(a), \mathrm{diag}(e_2(a)))$, and we arbitrarily choose $W \sim \mathcal{N}(0, 1)$ as the marginal distribution, which is also the $\beta$-VAE's prior. On the other hand, decoder $k$ is tasked with generating a reconstruction given the side information $t_k$ and an embedding $w_k \in \mathbb{R}^4$, which is selected depending on the message sent by the encoder. Rather than using the $14 \times 14$ side information image directly, we employ a projection network to extract a length-128 feature vector $e(t_k)$ before feeding this representation into the decoder network along with $w_k$ to get $\hat{a}^{(k)} = g(w_k, e(t_k))$ for $1 \leq k \leq K$. The final estimate $\hat{a}$ is chosen from among the $\hat{a}^{(k)}$'s such that the distortion is minimized.

The estimator network is another important component of our compression protocol, since it acts as a proxy for $p_{W|T}$. Recall from section 5.1 and its extension in appendix C that this distribution is used to select the index at the decoder; using this index, decoder $k$ picks $W_k$ from the shared list of samples taken from the prior. In practice, the estimator network takes a $14 \times 14$ side information image as its input and extracts 128-dimensional features, which are then concatenated with a sample $w \in \mathbb{R}^4$. The final part of the network is classifies whether this joint embedding comes from the joint distribution $p_{W,T}$ or the product of the marginals $p_W p_T$. Its output therefore stands in for $p_{W|T}$.

**Network loss functions.** The $\beta$-VAE is trained using the rate-distortion loss function

$$\mathcal{L}_{\mathrm{VAE}}(a, \hat{a}) = \beta(a - \hat{a}) - D_{\mathrm{KL}}[p_{W|A}(\cdot \mid a) \,\|\, p_W].$$

Note that since the marginal and conditional distributions $p_W$ and $p_{W|A}$ are both Gaussian, $D_{\mathrm{KL}}$ has a closed form. The neural estimator uses the binary cross-entropy (BCE) loss function. If we let its output as a function of side information $t$ and given sample $w$ be $h(w, t)$, the loss is

$$\mathcal{L}_{\mathrm{estimator}}(w, t) = \mathrm{BCE}(h(w, t), \mathbb{1}\{w \text{ was sampled from } p_{W|T}(\cdot \mid t)\})$$

| 0 | Input $(1 \times 28 \times 28)$ |
|---|---|
| 1 | $\text{conv}(1, 128, 3, 1, 1)$, ReLU |
| 2 | $\text{conv}(128, 128, 3, 2, 1)$, ReLU |
| 3 | $\text{conv}(128, 128, 3, 2, 1)$, ReLU |
| 4 | $\text{fc}(6272, 512)$, ReLU |
| 5 | $\text{fc}(512, 8)$ |

(a) Encoder

| 0 | Input $(132)$ |
|---|---|
| 1 | $\text{fc}(132, 512)$, ReLU |
| 2 | $\text{fc}(132, 6272)$, ReLU |
| 3 | $\text{upconv}(128, 64, 3, 2, 1, 1)$, ReLU |
| 4 | $\text{upconv}(64, 32, 3, 2, 1, 1)$, $\text{do}(0.5)$, ReLU |
| 5 | $\text{upconv}(32, 1, 3, 1, 1, 0)$, tanh |

(b) Decoder

| 0 | Input $(1 \times 14 \times 14)$ |
|---|---|
| 1 | $\text{conv}(1, 32, 3, 1, 1)$, ReLU |
| 2 | $\text{conv}(32, 64, 3, 2, 1)$, ReLU |
| 3 | $\text{conv}(64, 128, 3, 2, 1)$, ReLU |
| 4 | $\text{fc}(2048, 512)$, ReLU |
| 5 | $\text{fc}(512, 128)$ |

(c) Projection

| 0 | Input $(1 \times 14 \times 14)$ |
|---|---|
| 1 | $\text{conv}(1, 32, 3, 1, 1)$, ReLU |
| 2 | $\text{conv}(32, 64, 3, 2, 1)$, ReLU |
| 3 | $\text{conv}(64, 128, 3, 2, 1)$, ReLU |
| 4 | $\text{fc}(2048, 512)$, ReLU |
| 5 | $\text{fc}(512, 128)$, $\text{cat}(128, 4)$ |
| 6 | $\text{fc}(132, 128)$, LeakyReLU |
| 7 | $\text{fc}(128, 128)$, LeakyReLU |
| 8 | $\text{fc}(128, 128)$, LeakyReLU |
| 9 | $\text{fc}(128, 128)$, LeakyReLU |
| 10 | $\text{fc}(128, 1)$, Sigmoid |

(d) Estimator

Table 12: Neural network architectures.

where $\mathbb{1}$ is the indicator function.

**Training and evaluation procedure.** We use the MNIST dataset [24] with the usual train-test split of $60\,000$ and $10\,000$ images respectively and batch size $64$. All models are trained for 30 epochs on a Nvidia Tesla T4 GPU with $16\,$GB of memory using the Adam optimizer [21]. The learning rate is $10^{-3}$ and we set $\beta_1 = 0.9$, $\beta_2 = 0.99$. The encoder, decoder, projection and estimator networks are trained jointly in an end-to-end manner, and we create sets of models for $\beta \in \{0.15, 0.35, 0.55, 0.75, 0.95\}$ to cover a broad range of rate-distortion tradeoffs at the encoder side. Jointly training the networks takes around 45 minutes for each $\beta$.

At test time, we vary $L_{\max}$ to control the rate, considering $L_{\max} \in \{2^2, 2^3, 2^4, 2^5, 2^6\}$. For each configuration, we additionally optimize over $N$, which is the number of samples from the prior, and the VAE parameter $\beta$ using a grid search where $N \in \{2^7, 2^8, 2^9, 2^{10}, 2^{11}, 2^{12}\}$ and $\beta \in \{0.15, 0.35, 0.55, 0.75, 0.95\}$. The experiment is repeated 5 times and the results averaged, with the same procedure also being followed for the baseline scheme described in the main paper. We provide error bars showing one standard error of the mean, which is again calculated as in appendix D.1 with the number of trials now being 5. Each instance of the full experiment takes approximately 6 hours to run, and this is done for $K \in \{1, 2, 3, 4\}$. Complete results are given in tables 13 and 14, where we also give the values of $N$ and $\beta$ that are most often optimal in each case.

| $K$ | $L_{\max}$ | $N$ | $\beta$ | MSE | $K$ | $L_{\max}$ | $N$ | $\beta$ | MSE |
|---|---|---|---|---|---|---|---|---|---|
| 1 | $2^2$ | $2^7$ | 0.15 | $0.1027 \pm 0.0002$ | 3 | $2^2$ | $2^9$ | 0.15 | $0.0791 \pm 0.0000$ |
|  | $2^3$ | $2^7$ | 0.15 | $0.0942 \pm 0.0001$ |  | $2^3$ | $2^7$ | 0.15 | $0.0738 \pm 0.0002$ |
|  | $2^4$ | $2^7$ | 0.15 | $0.0852 \pm 0.0002$ |  | $2^4$ | $2^7$ | 0.15 | $0.0694 \pm 0.0001$ |
|  | $2^5$ | $2^7$ | 0.15 | $0.0766 \pm 0.0001$ |  | $2^5$ | $2^7$ | 0.15 | $0.0660 \pm 0.0001$ |
|  | $2^6$ | $2^7$ | 0.15 | $0.0693 \pm 0.0002$ |  | $2^6$ | $2^8$ | 0.35 | $0.0599 \pm 0.0002$ |
| 2 | $2^2$ | $2^9$ | 0.15 | $0.0860 \pm 0.0001$ | 4 | $2^2$ | $2^7$ | 0.15 | $0.0751 \pm 0.0001$ |
|  | $2^3$ | $2^7$ | 0.15 | $0.0792 \pm 0.0001$ |  | $2^3$ | $2^7$ | 0.15 | $0.0710 \pm 0.0001$ |
|  | $2^4$ | $2^7$ | 0.15 | $0.0734 \pm 0.0001$ |  | $2^4$ | $2^7$ | 0.15 | $0.0671 \pm 0.0001$ |
|  | $2^5$ | $2^7$ | 0.15 | $0.0687 \pm 0.0002$ |  | $2^5$ | $2^8$ | 0.35 | $0.0635 \pm 0.0001$ |
|  | $2^6$ | $2^7$ | 0.35 | $0.0636 \pm 0.0002$ |  | $2^6$ | $2^8$ | 0.35 | $0.0564 \pm 0.0001$ |

Table 13: Results using GLS for distributed image compression on MNIST.

**More compression experiments on CIFAR-10.** To show that GLS generalizes to more complex image datasets, we provide some further results on CIFAR-10 [23] under the same setup as our

| $K$ | $L_{\max}$ | $N$ | $\beta$ | MSE | $K$ | $L_{\max}$ | $N$ | $\beta$ | MSE |
|---|---|---|---|---|---|---|---|---|---|
| 1 | $2^2$ | $2^7$ | 0.15 | $0.1025 \pm 0.0001$ | 3 | $2^2$ | $2^7$ | 0.15 | $0.0906 \pm 0.0002$ |
|   | $2^3$ | $2^7$ | 0.15 | $0.0937 \pm 0.0002$ |   | $2^3$ | $2^7$ | 0.15 | $0.0832 \pm 0.0003$ |
|   | $2^4$ | $2^7$ | 0.15 | $0.0850 \pm 0.0002$ |   | $2^4$ | $2^7$ | 0.15 | $0.0757 \pm 0.0002$ |
|   | $2^5$ | $2^7$ | 0.15 | $0.0764 \pm 0.0002$ |   | $2^5$ | $2^7$ | 0.15 | $0.0704 \pm 0.0001$ |
|   | $2^6$ | $2^7$ | 0.15 | $0.0693 \pm 0.0002$ |   | $2^6$ | $2^7$ | 0.35 | $0.0653 \pm 0.0001$ |
| 2 | $2^2$ | $2^7$ | 0.15 | $0.0941 \pm 0.0002$ | 4 | $2^2$ | $2^9$ | 0.15 | $0.0886 \pm 0.0001$ |
|   | $2^3$ | $2^7$ | 0.15 | $0.0865 \pm 0.0002$ |   | $2^3$ | $2^7$ | 0.15 | $0.0815 \pm 0.0001$ |
|   | $2^4$ | $2^7$ | 0.15 | $0.0783 \pm 0.0002$ |   | $2^4$ | $2^7$ | 0.15 | $0.0747 \pm 0.0002$ |
|   | $2^5$ | $2^7$ | 0.15 | $0.0718 \pm 0.0002$ |   | $2^5$ | $2^7$ | 0.15 | $0.0694 \pm 0.0002$ |
|   | $2^6$ | $2^7$ | 0.15 | $0.0669 \pm 0.0001$ |   | $2^6$ | $2^7$ | 0.35 | $0.0639 \pm 0.0002$ |

Table 14: Results using the baseline decoding scheme for distributed image compression on MNIST.

MNIST experiments. The side information samples are now $16 \times 16$ patches uniformly selected from the left half of the 3-channel, $32 \times 32$ image. Results for the MSE distortion are shown in tables 15 and 16, where we vary both $K$ and $L_{\max}$. Note that the baseline and GLS compression schemes are functionally identical for $K = 1$, but as is the case on MNIST, GLS offers gains for $K > 1$. For each $K, L_{\max}$ pair, setting $N = 2^7$ and $\beta = 0.15$ gave the strongest results.

| $K$ | $L_{\max}$ | $N$ | $\beta$ | MSE | $K$ | $L_{\max}$ | $N$ | $\beta$ | MSE |
|---|---|---|---|---|---|---|---|---|---|
| 1 | $2^2$ | $2^7$ | 0.15 | $0.1119 \pm 0.0003$ | 3 | $2^2$ | $2^7$ | 0.15 | $0.0839 \pm 0.0001$ |
|   | $2^3$ | $2^7$ | 0.15 | $0.1054 \pm 0.0002$ |   | $2^3$ | $2^7$ | 0.15 | $0.0778 \pm 0.0003$ |
|   | $2^4$ | $2^7$ | 0.15 | $0.0963 \pm 0.0001$ |   | $2^4$ | $2^7$ | 0.15 | $0.0710 \pm 0.0001$ |
|   | $2^5$ | $2^7$ | 0.15 | $0.0846 \pm 0.0003$ |   | $2^5$ | $2^7$ | 0.15 | $0.0641 \pm 0.0000$ |
|   | $2^6$ | $2^7$ | 0.15 | $0.0710 \pm 0.0001$ |   | $2^6$ | $2^7$ | 0.15 | $0.0590 \pm 0.0001$ |
| 2 | $2^2$ | $2^7$ | 0.15 | $0.0925 \pm 0.0001$ | 4 | $2^2$ | $2^7$ | 0.15 | $0.0788 \pm 0.0001$ |
|   | $2^3$ | $2^7$ | 0.15 | $0.0860 \pm 0.0002$ |   | $2^3$ | $2^7$ | 0.15 | $0.0733 \pm 0.0001$ |
|   | $2^4$ | $2^7$ | 0.15 | $0.0787 \pm 0.0003$ |   | $2^4$ | $2^7$ | 0.15 | $0.0673 \pm 0.0002$ |
|   | $2^5$ | $2^7$ | 0.15 | $0.0698 \pm 0.0003$ |   | $2^5$ | $2^7$ | 0.15 | $0.0618 \pm 0.0001$ |
|   | $2^6$ | $2^7$ | 0.15 | $0.0616 \pm 0.0002$ |   | $2^6$ | $2^7$ | 0.15 | $0.0579 \pm 0.0001$ |

Table 15: Results using GLS for distributed image compression on CIFAR-10.

| $K$ | $L_{\max}$ | $N$ | $\beta$ | MSE | $K$ | $L_{\max}$ | $N$ | $\beta$ | MSE |
|---|---|---|---|---|---|---|---|---|---|
| 1 | $2^2$ | $2^7$ | 0.15 | $0.1113 \pm 0.0001$ | 3 | $2^2$ | $2^7$ | 0.15 | $0.0982 \pm 0.0002$ |
|   | $2^3$ | $2^7$ | 0.15 | $0.1048 \pm 0.0002$ |   | $2^3$ | $2^7$ | 0.15 | $0.0914 \pm 0.0002$ |
|   | $2^4$ | $2^7$ | 0.15 | $0.0959 \pm 0.0003$ |   | $2^4$ | $2^7$ | 0.15 | $0.0831 \pm 0.0002$ |
|   | $2^5$ | $2^7$ | 0.15 | $0.0846 \pm 0.0003$ |   | $2^5$ | $2^7$ | 0.15 | $0.0734 \pm 0.0004$ |
|   | $2^6$ | $2^7$ | 0.15 | $0.0710 \pm 0.0002$ |   | $2^6$ | $2^7$ | 0.15 | $0.0639 \pm 0.0001$ |
| 2 | $2^2$ | $2^7$ | 0.15 | $0.1019 \pm 0.0003$ | 4 | $2^2$ | $2^7$ | 0.15 | $0.0956 \pm 0.0002$ |
|   | $2^3$ | $2^7$ | 0.15 | $0.0958 \pm 0.0002$ |   | $2^3$ | $2^7$ | 0.15 | $0.0888 \pm 0.0002$ |
|   | $2^4$ | $2^7$ | 0.15 | $0.0867 \pm 0.0003$ |   | $2^4$ | $2^7$ | 0.15 | $0.0812 \pm 0.0001$ |
|   | $2^5$ | $2^7$ | 0.15 | $0.0767 \pm 0.0001$ |   | $2^5$ | $2^7$ | 0.15 | $0.0718 \pm 0.0003$ |
|   | $2^6$ | $2^7$ | 0.15 | $0.0659 \pm 0.0002$ |   | $2^6$ | $2^7$ | 0.15 | $0.0629 \pm 0.0001$ |

Table 16: Results using the baseline decoding scheme for distributed image compression on CIFAR-10.

