# OpenReview forum: "List-Level Distribution Coupling with Applications to Speculative Decoding and Lossy Compression"
_NeurIPS.cc/2025/Conference — NeurIPS 2025 poster_

### Official Review · Reviewer_s4tY · 2025-06-25

**Clarity:** 3
**Significance:** 2
**Originality:** 2
**Rating:** 3
**Confidence:** 4

**Summary:**

This paper introduces Gumbel-max List Sampling (GLS), a new algorithmic framework for sampling from discrete distributions when one party samples a list of size K, and a match is declared if any of the K samples equals another sample drawn from a target distribution. The authors provide a theoretical lower bound on the matching probability, named the List Matching Lemma (LML), which extends classical coupling bounds.
Two primary applications are discussed:
Multi-draft Speculative Decoding for LLMs: The paper proposes a drafter-invariant approach based on GLS, outperforming or matching baselines like SpecInfer and SpecTr in various decoding settings.
Distributed Lossy Compression with Side Information: A generalized version of GLS is applied to a Wyner-Ziv-like compression setup, achieving gains over baselines on synthetic Gaussian and MNIST data.

**Questions:**

1) Theoretical Tightness: Is the list matching lemma (LML) tight in the worst case, or can it be improved further? Are there classes of distributions for which the bound is known to be loose?
2) Drafter Invariance in Practice: Can you show a case where non-invariant speculative decoding causes divergence or instability during inference? An empirical ablation would strengthen the case for its necessity.
3) Compression Baselines: Why not include more recent variational or neural compression schemes as baselines for the MNIST experiments? Shared randomness baselines are quite weak.
4) Computational Efficiency: Could you provide concrete latency numbers (e.g., ms/token) for your method vs SpecInfer, especially on large vocabularies?
5) Scalability: Have you considered how GLS performs in high-cardinality spaces, e.g., vocab sizes 50k–100k? Memory usage of O(NK) could become problematic.

**Ethical Concerns:**

["NO or VERY MINOR ethics concerns only"]

**Limitations:**

The paper discusses limitations to some extent, particularly the extension to continuous distributions and finite sample approximations via importance sampling. However, more explicit discussion of computational overhead and failure modes (e.g., degraded performance at high temperature mismatches, decoding bias) would improve transparency.

**Paper Formatting Concerns:**

None noticed.

**Quality:**

3

**Strengths And Weaknesses:**

Strengths
1) Theoretical Rigor: The derivation of the LML and conditional LML is mathematically sound and nicely generalizes previous coupling bounds.
2) Conceptual Clarity: The extension from one-to-one matching to one-to-many (list-level) is natural yet nontrivial, and the paper articulates this progression well.
3) Novelty in Applications:
- Drafter-invariant speculative decoding is a valuable contribution in a field where practical methods often lack such guarantees.
- The compression setup with multiple decoders and side information is well-motivated and leverages the theoretical core effectively.
4) Experimental Coverage: LLM experiments cover both i.i.d. and diverse draft models; compression is evaluated on synthetic and real data (MNIST).
5) Error Analysis: The authors carefully analyze matching probability and rate-distortion trade-offs under different setups.

Weaknesses
1) Theoretical Originality: While well-executed, the core theoretical contribution (GLS + LML) can be seen as a relatively incremental extension of Gumbel-max sampling and Poisson Matching Lemma (PML). The analogy is strong but not fundamentally novel.
2) Limited Empirical Diversity:
- Speculative decoding experiments use only Qwen 2.5 models. No results on popular open-source LLMs like LLaMA, Mistral, etc.
- Compression baselines are somewhat weak; only a shared-randomness baseline is used. No comparisons with alternative list-decoding or variational schemes from recent literature.
3) Ablation Gaps: There’s no ablation of the importance of drafter invariance, nor is there a comparison of acceptance probabilities with non-GLS alternatives under similar resource constraints.
4) Algorithmic Complexity: While the method is conceptually simple, it involves O(NK) operations and repeated argmins, which can be computationally intensive in high-cardinality alphabets. The authors mention efficiency but don’t provide run-time benchmarks.
5) Limited Discussion of Limitations: While some limitations are acknowledged, key concerns — e.g., applicability of GLS to large vocabulary sizes and scalability of decoder-side computation — could be better articulated.

---

> ### Author Rebuttal · Authors · 2025-07-30
>
> Thank you for your detailed response. We are glad that you found our presentation of GLS clear and appreciate your comments on the mathematical rigor and novelty of the applications. Please find our responses below:
>
> 1. **Originality:**
>    While we agree that our results are natural extensions of Gumbel-max sampling [1] and PML [2], we believe that our method's simplicity is a strength rather than an indicator of insufficient novelty, which is a view shared by reviewers W4vi and fX9A. While our generalization appears simple in hindsight, it is not trivial and requires careful proof. Also, our new list-level setting is more than a theoretical curiosity, having practical applications to LLM inference and distributed compression. We therefore feel that our contribution demonstrates sufficient originality despite the core idea's simplicity.
>
> 2. **Empirical diversity:**
>    While Qwen2.5 models are popular open-source LLMs and are representative of many current offerings, we agree that adding experiments on other LLMs would strengthen our claims. We provide results below mirroring those in Tabs. 1 and 2 but for Llama models, showing block efficiencies and percentage token-rate speedups. The first table uses a Llama2 68M drafter distilled by Miao et al. [3], coupled with a Llama2 7B target. As in Tab. 1, this experiment uses $K=8$. Since the small 68M drafter performs badly with a temperature mismatch due to its limited capacity, we use larger Llama3 models in the second table with a 1B drafter and 8B target. We take $K=2$ as in Tab. 2.
>
>    |Strategy|GSM8K BE|TR (%)|HumanEval BE|TR (%)|NaturalReasoning BE|TR (%)|
>    |-|-|-|-|--|-|-|
>    |SpecInfer|2.35±0.00|14.16±1.55|1.98±0.01|-0.29±0.66|2.14±0.01|11.44±0.73|
>    |SpecTr|2.35±0.01|13.41±2.10|1.99±0.01|-0.39±1.01|2.14±0.00|11.32±0.48|
>    |Ours|2.34±0.01|14.65±1.90|2.00±0.01|1.58±0.74|2.13±0.00|12.08±0.38|
>    |Daliri et al.|1.64±0.00|1.39±0.30|1.48±0.00|2.69±0.49|1.54±0.00|2.83±0.66|
>
>    |Strategy|Tmp. 1/2|GSM8K BE|TR (%)|HumanEval BE|TR (%)|MBPP BE|TR (%)|
>    |-|-|-|-|-|-|-|-|
>    |SpecInfer|0.5/1.0|2.21±0.01|-12.51±1.09|2.61±0.04|-4.73±1.67|2.46±0.03|-5.99±1.14|
>    ||1.0/0.5|2.44±0.02|-3.47±0.88|2.79±0.03|3.02±1.17|2.65±0.02|1.09±1.85|
>    ||1.0/1.0|2.57±0.01|2.11±0.73|2.94±0.02|7.50±1.04|2.79±0.01|6.86±1.23|
>    |Ours|0.5/1.0|2.67±0.03|5.88±1.46|3.63±0.02|33.85±1.11|3.33±0.02|25.71±1.81|
>    ||1.0/0.5|2.69±0.02|6.33±1.49|3.67±0.04|34.67±1.63|3.31±0.03|23.87±1.47|
>    ||1.0/1.0|2.88±0.01|13.00±1.12|3.72±0.03|35.90±1.40|3.51±0.03|32.20±2.37|
>
> 3. **Tightness of the LML:**
>    Thank you for mentioning the need for a theoretical analysis of our bound's tightness. As reviewer fX9A had similar concerns, we refer you to the discussion under item 2 of our response to their feedback above.
>
> 4. **Compression baselines:**
>    We clarify that both the baseline and GLS compression schemes make use of the *same VAE-based neural encoder/decoder pair*, which mirrors that adopted in Phan et al. [4] for a similar compression task.
>    That is, the weights and structure of the encoder and decoder networks, detailed in appendix D.3, are identical for both schemes. GLS itself is not a neural compressor but is rather an algorithm for the information-efficient communication of samples, which occurs *after* the VAE has produced embeddings from the source yet *before* the decoder network outputs its reconstruction. The decision to use GLS or the baseline scheme is therefore independent of the choice of neural compressor, VAE or otherwise. Using a more advanced compressor would likely benefit the rate-distortion performance of both GLS and the baseline, though we leave a full study of the interplay between sampling strategies and encoder/decoder architectures to future work.
>
> 5. **Ablation on drafter-invariance:**
>    We have conducted a new ablation study to show the practical benefits of drafter-invariance. Please find our results and discussion in item 3 of our response to reviewer Wv4i. We also have a concrete example where non-invariant speculative decoding causes the output to diverge from the original after changing the draft model, under item 6 of our response to reviewer Wfqz.
>
> 6. **Comparision of acceptance probabilities:**
>    We would like to stress that we give comparisons with non-GLS alternatives (SpecTr and SpecInfer) for all experiments in Tabs. 1 and 2, plus Tabs. 3 and 4 in the supplementary. In particular, the empirical token acceptance rate can be computed as $\Pr[\text{accept token}] = (\mathrm{BE} - 1) / L$, where $\mathrm{BE}$ is the block efficiency.
>    Note that due to extra steps needed to ensure sequence-level correctness, in particular checking $k \in \mathcal{S}$ on line 11 of Alg. 2, this is usually smaller than the one-step matching probability in Thm. 1.
>    For comparisons on the one-step matching probability, please refer to item 1 of our response to reviewer W4vi.
>
>    Our results in Tabs. 1 and 2 show that GLS achieves competitive performance in the i.i.d. case and often outperforms alternatives when the drafts are not identically distributed, both in terms of acceptance probabilities and wall-clock token rates.
>    Resource constraints (1×RTX6000 Ada GPU) are identical across these experiments.
>
>    Please note that the SpecTr and SpecInfer coupling methods are not suitable for distributed compression, since they would require both the encoder and decoder to have full knowledge of the other's distribution; conversely, a useful compression scheme must limit communication between the two. Hence we adopt a simpler baseline, which uses 1 instead of $K$ sets of common random numbers, for the compression experiments.
>
> 7. **Latency numbers and run-time benchmarks:**
>    We agree that run-time benchmarks are crucial for evaluating our algorithm.
>    With this in mind, we kindly point you to the results in Tabs. 1 and 2, plus Tabs. 3 and 4 in the supplementary, which contain our performance measurements.
>    All the experiments use the full Qwen2.5 vocabulary, which is large (containing 151,936 tokens).
>    In line with SpecInfer and SpecTr, we focused on token rates (TR, in tokens/second) and reported *relative speedups* instead of absolute measurements [3,6]. This allows for fairer comparisons, since the absolute token rate will vary depending on the hardware configuration and is thus not a good yardstick for judging the relative performance of different speculative decoding algorithms.
>
>    Nevertheless, we provide below versions of our Tabs. 1 and 2 from the main paper below with absolute latency numbers (in ms/token) instead of relative wall-clock percentage speedups. These tables do *not* represent new results, but are rather reinterpretations of the run-time benchmarks already provided.
>
>    |Strategy|GSM8K Latency|HumanEval Latency|NaturalReasoning Latency|
>    |-|-|-|-|
>    |SpecInfer|16.38±0.08|17.74±0.13|18.96±0.10|
>    |SpecTr|16.50±0.09|17.86±0.12|18.88±0.07|
>    |Ours|16.33±0.09|17.71±0.15|18.91±0.06|
>    |Daliri et al.|17.08±0.09|19.24±0.03|22.05±0.14|
>
>    |Strategy|Tmp. 1/2|GSM8K Latency|HumanEval Latency|MBPP Latency|
>    |-|-|-|-|-|
>    |SpecInfer|0.5/1.0|21.27±0.18|25.72±0.12|24.50±0.17|
>    ||1.0/0.5|20.36±0.24|23.96±0.24|22.83±0.20|
>    ||1.0/1.0|19.81±0.16|23.26±0.23|22.55±0.12|
>    |Ours|0.5/1.0|18.38±0.21|22.44±0.14|22.45±0.09|
>    ||1.0/0.5|18.43±0.13|22.73±0.24|22.36±0.20|
>    ||1.0/1.0|18.09±0.14|21.95±0.11|21.85±0.12|
>
> 8. **Scalability, algorithmic complexity and related limitations:**
>    Thank you for your interest in the details of our algorithm. While the $O(NK)$ space and time complexity may seem troubling if $N$ is very large, we note that SpecInfer also has the same complexity [3], which has not prevented its widespread adoption. By using top-K sampling as is standard in LLM inference, we can reduce the complexity to $O(MK)$ where $M \ll N$ is the number of high-probability tokens we wish to keep. These and similar "tricks" have meant that Gumbel-max sampling is already an efficient method for sampling from a single LLM [5], and our multi-draft extension can also benefit from them.
>
>    When considering the decoder-side computation in distributed compression, we have explicit control of $N$. As explained in appendices D.2 and D.3, it represents the number of shared samples taken from the prior distribution *before* decoding begins. Typically, this $N$ is quite small, e.g. $\leq 2^9$ in our MNIST experiments. As a result, the $O(NK)$ complexity does not pose difficulties in practice.
>
> 9. **Failure modes, e.g. decoding bias or performance degradation:**
>    First, we note that GLS provably maintains the target distribution. Therefore, our speculative decoding and compression schemes do not suffer from biased outputs. Second, you are right to mention that speculative decoding can sometimes cause degraded performance when the drafter and target are not well-aligned, e.g. with a large temperature mismatch. This limitation is not unique to GLS --- the latency numbers provided in item 7 above show that a large mismatch also increases the latency of SpecInfer and SpecTr. We will add more discussion of these limitations in a revised version of the paper.
>
> [1] M. Daliri, C. Musco, and A. T. Suresh. Coupling without communication and drafter-invariant speculative decoding, 2025. ArXiv: 2408.07978.
>
> [2] C. T. Li, and V. Anantharam. A unified framework for one-shot achievability via the Poisson matching lemma. *ISIT 2019*.
>
> [3] X. Miao, G. Oliaro, et al. Specinfer: accelerating large language model serving with tree-based speculative inference and verification. *ASPLOS 2024*.
>
> [4] B. Phan, A. Khisti, and C. Louizos. Importance matching lemma for lossy compression with side information. *AISTATS 2024*.
>
> [5] W. Kool, H. Van Hoof, and M. Welling. Stochastic beams and where to find them: the Gumbel-top-k trick for sampling sequences without replacement. *ICML 2019*.
>
> [6] Z. Sun, A. T. Suresh, et. al. SpecTr: fast speculative decoding via optimal transport. *NeurIPS 2023*.

---

> > ### Author Response · Authors · 2025-08-07
> >
> > Thank you once more for your in-depth feedback on our paper. Since the discussion period is ending soon, we would like to encourage you to respond to our rebuttal. As we value your input highly, your contribution to the discussion would be greatly appreciated and beneficial to our work. We are ready to answer any remaining questions or clarify any outstanding concerns, and if all are addressed to your satisfaction, we would appreciate it if you could consider increasing your score.

---

> ### Comment · Area_Chair_XapK · 2025-08-06
>
> Please respond to the authors' rebuttal and indicate if you are happy to reconsider your score based on the rebuttal and the other reviewers' comments.

---

### Official Review · Reviewer_fX9A · 2025-07-02

**Clarity:** 4
**Significance:** 3
**Originality:** 2
**Rating:** 5
**Confidence:** 3

**Summary:**

The paper presents Gumbel-max List Sampling (GLS), an extension of the single-draft Gumbel-max coupling of Daliri et al. to the case of $K$ drafts drawn from a “drafter” distribution $p$.
By sharing a structured pool of exponential noise across all drafts and the target draw from $q$, GLS maintains the correct marginals while boosting the probability that at least one draft matches the target.
The authors provide explicit lower bounds on the matching probability, which increase with $K$.
GLS is then applied to (i) multi-draft speculative decoding and (ii) single-encoder multi-decoder lossy compression, with supporting experiments in both settings.

**Questions:**

Q1: How does the matching probability bound behave in practice for different values of $K$?
Do we know anything about the tightness of the bound?

Q2: How was $K$ chosen in the experiments, and did you observe any practical trade-offs or patterns when varying $K$?

**Ethical Concerns:**

["NO or VERY MINOR ethics concerns only"]

**Final Justification:**

All my concerns were addressed. I remain at my initial assessment that I believe this is solid work that should be accepted.

**Limitations:**

yes

**Paper Formatting Concerns:**

no concerns

**Quality:**

4

**Strengths And Weaknesses:**

Strengths:

The paper is very well presented. Theoretical contributions are clearly motivated and placed in the context of related work. The proposed algorithm is simple and elegant, which is a strength.

Weaknesses:

Experimental evaluation is limited. While the applications are interesting, they are presented more as proofs of concept rather than full empirical studies.
There is limited discussion or analysis of the practical behaviour of GLS with respect to the number of drafts $K$, and other practical considerations.
Theorem 1 provides a useful, monotone lower bound on the match probability, but the paper does not analyse the tightness of this bound for $K>1$.

---

> ### Author Rebuttal · Authors · 2025-07-30
>
> Thank you for your thoughtful comments and feedback on our work. We are glad that you appreciated the simplicity of the GLS algorithm and the strength of our theoretical contributions. Please find our response to your concerns below:
>
> 1. **Behavior of the bound in practice for different values of K:**
>    We agree that computing the bound for different values of $K$ and comparing against the matching probabilities observed during experiments would strengthen our theoretical claims about the LML. To this end, we have used the target and proposal distributions for the next token encountered during speculative decoding experiments to compute the average empirical matching probability and its lower bound in Thm. 1 (i.e. Eq. 3). A table of results can be found in item 1 of our response to reviewer W4vi, where we test $K = 2, 4, 6, 8$. The table is also repeated below for your convenience.
>    The new results verify that our scheme's matching probability and the lower bound in Thm. 1 both increase with $K$, and the bound is quite close to the true probability in practice.
>
>    |Dataset|$K = 2$|$K = 2$, Eq. 3|$K = 4$|$K = 4$, Eq. 3|$K = 6$|$K = 6$, Eq. 3|$K = 8$|$K = 8$, Eq. 3|
>    |-|-|-|-|-|-|-|-|-|
>    |GSM8K|0.949|0.936|0.972|0.956|0.979|0.965|0.982|0.971|
>    |HumanEval|0.919|0.903|0.950|0.931|0.962|0.945|0.971|0.953|
>    |NaturalReasoning|0.862|0.844|0.905|0.884|0.926|0.904|0.937|0.916|
>
>    Below, we also give the corresponding matching probabilities for SpecInfer (SI) and SpecTr (ST), computed over the same next-token distributions. Our scheme's matching probability is similar to and in many cases exceeds that of SpecInfer and SpecTr, even though they do not offer any drafter-invariance property.
>
>    |Dataset|$K = 2$, SI|$K = 2$, ST|$K = 4$, SI|$K = 4$, ST|$K = 6$, SI|$K = 6$, ST|$K = 8$, SI|$K = 8$, ST|
>    |-|-|-|-|-|-|-|-|-|
>    |GSM8K|0.948|0.927|0.969|0.938|0.976|0.944|0.981|0.948|
>    |HumanEval|0.919|0.891|0.948|0.908|0.960|0.916|0.967|0.922|
>    |NaturalReasoning|0.864|0.837|0.904|0.864|0.921|0.878|0.932|0.887|
>
> 2. **Tightness of the LML bound:**
>    Our results in the table referenced above show that, in practice, the bound in Thm. 1 is quite close to the true matching probability for commonly encountered distributions such as those seen during LLM inference and draft verification. In response to reviewers' interest, we have also obtained some new results about the bound's theoretical tightness. In particular, we outline several situations below where our bound is provably tight for any $K$:
>
>    - In a pessimistic scenario where $p_X$ has all its mass on one element, the LML bound (Eq. 3) is tight regardless of $K$ and $q_Y$.
>    - In a best-case scenario where $p_i = q_i$ for all $i$, the LML bound and the true matching probability of GLS both equal 1.
>    - As you already alluded to in your comments, Eq. 3 yields the exact matching probability for $K = 1$.
>
>    We will include the required proofs in a revised version of the paper. Please note that the second, conditional version of the bound in Eq. 4 is a significant relaxation that is provided mostly to allow for a more intuitive expression when we write down the coding theorem of Prop. 4. It is generally quite loose, but lets us intuit the limiting behavior for large $K$. When tight bounds are needed, we advocate for the use of Eq. 3 instead, which also admits a conditional form by removing the outer summation and dividing by $q_j$.
>
> 3. **How K was chosen in the experiments:**
>    For the speculative decoding experiments, we tried $K = 2, 4, 6, 8$ --- similar values between 2 and 8 have commonly been used in prior work on multi-draft speculative decoding [1,2,3]. From Tab. 3 in the supplementary, either $K = 6$ or 8 tend to give the best performance in practice, depending on the dataset. This is again broadly consistent with prior works [1,2,3], which often find values of $K$ between 6 and 8 perform best.
>    We use $K = 8$ for Tab. 2 in the main paper as it best shows the relative performance of the different methods.
>    When we vary the draft temperature, we restrict ourselves to $K = 2$ to better isolate the effects of the temperature mismatch, following the basic experimental setup from Khisti et al. [4].
>
>    In the distributed compression experiments, $K$ is equal to the number of side information realizations and hence the number of decoders. Typically we do not have control over this parameter in practice, because the number of available side information samples is fixed.
>    With this in mind, we simply chose values of $K$ that would provide a representative spread of rate-distortion curves in Figs. 2d and 4.
>
> 4. **Practical trade-offs when varying K in applications of GLS:**
>    In speculative decoding, the choice of $K$ is a careful trade-off between the benefits of a higher acceptance probability and the additional overhead required to generate and verify more drafts in parallel [1]. We analyze this trade-off further in item 1 of our response to reviewer Wfqz. There, we conclude that increasing $K$ is beneficial for the token rate while there are still spare memory and parallel compute resources available, but the token rate starts decreasing if $K$ is set too high and these resources become over-subscribed. On our test hardware, this starts to occur near or above $K = 8$ in many cases.
>
>    As mentioned in item 3 above, we would typically have less control over $K$ in distributed compression settings, yet we can still make some interesting observations. As suggested by Prop. 4, the error probability and thus rate-distortion performance improves monotonically as $K$ increases. Meanwhile, the computational cost of running the scheme increases linearly with $K$ as the number of decoders increases. This was not a practical concern in our experiments, since the decoder networks are lightweight and moreover can be run independently in parallel, reducing the total decoder-side latency.
>
> [1] Z. Sun, A. T. Suresh, et al. SpecTr: fast speculative decoding via optimal transport. *NeurIPS 2023*.
>
> [2] W. Jeon, M. Gagrani, R. Goel, J. Park, M. Lee, and C. Lott. Recursive speculative decoding: accelerating LLM inference via sampling without replacement, 2024. ArXiv: 2402.14160.
>
> [3] X. Miao, G. Oliaro, et al. Specinfer: accelerating large language model serving with tree-based speculative inference and verification. *ASPLOS 2024*.
>
> [4] A. J. Khisti, M. R. Ebrahimi, et al. Multi draft speculative sampling: canonical decomposition and theoretical limits. *ICLR 2025*.

---

> > ### Comment · Reviewer_fX9A · 2025-08-04
> >
> > Thank you for the additional results and theoretical insights. I maintain my initial assessment that this is a technically solid paper.

---

> > > ### Author Response · Authors · 2025-08-06
> > >
> > > Thank you again for your review and positive assessment of our work. We are happy that you found our additional results useful and hope that they adequately addressed any remaining concerns.

---

### Official Review · Reviewer_Wfqz · 2025-07-02

**Clarity:** 3
**Significance:** 3
**Originality:** 3
**Rating:** 5
**Confidence:** 2

**Summary:**

This paper tackles the problem of coordinated sampling by introducing a list-based extension of the classic Gumbel-max coupling. The authors propose Gumbel-max List Sampling, which allows one party to generate K proposals and achieve a higher acceptance probability. They establish a List Matching Lemma that lower-bounds this probability and demonstrate two key applications: 1. A multi-draft speculative decoding scheme for large language models (LLMs) that is simple to implement, competitive with SpecTr and SpecInfer. 2. A distributed lossy compression mechanism with independent side information at K decoders, showing rate–distortion gains on synthetic Gaussian sources and MNIST images.

**Questions:**

1.	Overhead Analysis: How does the wall-clock latency and memory usage of multi-draft speculative decoding scale with K in practice, especially for large transformer-based LLMs?
2.	Choice of K: Do the authors have guidelines or heuristics for selecting the number of proposals K to balance acceptance probability gains against computational cost?
3.	Generalization to Correlated Side Information: How would the compression scheme adapt if decoders’ side information are not independent but partially correlated?
4.	Ablation on Drafter-Invariance: Can the authors quantify the practical impact of drafter-invariance on output quality and convergence, compared to non-invariant multi-draft baselines?

**Ethical Concerns:**

["NO or VERY MINOR ethics concerns only"]

**Final Justification:**

This is a technically solid paper introducing Gumbel-max List Sampling, a novel extension of the Gumbel-max coupling with formal guarantees and broad applicability. The theoretical contributions are clear, and the applications to speculative decoding and distributed lossy compression are well-motivated and convincingly demonstrated. Following the rebuttal, the authors have properly addressed my concerns. I recommend acceptance as my final rating.

**Limitations:**

yes

**Paper Formatting Concerns:**

No formatting concerns.

**Quality:**

4

**Strengths And Weaknesses:**

Strengths

1.	Novel Theoretical Framework: GLS generalizes single-proposal Gumbel-max coupling to the multi-proposal setting, with a clear algorithm (Algorithm 1) and a rigorous lower bound on acceptance (Theorem 1).
2.	Formal Guarantees: Introduction of List Matching Lemma and a token-level acceptance bound for speculative decoding gives the scheme provable reliability properties absent in prior heuristic approaches.
3.	Broad Applicability: Beyond decoding acceleration, GLS is applied to distributed lossy compression, demonstrating versatility across machine-learning and information-theory domains.

Weaknesses:

1.	Scalability of K: While acceptance probability grows with the number of proposals K, the computational and memory overhead of generating and verifying multiple drafts is not deeply analyzed.
2.	Continuous Distributions: The extension to continuous distributions via importance sampling is only sketched. Empirical validation on continuous tasks is absent.
3.	Compression Experiments: Lossy compression results are limited to synthetic Gaussians and MNIST; it remains unclear how GLS-based compression performs on colored high-resolution or more complex datasets (e.g., CIFAR-10 or ImageNet).
4.	Side-Information Assumptions: The distributed compression model assumes independent side information across decoders. Real-world scenarios often involve correlated side information, which may degrade performance.

---

> ### Author Rebuttal · Authors · 2025-07-30
>
> Thank you for taking the time to review our paper.  We greatly appreciate your suggestions on how to improve our work and are pleased that you found the theoretical framework of GLS novel and rigorous, and the proposed applications relevant and interesting. We have addressed your comments below:
>
> 1. **Practical scalability with K and overhead analysis:**
>    As you point out, increasing $K$ boosts the acceptance probability but demands more resources, especially when using large LLMs. The memory consumption and algorithmic complexity both increase linearly with $K$, since $K$ individual drafts must be generated, stored and verified. However, multi-draft speculative decoding leverages the fact that these operations can be done in parallel for all $K$ drafts simultaneously [1]. Therefore, increasing $K$ decreases the wall-clock latency in the real world so long as there are spare memory and parallel compute resources available, which is usually true for non-batched inference on a single GPU.
>
>    When $K$ is made too large, memory and compute become over-subscribed and latency can increase. Our results in Tab. 3 in the supplementary give empirical evidence of this phenomenon. Increasing $K$ from 2 to 6 gives consistent wall-clock speedups, yet the token rate sometimes decreases for certain datasets when going further to $K = 8$, as the increased overhead starts to outweigh the benefits of a higher acceptance probability.
>
> 2. **Guidelines on choosing K :**
>    As mentioned in our previous response, the value of $K$ must carefully balance gains in the acceptance probability against limits on available compute and memory resources. Therefore, the choice is necessarily hardware-dependent.
>    Nevertheless, our results in table 3 in the supplementary indicate that $K = 6$ or 8 is usually a good choice to maximize the token rate. Prior works on multi-draft speculative decoding come to similar conclusions about the best choice of $K$ [1,2,3], suggesting that our recommendation should generalize to a range of current hardware platforms.
>
> 3. **Applicability to correlated side information:**
>    While our theoretical discussion focuses on (conditionally) independent side information for the sake of simplicity and tractability, on the practical side our MNIST experiment *does* deal with partially correlated side information, providing empirical evidence that GLS works well in this case. More precisely, the $14 \times 14$ slices taken from the left-hand side of the image are often overlapping and therefore not independent. Nevertheless, the compression performance holds up well, with the distortion decreasing consistently in both the rate and number of side information samples as seen in Fig. 4.
>
> 4. **Clarification on the extension to continuous distributions:**
>    Thank you for raising this important point. While the main paper provides only a sketch of the extension to continuous distributions, a more complete description and mathematical analysis is given in Appendix C (pp. 30--32). Regarding empirical validation, we would like to emphasize that the Gaussian source and MNIST experiments are *both* continuous tasks --- the information sources are continuous-valued in both cases, validating that our extension to continuous distributions via importance sampling works well in practice.
>
> 5. **Additional compression experiments:**
>    To show that GLS generalizes to more complex datasets, we have completed a new image compression experiment on CIFAR-10 using the same basic setup as our MNIST experiment. The side information samples are now $16\times16$ patches randomly selected from the left half of the (3-channel, $32 \times 32$) image. Please find the results in the table below, showing the MSE distortion. We want to emphasize that the baseline and GLS methods are functionally identical for $K = 1$, but as is the case on MNIST, GLS offers gains for $K > 1$.
>
>    |Rate (bits)|$K = 1$, GLS|$K = 1$, BL|$K = 2$, GLS|$K = 2$, BL|$K = 3$, GLS| $K = 3$, BL|$K = 4$, GLS|$K = 4$, BL|
>    |-|-|-|-|-|-|-|-|-|
>    |2|0.1119±0.0003|0.1113±0.0001|0.0925±0.0001|0.1019±0.0003|0.0839±0.0001|0.0982±0.0002|0.0788±0.0001|0.0956±0.0002|
>    |3|0.1054±0.0002|0.1048±0.0002|0.0860±0.0002|0.0958±0.0002|0.0778±0.0003|0.0914±0.0002|0.0733±0.0001|0.0888±0.0002|
>    |4|0.0963±0.0001|0.0959±0.0003|0.0787±0.0003|0.0867±0.0003|0.0710±0.0001|0.0831±0.0002|0.0673±0.0002|0.0812±0.0001|
>    |5|0.0846±0.0003|0.0846±0.0003|0.0698±0.0003|0.0767±0.0001|0.0641±0.0000|0.0734±0.0003|0.0618±0.0001|0.0718±0.0003|
>    |6|0.0710±0.0001|0.0710±0.0002|0.0616±0.0002|0.0659±0.0002|0.0590±0.0001|0.0639±0.0001|0.0579±0.0001|0.0629±0.0001|
>
> 6. **Ablation on drafter-invariance:**
>    Thank you for raising this important point. We agree that it is important to provide experimental results showing the real-world impact of drafter-invariance, in particular our claim that it reduces the sensitivity of the generated outputs to changes affecting the draft model.
>
>    In response, we have provided a new ablation study that examines this question and shows that our approach promotes decoding consistency when the draft model changes, as measured by ROUGE scores [4], compared to non-invariant multi-draft speculative decoding baselines. Please find details of the new experiments and tabulated results in item 3 of our response to reviewer W4vi. The table is also repeated below for your convenience.
>
>    |Dataset|Decoding Algorithm|ROUGE-1|ROUGE-2|ROUGE-L|
>    |-|-|-|-|-|
>    |GSM8K|SpecInfer|0.737±0.004|0.592±0.007|0.647±0.008|
>    ||Ours|0.801±0.005|0.701±0.011|0.745±0.009|
>    |HumanEval|SpecInfer|0.656±0.003|0.466±0.006|0.545±0.005|
>    ||Ours|0.710±0.001|0.555±0.003|0.628±0.002|
>    |MBPP|SpecInfer|0.676±0.004|0.497±0.003|0.580±0.004|
>    ||Ours|0.728±0.006|0.588±0.011|0.659±0.010|
>
>    While our ablation study provides a more rigorous way to examine the real-world impact of drafter-invariance, we also give a concrete example from GSM8K where non-invariant speculative decoding causes the output to diverge significantly from the original after changing the draft model but keeping the same random seed. For this experiment we modify the draft model by decreasing its temperature. We stress that all forms of speculative decoding preserve the output distribution exactly; therefore, it is not the *quality* that is degraded, but rather the *stability* or *consistency* of the outputs. In this particular example, our scheme achieves perfect consistency between the two trials; we do not guarantee this in general, but the ROUGE scores in our ablation study demonstrate that we achieve better consistency than existing schemes on average.
>
>    - *SpecInfer (not invariant), draft temp. 1.0:*
>       ```
>       To determine how much Weng earned for babysitting for 50 minutes, we first need to convert the time from minutes to hours. We know that there are 60 minutes in an hour, so we can convert 50 minutes to hours by dividing 50 by 60:
>       \[
>       \text{Time in hours} = \frac{50}{60} = \frac{5}{6} \text{ hours}
>       \]
>       Next, we calculate the earnings by multiplying the time in hours by her hourly rate of $12:
>       \[
>       \text{Earnings} = 12 \times \
>       ```
>    - *SpecInfer (not invariant), draft temp. 0.5:*
>       ```
>       To determine how much Weng earned for her 50 minutes of babysitting, we need to follow these steps:
>       1. Identify Weng's hourly wage.
>       2. Convert the babysitting time from minutes to hours.
>       3. Calculate the earnings based on the time worked.
>       First, Weng's hourly wage is $12. Since she babysat for 50 minutes, we need to convert 50 minutes into hours. There are 60 minutes in an hour, so we can convert 50 minutes to hours by dividing 50 by 60:
>       \[ \text{Time in hours} = \frac{
>       ```
>    - *Our scheme (invariant), draft temp 1.0:*
>       ```
>       To determine how much Weng earned for 50 minutes of babysitting at a rate of $12 per hour, we need to follow these steps:
>       1. Convert the time from minutes to hours.
>       2. Multiply the time in hours by the hourly rate.
>       First, we convert 50 minutes to hours. Since there are 60 minutes in an hour, we divide 50 by 60:
>       \[
>       \frac{50}{60} = \frac{5}{6} \text{ hours}
>       \]
>      Next, we multiply the time in hours by the hourly rate of $12:
>       ```
>    - *Our scheme (invariant), draft temp. 0.5:*
>       ```
>       To determine how much Weng earned for 50 minutes of babysitting at a rate of $12 per hour, we need to follow these steps:
>       1. Convert the time from minutes to hours.
>       2. Multiply the time in hours by the hourly rate.
>       First, we convert 50 minutes to hours. Since there are 60 minutes in an hour, we divide 50 by 60:
>       \[
>       \frac{50}{60} = \frac{5}{6} \text{ hours}
>       \]
>       Next, we multiply the time in hours by the hourly rate of $12:
>       ```
>
> [1] Z. Sun, A. T. Suresh, et al. SpecTr: fast speculative decoding via optimal transport. *NeurIPS 2023*.
>
> [2] W. Jeon, M. Gagrani, R. Goel, J. Park, M. Lee, and C. Lott. Recursive speculative decoding: accelerating LLM inference via sampling without replacement, 2024. ArXiv: 2402.14160.
>
> [3] X. Miao, G. Oliaro, et al. Specinfer: accelerating large language model serving with tree-based speculative inference and verification. *ASPLOS 2024*.
>
> [4] C. Y. Lin. ROUGE: A package for automatic evaluation of summaries. *Text Summarization Branches Out*, 2004.

---

> > ### Comment · Reviewer_Wfqz · 2025-08-06
> >
> > Thank you for providing the additional results and theoretical insights. These reaffirm my initial assessment that the paper is technically strong.

---

> > > ### Author Response · Authors · 2025-08-06
> > >
> > > Thank you again for your feedback and positive assessment of our paper. We are glad that our additional results were able to address your questions and concerns.

---

> ### Comment · Area_Chair_XapK · 2025-08-06
>
> Please respond to the authors' rebuttal and indicate if you are happy to reconsider your score based on the rebuttal and the other reviewers' comments.

---

### Official Review · Reviewer_W4vi · 2025-07-03

**Clarity:** 3
**Significance:** 3
**Originality:** 3
**Rating:** 4
**Confidence:** 4

**Summary:**

This paper proposes Gumbel-max List Sampling (GLS), a method for coupling multiple samples from one distribution with a target sample from another, without requiring communication. It extends the classical Gumbel-max trick to improve the probability that at least one sample matches the target. The authors prove a theoretical lower bound on the match probability (List Matching Lemma), and apply GLS to two tasks: drafter-invariant speculative decoding for language models and distributed lossy compression with decoder-specific side information. GLS achieves comparable performance to baselines while offering better structural simplicity and system compatibility.

**Questions:**

- The authors are encouraged to include experimental results that illustrate how the actual matching probability scales with the number of samples \(K\), and how closely it aligns with the theoretical bound provided by the List Matching Lemma. This would significantly strengthen the empirical foundation of their core theoretical claim.

- A discussion of the likelihood that different draft models (or configurations) generate the same token sequences, along with an empirical analysis of whether such sequences lead to consistent decoding outputs, would help clarify the practical robustness of the proposed draft invariance property.

- The paper should clarify whether the sampling procedure for the target token \(Y\) is compatible with deployment in distributed or asynchronous systems. A discussion of how GLS can be implemented across heterogeneous devices without violating its assumptions would enhance the practical relevance of the method.

- Since GLS relies on shared floating-point computations (e.g., \(-\log U\), division, and \(\arg\min\)), it remains unclear whether decoding outputs are fully reproducible across devices with different hardware or numerical precision. The authors are encouraged to discuss how such numerical variation might affect decoding consistency in practice.


- I have a question about the determinism and decoding consistency of GLS in practice. Since the GLS procedure relies on floating-point operations, it’s possible that different devices may produce slightly different results even when using the same random seed. Have the authors considered the impact of hardware-level numerical differences on decoding consistency? If GLS is to be used in distributed or heterogeneous environments, this could affect reproducibility or correctness of the decoding outputs.

**Ethical Concerns:**

["NO or VERY MINOR ethics concerns only"]

**Final Justification:**

The paper introduces a technically solid and elegant generalization of the Gumbel-max trick with a rigorous theoretical guarantee (List Matching Lemma), and demonstrates its practical relevance in speculative decoding and distributed compression. The rebuttal convincingly addressed key concerns, providing additional empirical validation of matching probabilities and an ablation on drafter-invariance. While practical deployment limitations (e.g., floating-point consistency, shared randomness) remain, these are not unique to GLS but apply broadly to Gumbel-max–based methods. Overall, the combination of theoretical contribution, simplicity, and practical applicability makes this a valuable addition to the literature. I recommend acceptance.

**Limitations:**

While the paper is technically well-executed and does not raise immediate ethical concerns, the authors do not explicitly address the limitations of their work. Given that the proposed method is designed for large language model inference and distributed compression, it would be appropriate to discuss certain assumptions, such as the need for perfectly shared randomness and consistent numerical behavior across devices, which may limit practical deployment. The authors are encouraged to include a brief discussion acknowledging these limitations and potential risks to support responsible and transparent adoption of their approach.

**Quality:**

3

**Strengths And Weaknesses:**

### Strengths
- The authors propose a clean generalization of Gumbel-max sampling and back it with a nontrivial theoretical guarantee (List Matching Lemma) on the matching probability. The proposed GLS method is simple to implement, and its probabilistic correctness is rigorously proven. Experiments are conducted on both language modeling and compression tasks, demonstrating reasonable practical viability.

- The method addresses practically relevant problems in speculative decoding and distributed compression, both of which are increasingly important in large-scale, resource-constrained, or distributed systems. The approach provides system-level benefits like deterministic decoding and model-agnostic draft compatibility.

- The paper is clearly written and well-structured. The authors concisely explain the algorithm, provide motivating examples, and include relevant background. The connection between the theoretical core and practical applications is explicitly laid out.

### Weakness

- The paper centers on improving match probability via List Matching Lemma, but it does not report actual acceptance rates or how they scale with the number of proposals K. This weakens the empirical support for its theoretical contribution.

- Although the method is framed as "communication-free", the target sample Y is computed via a minimum over all proposal-side Gumbel variables, meaning it is structurally coupled with all proposal samples. This may challenge the interpretation of independence or decoupled sampling in practical implementations.

- The drafter-invariance claim only holds if the draft token sequences are fixed. In practice, changing the draft model even slightly can alter the token sequence, breaking this invariance. No ablation is conducted to examine the robustness of this property.

- The speculative decoding experiments report throughput (tokens/sec), but do not assess generation quality, token correctness, or how many tokens are accepted without fallback. This limits understanding of the method’s true effectiveness. Also, the distributed compression experiments evaluate distortion but do not measure actual bitrates or code length, leaving the full rate-distortion trade-off unquantified.

---

> ### Author Rebuttal · Authors · 2025-07-30
>
> Thank you for your insightful feedback on our paper. We are glad that you liked the simplicity and practical relevance of GLS and we found your comments about deployment to real-world distributed systems especially thought-provoking. Please find your concerns addressed below.
>
> 1. **Empirical matching probability, comparison to the LML bound and scaling with K:**
>    Thank you for your suggestion, which we agree will strengthen our main claims around the LML. We have provided further results comparing the empirical average matching probability (using target and proposal distributions for the next token seen during speculative decoding) to our lower bound in Thm. 1, testing $K = 2, 4, 6, 8$.
>    The new results confirm that the matching probability and lower bound increase monotonically with $K$, and the bound is quite close to the true probability in practice.
>
>    |Dataset|$K = 2$|$K = 2$, Eq. 3|$K = 4$|$K = 4$, Eq. 3|$K = 6$|$K = 6$, Eq. 3|$K = 8$|$K = 8$, Eq. 3|
>    |-|-|-|-|-|-|-|-|-|
>    |GSM8K|0.949|0.936|0.972|0.956|0.979|0.965|0.982|0.971|
>    |HumanEval|0.919|0.903|0.950|0.931|0.962|0.945|0.971|0.953|
>    |NaturalReasoning|0.862|0.844|0.905|0.884|0.926|0.904|0.937|0.916|
>
>    For completeness, we also show below the same empirical matching probabilities for SpecInfer (SI) and SpecTr (ST). Our approach performs similarly to or in many cases better than these methods, even though they do not offer any drafter-invariance property.
>
>    |Dataset|$K = 2$, SI|$K = 2$, ST|$K = 4$, SI|$K = 4$, ST|$K = 6$, SI|$K = 6$, ST|$K = 8$, SI|$K = 8$, ST|
>    |-|-|-|-|-|-|-|-|-|
>    |GSM8K|0.948|0.927|0.969|0.938|0.976|0.944|0.981|0.948|
>    |HumanEval|0.919|0.891|0.948|0.908|0.960|0.916|0.967|0.922|
>    |NaturalReasoning|0.864|0.837|0.904|0.864|0.921|0.878|0.932|0.887|
>
> 2. **Clarification of our "communication-free" claim:**
>    We stress that "communication-free" in the context of both our paper and Daliri et al. [1] does *not* mean that $Y$ is independent of the proposal samples. Indeed, the coupling of these variables is exactly what improves the matching probability beyond naïve independent sampling. Rather, our "communication-free" claim reflects the fact that $Y$ is selected *without any knowledge of the proposal distribution* $p_X$ and vice versa.
>
> 3. **Ablation and empirical analysis of drafter-invariance:**
>    Thank you for raising the need for more evidence around the robustness and practical importance of our conditional drafter-invariance property. As you point out, the output is no longer invariant if the input sequences change. Nevertheless, we find that adopting drafter-invariant speculative decoding significantly improves decoding consistency in practice, which we show through a new ablation study below. Moreover, in appendix B we demonstrate a slightly modified scheme that provides what we call *strong drafter-invariance*, which provably guarantees perfect decoding consistency even if the proposal samples change. However, this alternative scheme significantly reduces the block efficiency, as seen in Tabs. 3 and 4.
>
>    To quantify the effect of drafter-invariance on decoding consistency, we run 200 prompts from three different datasets using $K = 2$ drafts. Each prompt is run twice using the same random seed. The drafts are i.i.d., but the drafter model temperature is changed from 1.0 in the first run to 0.5 in the second. We compute ROUGE scores [2] to measure the similarity of the two output sequences; perfect decoding consistency would be a score of 1.0 for all of the ROUGE-1, ROUGE-2 and ROUGE-L metrics. We set $L = 4$ and average over 4 random seeds. Our conditionally invariant scheme gives higher similarity scores in each case compared to the non-invariant SpecInfer baseline, confirming that drafter-invariance provides more consistent outputs when faced with changes to the draft model, even though our notion is not strong enough to guarantee that the outputs always remain exactly the same.
>
>    |Dataset|Decoding Algorithm|ROUGE-1|ROUGE-2|ROUGE-L|
>    |-|-|-|-|-|
>    |GSM8K|SpecInfer|0.737±0.004|0.592±0.007|0.647±0.008|
>    ||Ours|0.801±0.005|0.701±0.011|0.745±0.009|
>    |HumanEval|SpecInfer|0.656±0.003|0.466±0.006|0.545±0.005|
>    ||Ours|0.710±0.001|0.555±0.003|0.628±0.002|
>    |MBPP|SpecInfer|0.676±0.004|0.497±0.003|0.580±0.004|
>    ||Ours|0.728±0.006|0.588±0.011|0.659±0.010|
>
> 4. **Clarification on generation quality, number of accepted tokens, bitrates and other experimental observations:**
>    Thank you for pointing out several details about our experimental results that deserve clarification. We address these separately below:
>
>    - *Generation quality and token correctness:*
>      Speculative decoding, including the original single-draft formulation [3] and multi-draft generalizations [4], comes with an *exact sampling guarantee*, as does our algorithm (Prop. 1). The probability distribution over the output tokens is unchanged from the target $\mathcal{M}_b$, and the output is thus statistically indistinguishable from what would be obtained using standard autoregressive decoding with $\mathcal{M}_b$. Hence, our method *provably* incurs no change in generation quality or token correctness, which is a property shared by existing speculative decoding algorithms.
>    - *Number of tokens accepted without fallback:*
>      This is the same as the block efficiency minus one, which we report in Tabs. 1 and 2 in the main paper and Tabs. 3 and 4 in the supplementary.
>    - *Bitrate and code length:*
>       We apologize that these quantities were not made clear enough in the paper. In our compression scheme, the only communication from the encoder to the decoder is the fixed-length message $M$, which has code length $\log_2(L_{\mathrm{max}})$. This is exactly the number of bits per transmission i.e. the rate, which is reported on the $x$-axis of our rate-distortion plots.
>
> 5. **Considerations and limitations for deployment on distributed or heterogeneous systems:** Thank you for your interest in the real-world deployment of our algorithm. We present our response in two parts, and will include a discussion of these issues in a revised version of the paper:
>
>    - *Availability of shared random numbers:*
>       As you mention, our algorithm assumes that both parties (i.e. encoder and decoder, or drafter and target) have access to a source of common random numbers. In practice, this is easily achieved by sharing or agreeing upon a random seed before beginning the procedure [8]. Since the seed is typically a single integer, the communication cost is small and is incurred only once, making implementation on distributed systems very much possible.
>    - *Consistency of floating-point operations:*
>       We would like to first point out that most speculative decoding applications currently use single-GPU or homogeneous multi-GPU architectures, where we can assume identical floating-point behavior [5,6]. While extensions to heterogeneous environments might make for interesting future work, we did not explicitly consider such settings in our paper and instead chose to limit our attention to the more common single-GPU case. On the other hand, our compression application is more likely to see deployment in distributed and/or heterogeneous systems. Nevertheless, the encoder and decoders can often be assumed to share the same floating-point behavior, as for example enforced by the IEEE 754 standard.
>
>      If this is not the case, you are right to point out that floating-point errors may cause the behavior of GLS to differ from that predicted by the theory. Nevertheless, given the popularity of Gumbel-max sampling in machine learning [7], there are common "tricks" to improve numerical stability which we adopt in our code. In particular, unnormalized log probabilities, read directly from the output layer of a neural network, are used instead of $p_X$ and $q_Y$ and the selection step becomes $Y = \arg\max_{1 \leq i \leq N} [\ln q_i + \max_{1 \leq k \leq K} [-\ln(-\ln(U_i^{(k)}))]]$, $X^{(k)} = \arg\max_{1 \leq i \leq N} [\ln p_i - \ln(-\ln(U_i^{(k)}))]$ [1]. This removes floating-point division and the need to compute softmax probabilities, reducing the impact of floating-point errors.
>
> [1] M. Daliri, C. Musco, and A. T. Suresh. Coupling without communication and drafter-invariant speculative decoding, 2025. ArXiv: 2408.07978.
>
> [2] C. Y. Lin. ROUGE: A package for automatic evaluation of summaries. *Text Summarization Branches Out*, 2004.
>
> [3] Y. Leviathan, M. Kalman, and Y. Matias. Fast inference from transformers via speculative decoding. *ICML 2023*.
>
> [4] A. J. Khisti, M. R. Ebrahimi, et al. Multi draft speculative sampling: canonical decomposition and theoretical limits. *ICLR 2025*.
>
> [5] Y. Li, F. Wei, C. Zhang, and H. Zhang. EAGLE: speculative sampling requires rethinking feature uncertainty. *ICML 2024*.
>
> [6] T. Cai, Y. Li, Z. Geng, H. Peng, and J. D. Lee. Medusa: simple LLM inference acceleration framework with multiple decoding heads. *ICML 2024*.
>
> [7] W. Kool, H. Van Hoof, and M. Welling. Stochastic beams and where to find them: the Gumbel-top-k trick for sampling sequences without replacement. *ICML 2019*.
>
> [8] L. Theis and N. Yosri. Algorithms for the communication of samples. *ICML 2022*.

---

> > ### Author Response · Authors · 2025-08-07
> >
> > Thank you once again for your insightful review of our paper. As the discussion period is coming to a close, we kindly encourage you to respond to our rebuttal --- we value your feedback highly and would appreciate your contribution to the discussion. We are happy to answer any remaining questions, and if all concerns are addressed and you are fully satisfied, we would appreciate it if you could consider increasing your score.

---

> > > ### Comment · Reviewer_W4vi · 2025-08-08
> > >
> > > The rebuttal successfully addresses several of my concerns, especially by providing empirical match probabilities and a detailed ablation on drafter-invariance. That said, I still believe the paper would benefit from a clearer discussion of its limitations and assumptions (e.g., floating-point consistency, shared randomness, input determinism), ideally in the main text. While the clarifications are sound, I remain slightly concerned about practical deployment and reproducibility in heterogeneous systems. Unless other reviewers raise stronger points for upgrade, I would be inclined to keep my current score.

---

> > > > ### Author Response · Authors · 2025-08-08
> > > >
> > > > Thank you for acknowledging that our rebuttal addressed your concerns. We are glad that our additional empirical results on matching probabilities and the ablation on drafter-invariance were able to clarify key points that you raised in your original review.
> > > >
> > > > We would like to stress that a discussion of the limitations mentioned in our rebuttal, namely the shared-randomness requirement and the assumption of consistent and deterministic floating-point computations, will be included in the final version of the main paper as you suggested. As done in the rebuttal, we will include in the discussion our justifications for making these assumptions with reference to common settings for speculative decoding [1,2], Gumbel-max sampling [3,4] and common practices in similar compression algorithms [5,6].
> > > >
> > > > Finally, please note that these concerns are not limited to our approach (GLS) but are rather applicable to Gumbel-max sampling in general, as well as any communication or sampling scheme based on common randomness more generally. Such methods nevertheless remain widely useful in coordinated sampling [7,8], machine learning [4] and compression [5,6].
> > > >
> > > > Thank you once again for your insightful review and positive assessment of our paper.
> > > >
> > > > [1] Y. Li, F. Wei, C. Zhang, and H. Zhang. EAGLE: speculative sampling requires rethinking feature uncertainty. *ICML 2024*.
> > > >
> > > > [2] T. Cai, Y. Li, Z. Geng, H. Peng, and J. D. Lee. Medusa: simple LLM inference acceleration framework with multiple decoding heads. *ICML 2024*.
> > > >
> > > > [3] M. Daliri, C. Musco, and A. T. Suresh. Coupling without communication and drafter-invariant speculative decoding, 2025. ArXiv: 2408.07978.
> > > >
> > > > [4] W. Kool, H. Van Hoof, and M. Welling. Stochastic beams and where to find them: the Gumbel-top-k trick for sampling sequences without replacement. *ICML 2019*.
> > > >
> > > > [5] B. Phan, A. Khisti, and C. Louizos. Importance matching lemma for lossy compression with side information. *AISTATS 2024*.
> > > >
> > > > [6] L. Theis and N. Yosri. Algorithms for the communication of samples. *ICML 2022*.
> > > >
> > > > [7] M. Bavarian, B. Ghazi, E. Haramaty, P. Kamath, R. L. Rivest, and M. Sudan. Optimality of correlated sampling strategies. *Theory of Computing*, 2020.
> > > >
> > > > [8] P. Cuff. Communication requirements for generating correlated random variables. *ISIT 2008*.

---

> ### Comment · Area_Chair_XapK · 2025-08-06
>
> Please respond to the authors' rebuttal and indicate if you are happy to reconsider your score based on the rebuttal and the other reviewers' comments.

---

### Note · Authors · 2025-08-12

We thank all reviewers for their feedback, which has prompted new results and discussions that will be added to the final paper. We would like to use this space to summarize what we consider our paper's main strengths as seen by reviewers, key points made during the rebuttal and our responses.

### Main Strengths

1. *Theoretical Framework:* It was broadly agreed that GLS provides a novel generalization of single-proposal Gumbel-max sampling to multiple samples.
2. *Matching Probability Guarantee:* The main analysis, in which we state and derive a nontrivial bound (LML) on the GLS matching probability, was well-received. Our compression error probability bound was also noted to be mathematically sound.
3. *Simplicity, Clarity:* Several reviewers highlighted that GLS is elegant, clear and simple to implement.
4. *Applications' Breadth, Relevance:* Applications to speculative decoding and compression were seen as relevant, showing the versatility of GLS in machine learning and information theory.

### Key Points of Discussion and Responses

1. *Empirical Matching Probability:* We gave more data on the empirical matching probability, showing that it increases with $K$ as predicted by our theory and is well-approximated by the LML.
2. *Drafter-Invariance Ablation:* We justified drafter-invariance via a new ablation study which measures ROUGE scores between outputs to show that drafter-invariance indeed encourages better decoding consistency when the draft model changes.
3. *Expanded Experiments:* We additionally tested image compression on CIFAR-10 and speculative decoding on Llama models; the results align with our existing ones. We clarified why our benchmarks, e.g. the baseline compression scheme, are appropriate.
4. *Overhead Analysis, Choosing $K$:* We further discussed performance concerns, including computational complexity comparisons to existing multi-draft algorithms, and mentioned practical strategies to reduce complexity. We also gave advice on choosing $K$ amid the tradeoff between higher acceptance rates and increased overhead.
5. *Implementation in Distributed Systems:* While these concerns are not unique to GLS and are shared by all approaches using Gumbel-max sampling or shared randomness, we pointed out that any implementation techniques used in such existing algorithms are immediately applicable to GLS, and gave examples. A discussion of implications for deployment, limitations and necessary assumptions will be included in the final paper.

---

### Decision · Program_Chairs · 2025-09-17

**Decision:**

Accept (poster)

**Comment:**

Paper summary:

The paper tackles the problem of coordinated sampling by introducing a list-based extension of the classic Gumbel-max coupling. The authors propose Gumbel-max List Sampling, which allows one party to generate K proposals and achieve a higher acceptance probability. They establish a List Matching Lemma that lower-bounds this probability and demonstrate two key applications: 1. A multi-draft speculative decoding scheme for large language models (LLMs) that is simple to implement, competitive with SpecTr and SpecInfer. 2. A distributed lossy compression mechanism with independent side information at K decoders, showing rate–distortion gains on synthetic Gaussian sources and MNIST images.

Summary of the discussion:

3 reviewers vote for acceptance, with one of them weakly. One more reviewer votes weakly for rejection. This last reviewer did not respond to the authors' rebuttal and did not engage in the discussion phase. The authors' rebuttal seems to address the reviewer's concerns.

Recommendation:

This is a borderline paper with 3 reviewers voting for acceptance (one weakly) and another reviewer voting weakly for rejection. However, the latter reviewer did not participated in the discussion or engaged with the authors who seemed to address their concerns in the rebuttal. Based on this, I recommend accepting the paper and encourage the authors to use the feedback provided to improve the paper for its camera ready version.